# A Study on the In-Plane Shear-after-Impact Properties of CFRP Composite Laminates

**DOI:** 10.3390/ma15145029

**Published:** 2022-07-20

**Authors:** Longquan Liu, Wenjun Xu

**Affiliations:** School of Aeronautics and Astronautics, Shanghai Jiao Tong University, Shanghai 200240, China; xuwenjun@sjtu.edu.cn

**Keywords:** shear-after-impact (ShAI), picture frame test, composite laminate, finite element analysis, two-step model

## Abstract

Impact loading on carbon fiber reinforced polymer matrix (CFRP) composite laminates can result in a significant reduction in their residual properties, and the (ShAI) properties of the composite material are essential to obtain the material allowable values of the shear dominated composite structures. In order to obtain the ShAI properties of the composite material in pure shear stress at a coupon level, this study presents theoretical, experimental, and numerical methods and analysis work on the in-plane shear and ShAI properties of the composite laminates. Theoretically, a method of sizing the composite specimen loading in shear is developed through comparing the load values due to buckling and the material failure. Following this, both impact tests using the drop-weight method and ShAI tests using the picture frame test method are conducted, and the influences of the impact energies on the impact damage and the residual ShAI values are evaluated. Moreover, a progressive failure finite element model based on the Hashin’s failure criterion and the cohesive zone model is developed, and a two-step dynamic analysis method is performed to simulate the failure process of the composite laminates under impact loading and ShAI loading. It is found that the impact damage with the cut-off energy, 50 J, causes a 26.8% reduction in the residual strength and the residual effective shear failure strain is about 0.0132. The primary reason of the shear failure is the propagation of both the matrix tensile failure and interlaminar delamination. It can be concluded that the proposed theoretical, experimental, and numerical methods are promising factors to study the ShAI properties of the composite materials.

## 1. Introduction

Carbon fiber reinforced polymer matrices have been employed in an ever-increasing way in aircraft structures due to their excellent properties, such as specified strength and stiffness, as well as their abilities that allow designers to tailor the layups to obtain an optimized structure for particular applications, etc. However, polymer matrix composites materials have high impact damage susceptibilities which are inevitable throughout the manufacturing process and service life of the airplane structures [1]. Low velocity impacts (LVI) on composites materials could cause such damage that it is nearly invisible from the outer surface but cause extensive inner damage. This results in damage that is difficult to be detect by eyes but with a significant reduction in strength, as much as 60% [2]. Therefore, the mechanical properties of the composite materials with barely visible impact damage (BVID) or impact damage of cut-off energy often are used as the material allowable values of the aircraft structures in accordance with the federal aviation administration regulation 25.571 [3] and the advisory circular 20-107B [4].

Extensive research has been conducted to understand on the effects of low-velocity impact damage on the residual strength of composite laminates, both from an experimental and a numerical focus. For example, Cantwell and Morton [5] provided a comprehensive review of the impact response of continuous fiber-reinforced composites a long time ago, and the effects of varying the properties of the fiber, matrix and interphase are examined, as well as the role of target geometry and loading rate on the dynamic response. Ali et al. [6] also presented a review of different strategies dealing with development of new composite materials investigated by several research groups that can be used to mitigate the low velocity impact damage in laminated composites. However, because of the importance and complexity associated with impact and following mechanical loading process, the compression-after-impact is still one of the hottest topics for composite materials. Lin and Wass [7] carried drop-tower tests and experimentally investigated the low velocity impact on a shear dominated composite laminate with stacking sequence [45/0/−45/90]_3s_. He et al. [8] investigated the effects of material parameters, and constitutive relationships on the dynamic mechanical response of composite laminates subjected to high- and low-velocity impacts. Additionally, they also studied the role of impactor shape on the damage area of the adhesive layer and displacement of the center of the laminated plates. Umair et al. [9] conducted low velocity drop-weight impact tests on polymeric composites and investigated the maximum force, displacement and impact energy absorption during the impact events. 

In the cases of numerical simulations, two independent numerical damage models were often used to simulate the failure of the composite materials [10]. The first simulation constitutes the progressive failure analysis (PFA) considering the damage initiation criteria using Hashin’s theory and damage evolution based on the energy criterion. The second simulation constitutes the cohesive zone model (CZM), which was used to simulate failure of composite material caused by the delamination phenomenon. The compressive failure process of the composite materials subjected to impact damage were simulated by scholars following the two above-mentioned methods. For example, Tan et al. [11] developed a high-fidelity three-dimensional composite damage model to predict both low-velocity impact damage and CAI strength of composite laminates. The intralaminar damage model accounts for physically based tensile and compressive failure mechanisms of the fibers and matrix, when subjected to a three-dimensional stress state. Cohesive behavior was employed to model the interlaminar failure between plies with a bi-linear traction–separation law. Abdulhamid et al. [12] investigated the impact and CAI of a composite laminate using a discrete ply model. Their study highlights the importance of modeling intra-ply matrix cracking for impact simulation. Sun and Hallett [13] tested two similar quasi-isotropic laminates and investigated the key driving mechanisms and damage evolution of the compressive failure of laminated composites containing BVID using compression after low-velocity impact and indentation tests. The effect of ply-blocking and influence of factors, such as impact energy, delamination area and surface indentation, on compressive failure was studied. Rozylo et al. [14] presented a simplified model of damage of composite plates (SDM) subjected to low-velocity impact testing, and the damage model assumes reduced thickness of laminate plies versus impact energy. Ouyang et al. [15] proposed an equivalent damage model to quickly predict the CAI strength of composite laminates. Progressive failure analysis and the virtual crack closure technique (VCCT) are used to simulate the intra-laminar and inter-laminar failure of impacted laminates under axial compression. 

These studies provide valuable evidence for the composite structure design of the upper wing panel and the fuselage belly, where compression is the primary loading. However, these achievements are not appropriate to the webs of the composite spars, ribs and the sides of the fuselage, where in-plane shear is the primary loading. In plane shear-after-impact (ShAI) properties of the composite laminates are essential to the design of these structures. 

There are many distinct standardized test methods to obtain the in-plane shear properties of the composite materials. Some of the most well-known test methods are Iosipescu shear (ASTM D 5379), ±45° tension shear (ASTM D 3518), two-rail and three-rail shear (ASTM D 4255), double-notched shear (ASTM D 3846), V-Notched Beam (ASTM D5379), torsion of a thin tube (ASTM D 5448), torsion of a solid rod (ASTM E 143), V-notched rail shear (ASTM 7078), picture frame shear (ASTM D 8067), and so on. These in-plane shear test methods are applied and discussed in many studies [16,17,18,19,20,21], in addition to these, the Arcan and modified Arcan test methods were also applied by some scholars to obtain the in-plane pure shear properties of the composite materials [22,23,24,25]. However, most of these above-mentioned methods are limited to the specimen geometry, layups and shear stress area. The pure shear stress region lies solely in a quite narrow area between the notches according to the Iosipescu shear and all the other notched shear methods, and this makes them inappropriate to test the shear-after-impact responses. The ±45° Tensile Shear test is characterized by off-axis tensioning of a rectangular cross ply lamina, typically at a ±45° orientation and thus, cannot be applied to the composite laminates with complex layup in the actual application. In addition, due to the fixed boundary conditions of the end clamps, a very complex stress state is imparted into the specimen once tensioned. The torsional shear of a thin-walled circular tube is considered the ideal shear test from the applied mechanics viewpoint, but it also presents many challenges such as high specimen manufacturing cost and difficulties of conducting the test. Furthermore, it cannot account for the property variations induced by the different manufacturing process between the flat specimens and tubes. 

The above-mentioned methods have been applied to test the in-plane shear properties of the composite materials, but only a few of them can provide a large area in shear stress, which is necessary to investigate the material’s ShAI properties. Among these, Souza et al. [26] executed three-rail shear tests on the fiber-reinforced laminated composite laminates with barely visible impact damage and studied the shear-after-impact performance. However, the two-rail or three-rail shear methods produced simple shear stress but not pure shear stress in the specimens and thus, significant normal stresses are present in the test section. Whereas, in the picture frame shear test, a square composite sheet is clamped with four rigid arms on all four sides and these arms are pivoted together at each corner, which can represent the real geometric features and loading conditions of the webs that are surrounded by chords and stiffeners on the four sides. The two diagonally opposite corners of the jig are then displaced using a mechanical testing machine and thus, a quite uniform shear stress state within most of the area of the specimen can be produced. The picture frame test method has been applied in some research to study the ShAI behaviors of the stiffened composite panels at a subcomponent level [27,28,29,30,31]. However, to the best of the authors’ knowledge, very few studies have been published on getting the ShAI properties of the composite materials at a coupon level using the picture frame test method. According to the building block approach [4], the material data at the coupon level are necessary to obtain the material allowable values and they are the basis of the airplane structural design. 

In this study, a set of methods combined with theoretical, experimental and simulation work is developed and proposed to obtain the ShAI properties of the composite laminates in pure shear stress at a couple level. The buckling load and material failure load of the balanced and symmetric laminate are compared, and on this basis, the layup and sizes of the tested specimen are defined. Subsequently, impact tests of the composite laminate specimens with different impact energies and picture frame tests of the impacted specimens are conducted to investigate the ShAI properties. Furthermore, a finite element simulation method has been developed to analyze the failure process.

## 2. Specimen Sizing

The buckling load of the in-plane shear specimens should be higher than the material failure load in order to avoid global buckling during the loading process. Additionally, the specimens should be large enough to eliminate the interactive effects between the impact damage and the specimens’ edges. The shear buckling load of an orthotropic laminate with symmetric and balanced layups can be expressed as Equation (1).
(1)$$N_{xycr}=K_{s}\frac{\pi^{2}\sqrt[{4}]{D_{11}D_{22}^{3}}}{b^{2}}$$
where the *D*_11_ and *D*_22_ are the relative values in the ABD matrix of the composite laminate, *b* is the length of the shorter side of the rectangular specimen, *K_s_* is the shear buckling factor and its value can be obtained in accordance with the reference [32].

Through calculation, the buckling and material failure load values of the orthotropic laminates with different number of plies (layups are [45/0/−45/90]_ns_, where *n* = [1, 2, 3]) and different specimen sizes can be obtained as shown in Figure 1. The mechanical parameters of the unidirectional lamina are listed in Table 1, and three different failure criteria, Tsai-Hill, Tsai-Wu [33], and maximum shear strain (MSE) are used to predict the failure load of the composite laminates. 

From Figure 1, it can be seen that the laminates with ply numbers of less than 16 are not appropriate to test the in-plane shear-after impact properties of the composite materials. As for the laminates with 24 plies, when the length of the specimen’s side is shorter than 171 mm, the buckling load will be higher than the material failure loads obtained from all the three criteria. With the ply number of the laminate continuing to increase, for example 32 plies, the requirement on the testing machine will be greatly increased due to the higher failure load and further, the manufacturing costs of the specimens and jigs will also rise greatly. Thus, with the combined consideration of the material and manufacturing costs, the stacking layups of the specimen was decided to be [45/0/−45/90]_3s_ and the area of testing section was decided to be 150 mm × 150 mm. Through calculation, the failure load based on maximum shear strain criterion and the buckling load of this specimen are 2250 N/mm and 2955 N/mm, respectively.

## 3. Experimental Tests

### 3.1. Impact Tests

To introduce impact damage, low-velocity drop-weight impact tests were performed with an Instron Ceast 9350 test machine following the test standard, ASTM D7136 [34]. The testing setup comprised the test machine, built-in software and the data acquisition system which together allowed the test data to be recorded automatically. A standard steel impactor with a hemispherical striker tip of 16 mm in diameter and 6.277 kg in weight was used. During the tests, the impactor was lifted to a height automatically according to the input parameters of impactor’s mass and desired impact energy, and then was released to drop down onto the surface of the specimens. A set of anti-rebound catcher devices were mounted in the tower house of the test machine and the impactor was caught after bouncing to prevent repeated impacts. 

The test specimens were fixed on the standard support fixture of the testing machine with a 125 mm × 75 mm notch in the middle and were impacted at their centers with four different energy levels, i.e., 20 J, 30 J, 40 J and 50 J, respectively. Three repeated tests were conducted with each impact energy level. After the introduction of impact damage, each impacted dent depth was measured immediately three times using a INSIZE IP54 digital dial depth gauge of 0.001 mm precision and the dent depth was taken as the average value of the three measurements. A non-destructive inspection instrument, GE Phasor XS Phased array ultrasonic detector, was used to characterize the internal impact damage. The delaminate area was calculated by means of a self-made procedure with the scanned pictures as the input.

### 3.2. Picture Frame Shear Tests

Shear tests of all the impacted specimens were performed with an MTS C64.106 servo-hydraulic universal testing machine equipped with a 1000 kN load cell following the picture frame test standard, ASTM D8067 [35]. The test setup is shown in Figure 2. The composite laminate specimens were clamped by the test jig and a diagonal tensile load was applied by the test machine in a quasi-static displacement-controlled mode at a constant loading rate of 1.0 mm/min.

The test jig consists of eight identical steel plates (two plates composed of a pair of arms) with 270 mm in length, 52 mm in width and 10 mm in thickness. There are two 26 mm-diameter holes at both sides of each plate and three 16 mm-diameter holes in the middle. The distance between the two 26 mm-diameter holes is 218 mm. The eight plates were pivoted together at the four corners through the 26 mm-diameter holes and the plates were assembled with the specimen through the three 16 mm-diameter holes at each side of the specimen. The environmental temperature was maintained at (23 ± 5) °C and the relative humidity was at (55 ± 5)% during tests. The axial load and deformation were measured using the force and displacement transducers embedded in the testing machine. The sampling rates were all set to 10 Hz.

## 4. Test Results

### 4.1. Impact Test Results

Typical ultrasonic scan results of the impacted specimens with different impact energy levels are shown in Figure 3. Each figure represents an area of 120 mm × 80 mm. From Figure 3, it can be seen that the delamination area is larger with the impact energies, which is reasonable. 

The relationships between the impact damage and the impact energy are also shown in Figure 4. The mean values, standard deviations and coefficient of variation are obtained through the Equations from (2) to (4).
(2)$$\overset{¯}{x}=\left(\overset{n}{\underset{i=1}{\sum }}x_{i}\right)/n$$
(3)$$S_{n-1}=\sqrt{\left(\sum_{i=1}^{n}x_{{}_{i}}^{2}-n{\bar{x}}^{2}\right)/\left(n-1\right)}$$
(4)$$CV=100\times S_{n-1}/\overset{¯}{x}$$
where:
$\bar{x}$ = sample mean (average);*s_n−_*_1_ = Sample standard deviation;*CV* = Sample coefficient of variation;*n* = number of specimens;*x_i_* = measured or derived property.
From Figure 4 it can be seen that both the dent depth and the delamination area increase with the impact energies. When the impact energies are increased from 20 J to 50 J, the dent depth is increased by about 2.2 times, whereas that of the delamination area is about 12.1 times. It can also be seen that the dispersion of impact damage generally decreases with the impact energy, and the dispersion of the dent depth is much lower than that of the delamination area. The dent depth is lower than 1 mm as the impact energy is 50 J, which is treated as the energy cut-off value. This indicates that the energy cut-off criterion should be adopted to obtain the material allowable values of this composite structure.

### 4.2. In-Plane Shear-after-Impact Test Results

The load-displacement curves of the composite laminate specimens with different impact damages under picture frame tests are shown in Figure 5, from which it can be seen that stiffness does not change much by the impact damage.

According to the mechanical analysis of the picture frame loading [36], the specimen shear strain *γ* can be calculated with the transversal displacement *δ_a_* and the picture frame edge length *L_a_*, as in Equation (5).
(5)$$\gamma =\frac{\pi }{2}-2\cos^{-1}\left(\frac{\sqrt{2}}{2}+\frac{\delta_{a}}{2L_{a}}\right)$$

Figure 5 shows that the displacements of the loading head of the testing machine at specimens’ failure are between 12 mm and 13 mm. The maximum deformation of the composite specimen along the diagonal direction, *δ_a_*, will be obtained as about 2.4 mm if the maximum shear strain *γ* being 0.022 is substituted into the Equation (3). This indicates that only a small portion of the displacement comes from the shear deformation of the specimens, and most of the displacement measured by the testing machine was contributed by the testing system, such as the elongation of the steel bar shown in Figure 2.

The shear force can be calculated directly for standard hinged picture frames depending on the shear angle as shown below:(6)$$F_{sh}=\frac{F}{2\times \cos\frac{a}{2}}$$
(7)$$\alpha =90^{\circ }-\frac{180\gamma }{\pi }$$
where the *F* is the applied load by the material testing machine.

Since the stiffness of the specimens is high, the tensile displacement is low and the angles between the arms did not change much till the failure of the specimens. Therefore,
(8)$$F_{sh}\approx \frac{F}{2\times \cos45^{\circ }}\approx 0.707F$$
and the shear load per unit length can be calculated using Equation (9).
(9)$$N_{xy}=\frac{F_{sh}}{L_{a}}=\frac{0.707\;F}{L_{a}}$$

The relationship between failure load per unit length of the specimen with the impact energies is shown in Figure 6, from which it can be seen the residual ShAI strength of the composite laminate is decreased with the impact energy, and that it is also reasonable. However, the residual ShAI strength of the laminate with 50 J impact damage is about 7.8% lower than that with 20 J of impact damage. The decreasing level of strength with impact energy is much less than the increasing level of the dent depth and delamination area. Another difference is that the dispersion of the residual ShAI strength is increased with the impact energies and the dispersion of the residual ShAI strength is lower than those of the impact damages. 

From the comparison between Figure 1 and Figure 6, it can also be concluded that it is conservative to estimate the in-plane shear strength of the composite laminate using both the Tsai-Hill and Tsai-Wu failure criteria since the residual tested ShAI strength is even higher than the estimated failure load of unimpacted laminates. Compared with the in-plane shear strength of the undamaged composite laminate estimated from the maximum shearing strain criterion, the residual ShAI strength of the composite laminate is decreased to about 79.4%, 76.5%, 75.6% and 73.2% by the four different impact damages with 20 J, 30 J, 40 J and 50 J impact energies, respectively. This indicates the impact damage with the cut-off energy, 50 J, causes a 26.8% reduction in the residual strength.

The ShAI failure modes of the specimens are shown in Figure 7, from which it can be seen that the surface cracks of the specimens with 20 J impact damage are all along the loading direction. Whereas, with the increase of the impact energies, there are some cracks perpendicular with the loading direction. This indicates that the primary final failure mode was transmitted from fiber-compression failure to fiber-tension failure with the impact damage being more serious. 

## 5. Finite Element Modelling and Simulations

Two finite element models of the composite laminate were established using the commercial finite element code, Abaqus [37]. The first model is the buckling analysis model of the composite laminate and the other model is used to analyze the composite laminate subjected to impact loading and shear-after-impact loading. A linear perturbation-Buckling algorithm was used to analyze the buckling load and buckling mode of the composite specimen under shear loading. A two-step explicit algorithm was used to simulate both the impact and the shear-after-impact loading processes. The time of the impact step and shear-after-step are 0.005 s and 0.2 s, respectively. The loading rate of 10 mm per second can eliminate the dynamic loading effect since the simulated responses of the undamaged laminate obtained by the explicit algorithm and static algorithm are similar.

### 5.1. Meshes and Boundary Conditions

The meshes of finite element model of the composite laminate plate under impact and shear-after-impact are shown in Figure 8. The model is composed of one laminate plate, one tup, one supporting plate at the bottom of the laminate plate and eight plates at the four sides of the laminate plate, and all their dimensions are consistent with the test specimens and jigs. Eight-node in-plane reduced integral continuum shell elements with enhanced hourglass control and reduced integration, SC8R, were used to model the plies of the composite laminate plate to avoid the shear locking problem and to increase the computing efficiency. The modelling method of one element per ply lamina of the composite laminate provides a reasonable approximation of the through-thickness stresses of the composite laminates. The 0° axis of the composite laminate is along the horizontal direction and the ply angles are in a counterclockwise arrangement, which means the 45° direction is the direction from the bottom left to the upper right. There is a layer of zero mm thick eight-node three-dimensional cohesive elements, COH3D8, between two adjacent layers of SC8R elements, and these COH3D8 elements share nodes with the SC8R elements on both sides to simulate the delamination failure. The mesh density in the center of the composite laminate is finer than other areas in order to simulate the impact damage and the shear failure correctly, and in the meantime, save the simulation cost. 

As shown in Figure 9, the composite laminate plate was partitioned by five planes in horizontal and vertical directions, respectively, and the center of the composite laminate was partitioned by two inclined planes that are ±45˚ with the horizontal plane and by four circles with diameters of 4 mm, 8 mm, 16 mm, and 24 mm, respectively. The numbers inside the symbols “()” are the serial numbers of the edges which have special seeding method as listed in Table 2, and the global mesh density of the laminate plate is 5.3 mm.

There are 100,928 SC8R and 55,915 COH3D8 in the meshes of the composite plate in total, and the mesh size in the center of the specimen is about 0.3 mm. The supporting jig was simulated as a discrete rigid plate with the four side length values being 150 mm with a 125 × 75 mm notch in the center, which is the same as the jig’s dimension defined in the test standards, ASTM D 7136. The tup was simulated as an analytical rigid surface with a diameter of 16 mm and a mass of 6.277 kg. Two hundred and ten linear quadrilateral elements of type S4R were used to simulate each steel plates in the picture frame test jig and the thickness of the S4R element were set to be 10 mm. The pivot relationships between the steel plates were defined using the MPC Pin function in the Abaqus.

In the impact loading step of the impact and ShAI loading model, the composite laminate was fixed all around and a predefined velocity perpendicular to the impacting surface of the composite laminate plate was set to the tup. A surface-to-surface contact relationship was used to define the contact relationships between the rigid tup and the composite laminate. “Hard” contact was used to define the normal interaction properties of the contact to avoid penetration during the analysis process. Penalty friction formulation were used to define the tangential interaction properties of the contact. The friction coefficient between the composite skin and the steel impactor was set to be 0.1. In the following shear loading step, the tup was fixed, and the bottom left corner of the test jig was fixed, whereas the upper right corner was applied with a displacement of 3 mm along the diagonal direction.

In the buckling analysis model, the tup was suppressed, and the bottom left corner of the test jig was fixed whereas the upper right corner was applied with a unit load along the diagonal direction.

### 5.2. Constitutive Models and Material Parameters

(1) Intralaminar failure Criterion

Progressive damage modeling was used to model the damage propagation process of the composite laminate and degradation of the strength under impact loading and shear loading. The Hashin failure criteria [38] were used to evaluate the damage initiation of the composite material, as shown in Table 3. The stress of the lamina model was calculated by classical composite laminate plate theory. In this model, the parameter α in Table 3 is set to be the default value of 0, and the point at which damage initiates is determined by measuring when each of several damage criteria is met. Each damage criteria compares the different measured stresses against the strength properties of the respective material. Losses in strength and stiffness of composite laminates were considered as failures. The failure modes of fiber damage fracture, matrix damage fracture and delamination failure could be drawn from the simulation. 

An energy-based method was employed to simulate the damage evolution after damage was initiated [37]. Strain softening, which means the load carrying ability decreases as the strain increases, was used to model how the material will respond after damage has already occurred. The response is computed using $\sigma =C_{d}\varepsilon$, where ε is the strain and $C_{d}$ is the damaged elasticity matrix, as shown in Equation (10).
(10)$$C_{d}=\frac{1}{D}\left[\begin{array}{ccc} \left(1-d_{f}\right)E_{1} & \left(1-d_{f}\right)\left(1-d_{m}\right)v_{21}E_{1} & 0\\ \left(1-d_{f}\right)\left(1-d_{m}\right)v_{12}E_{2} & \left(1-d_{m}\right)E_{2} & 0\\ 0 & 0 & \left(1-d_{s}\right)DG \end{array}\right]$$

In this equation, *D* is a combination of variables as shown in Equation (11), while *d_f_*, *d_m_*, and *d_s_* are damage state variables for the fiber, matrix, and shear, respectively.
(11)$$D=1-\left(1-d_{f}\right)\left(1-d_{m}\right)\nu_{12}\nu_{21}$$

When damage has been initiated in a particular mode, the equivalent stress-equivalent displacement relationship controls how the material properties will degrade. This damage variable varies from 0, as damage initiates, to 1, as the deformation energy equals to the fracture energies of the composite material in fiber direction and transverse direction.

(2) Interlaminar damage

The cohesive zone model approach was used to predict onset and propagation of interlaminar cracks, such as delamination between the adjacent plies of composite and debonding between the stiffener and skin. The cohesive zone model approach combines both, a strength-based and a fracture mechanics-based criteria for the prediction of interlaminar failure. The interface started to degrade once the interface strength reached the mechanical properties of the interface. The degradation of the interface properties involves energy dissipation. Once the energy dissipated by surface area is equal to the fracture toughness of the interface, a new free area has been created. 

The cohesive zone model approach assumes that all the dissipation mechanisms occurring at the fracture process zone can be lumped to an interface or surface coincident with the crack plane. The mechanical behavior of this interface is representative of all the nonlinear mechanisms occurring within the fracture process zone and it is usually formulated in terms of a traction–separation law, also called cohesive law.

Ye’s criterion [39] was used as strength degradation initiation criterion to predict the initiation of the degradation process, which is written as Equation (12).
(12)$${\left(\frac{\langle \tau_{33}\rangle }{\tau_{33}^{\circ }}\right)}^{2}+\frac{{\left(\tau_{23}\right)}^{2}+{\left(\tau_{31}\right)}^{2}}{{\left(\tau_{23}^{\circ }\right)}^{2}+{\left(\tau_{31}^{\circ }\right)}^{2}}=1$$
where $\tau_{33}$, $\tau_{23}$, $\tau_{31}\;$ are the normal traction and out of plane shear traction, and $\tau_{33}^{\circ }$, $\tau_{23}^{\circ }$, $\tau_{31}^{\circ }$ are the normal and shear interface strengths. The 〈⬚〉 is the Macaulay symbol, it is written as Equation (13). It represents that the compressive through-thickness stress has no influence on the failure of the interface, and the interface is considered to be in a pure mode II crack state.
(13)$$\langle \tau \rangle =\left\{\begin{array}{c} \tau ,\;\tau >0\;\\ 0,\tau \leq 0 \end{array}\right.$$

The interface of adjacent plies would be fully degraded when the energy dissipated equals the fracture toughness. The B-K criterion [40] was implemented to predict the damage propagation under mixed-mode loading conditions, which is shown as Equation (14). The value of η, which is the power exponent, is set to be 1.45.
(14)$$G_{C}=G_{IC}+\left(G_{\mathrm{II}C}-G_{IC}\right){\left(\frac{G_{\mathrm{II}}}{G_{I}+G_{\mathrm{II}}}\right)}^{\eta }$$
where G_IC_, G_IIC_ and G_IIIC_ refer to the critical fracture energies required to cause failure in the normal, the first and the second shear direction, respectively. 

In ABAQUS, the damage factor *D* (SDEG: Scalar stiffness degradation, 0 ≤ *D* ≤ 1) is introduced to characterize the degree of damage for the cohesive element [39]. The stiffness coefficient (*K*) in damage evolution is expressed by Equation (15).
(15)$$K=\left(1-D\right)K_{0}$$
where *K*_0_ is the stiffness of complete material. When *D* equals to 0, it means that the material does not yield or has just begun to yield; when *D* equals to 1, it means the material has been damaged and the load carrying capacity is lost. The material parameters of the cohesive elements were listed in Table 4.

## 6. Simulation Results

### 6.1. Buckling Analysis Results

The first two buckling modes of the composite laminate under shear loading are shown in Figure 10. The eigenvalues are 6.60624 × 10^5^ and 7.87751 × 10^5^, respectively. This indicates the calculated initial buckling load is about 660.624 kN, and it is 3113.7 N/mm which is also 5.4% higher than the theoretical result, 2955 N/mm.

### 6.2. In-Plane Shear-after-Impact Analysis Results

The simulated deformation status is shown in Figure 11, where the U1 axis is along the diagonal direction (from the bottom left corner to the upper right corner) of the specimen. The simulated load-displacement curves of the ShAI are shown in Figure 12, from which it can be seen that the stiffness values of the composite laminate are generally linear before failure and do not change much with the different impact damage.

Using Equations (5) and (9), the effective failure strain and failure load values can be obtained as shown in Table 4. It shows that the simulated load values agree well with the test results, and the residual effective shear failure strain of the composite laminate with the 50 J impact damage is about 0.0132.

Table 5 and Table 6 show the delamination failure results of the composite laminate impacted with 20 J and 50 J after the impact loading and after the in-plane shearing loading, and Table 7 and Table 8 show the matrix tension failure results of composite laminate impacted with 20 J and 50 J after the impact loading and after the in-plane shearing loading. There is not much fiber failure after the shearing loading. In both the two tables, the damage factor D of the elements in red equal to 1, the elements in blue equal to 0, and the elements in green are between 1 and 0. The twenty-fourth layer is on the impacted surface and the first layer is on the backside surface. The first layer of cohesive elements is between the first and second layer of continuum elements that represent the lamina. 

The main impact damage is delamination, and the area of damage is increased with the impact energies. The changes of delamination area with the shear loading are little. The matrix failure statuses after impact loading with different energies are all strength failures in a local area, and the matrix failure statuses turn into energy failures after the shear loading. This indicates the damage area in the center of the specimen loose load capacities completely as the in-plane shear failure occurs. However, the matrix failure does not propagate to the area that is undamaged by the impact loading. 

## 7. Conclusions

(1)A set of methods combined with theoretical, experimental and simulation work is developed and proposed, and it is demonstrated to be reliable to study the ShAI responses of the composite laminates in pure shear stress at a couple level.(2)A conservative result is obtained on the shear strength of the composite laminates using the Tsai-Hill and Tsai-Wu failure criteria, and the maximum shear strain criteria can be used to produce a reasonable prediction.(3)The dent depth is lower than 1 mm as the impact energy being the cut-off energy, 50 J, which indicates the ShAI value with impact cut-off energy should be used to obtain the material allowable values of the composite laminate.(4)The impact damage with the cut-off energy, 50 J, causes a 26.8% reduction in the residual strength and the equivalent residual shear failure strain is about 0.0132.(5)A two-step progressive failure analysis model was developed to simulate the impact and ShAI responses of the composite laminates, and it can be concluded that primary reason of the shear failure is the propagation of both the matrix tensile failure and interlaminar delamination, but not fiber break failure.

## Figures and Tables

**Figure 1 materials-15-05029-f001:**
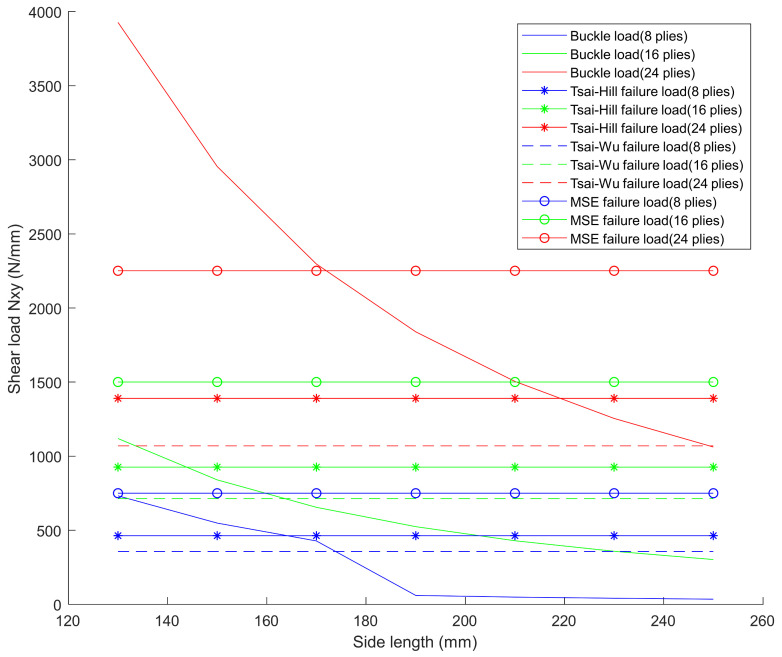
Buckle load and failure load results from the theoretical calculations.

**Figure 2 materials-15-05029-f002:**
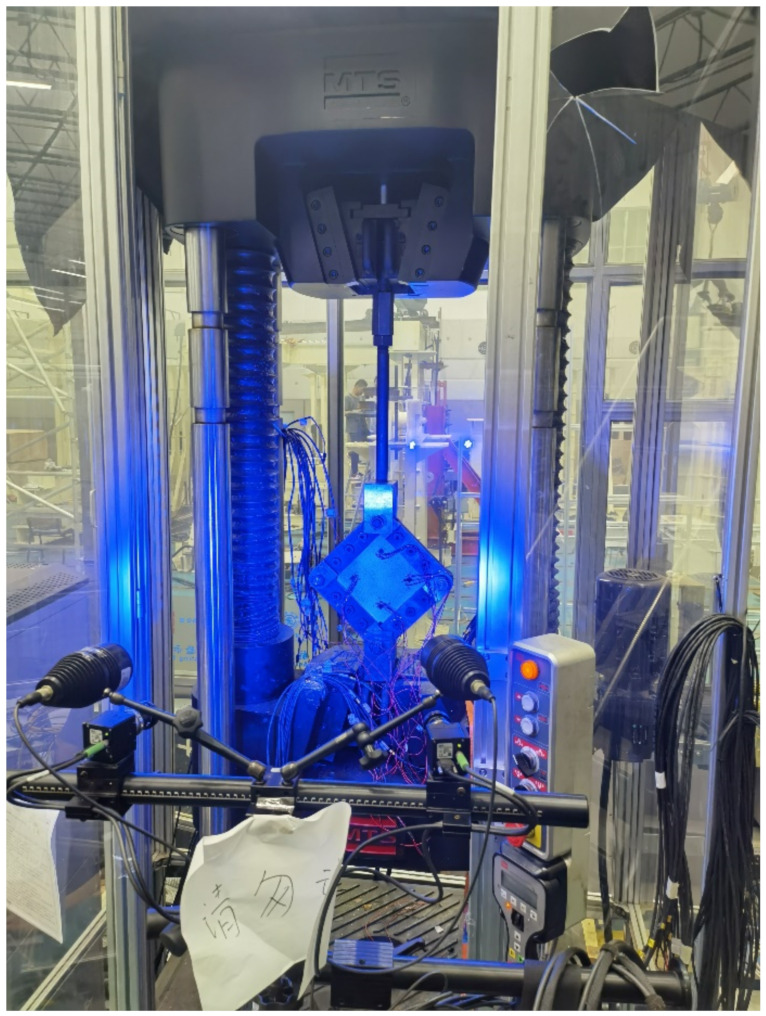
In-plane shear test setup (The Chinese words mean “Please do not move it”).

**Figure 3 materials-15-05029-f003:**
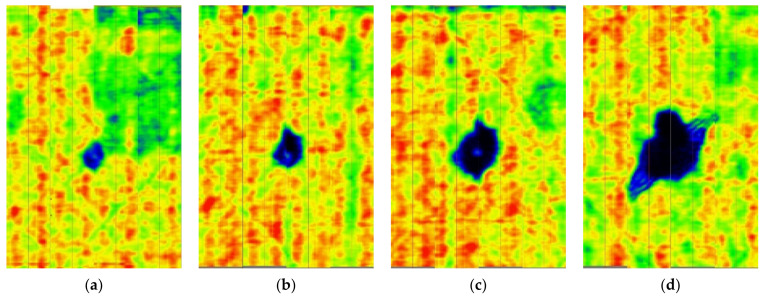
Ultrasonic scan results of the typical specimens with different impact energies: (**a**) 20 J; (**b**) 30 J; (**c**) 40 J; (**d**) 50 J.

**Figure 4 materials-15-05029-f004:**
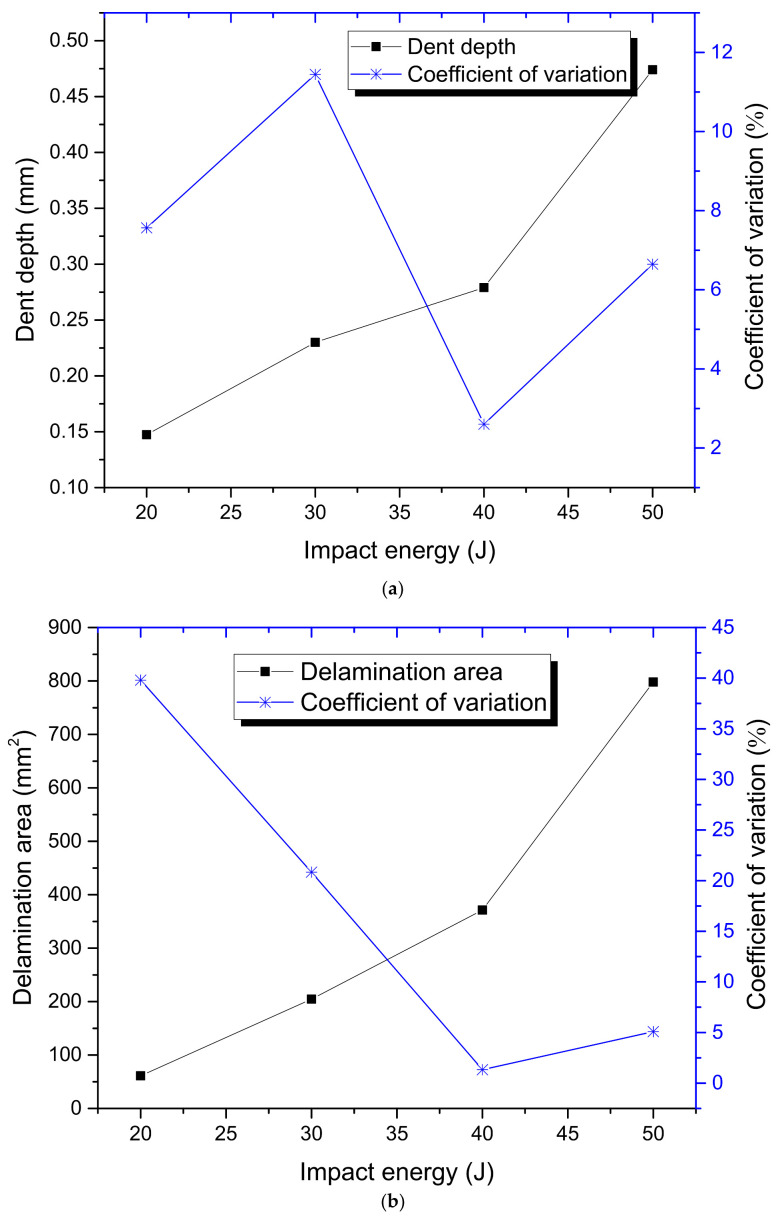
Relationships between impact damage conditions and impact energies: (**a**) Dent depth vs. impact energy curves; (**b**) delamination area vs. impact energy curves.

**Figure 5 materials-15-05029-f005:**
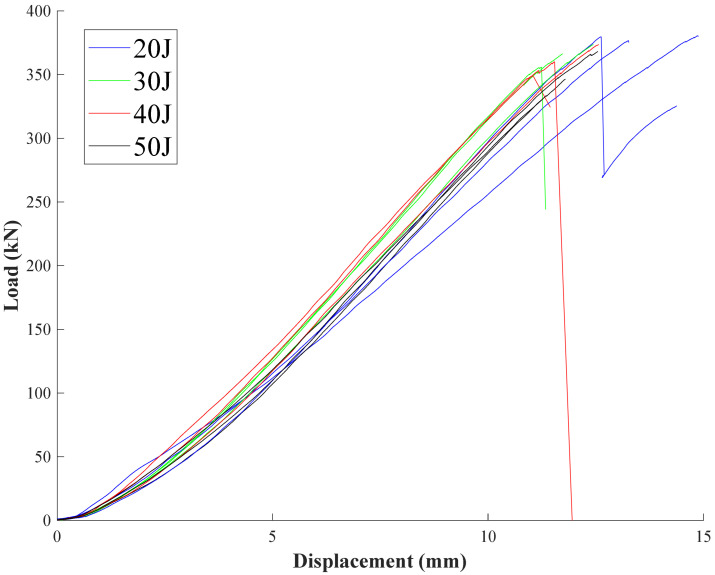
Load-displacement curves of the composite laminates with different impact damages.

**Figure 6 materials-15-05029-f006:**
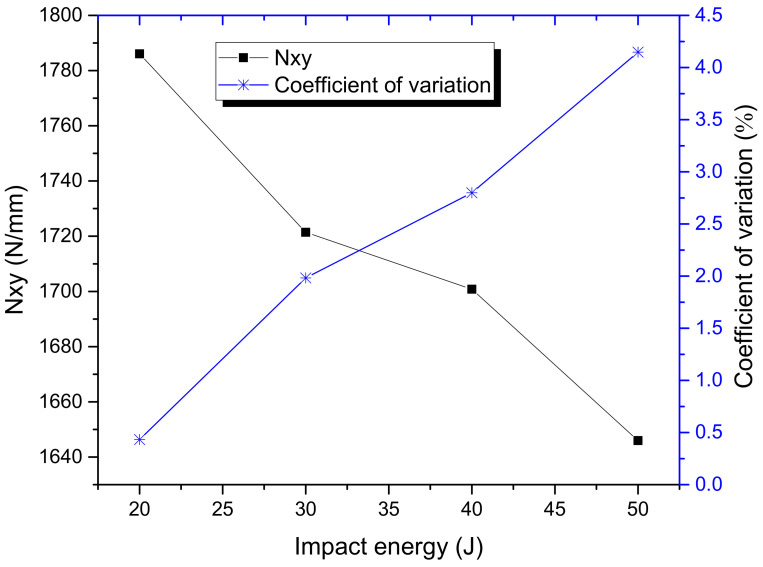
Relationships between residual shearing strength and impact energies.

**Figure 7 materials-15-05029-f007:**
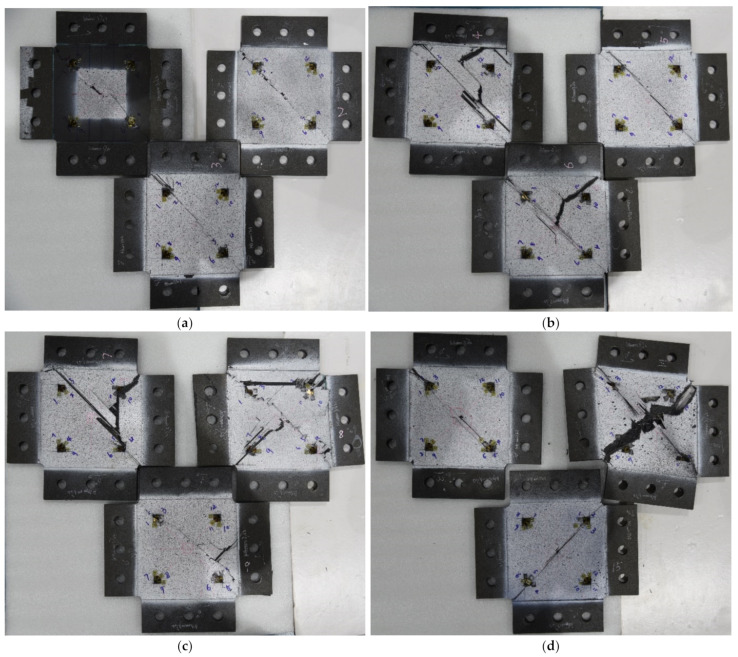
In-plane shear failure modes of the composite specimens: (**a**) 20 J; (**b**) 30 J; (**c**) 40 J; (**d**) 50 J.

**Figure 8 materials-15-05029-f008:**
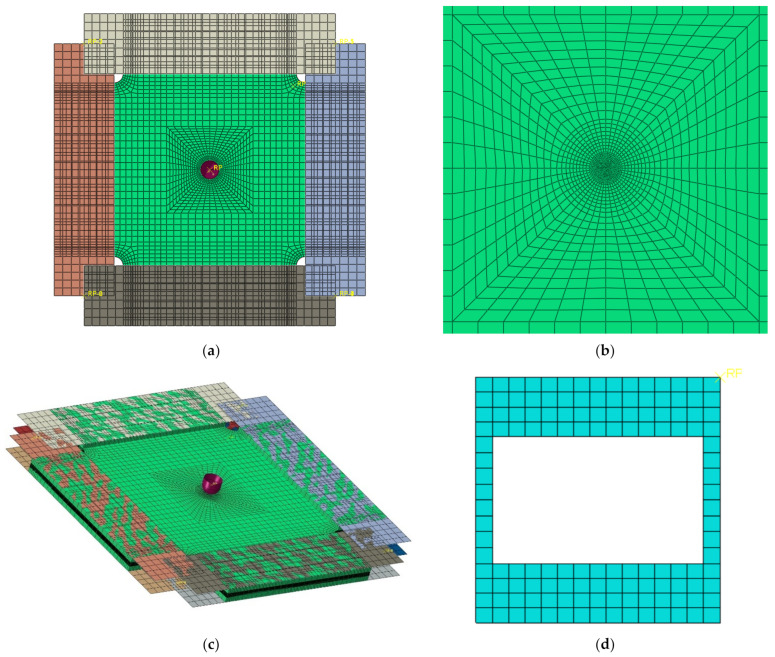
Meshes of the composite laminate under shear and ShAI loading: (**a**) From view of the model with jig; (**b**) detail view of the center of the specimen; (**c**) ISO view the model; (**d**) meshes of the support plate; (**e**) side view in detail.

**Figure 9 materials-15-05029-f009:**
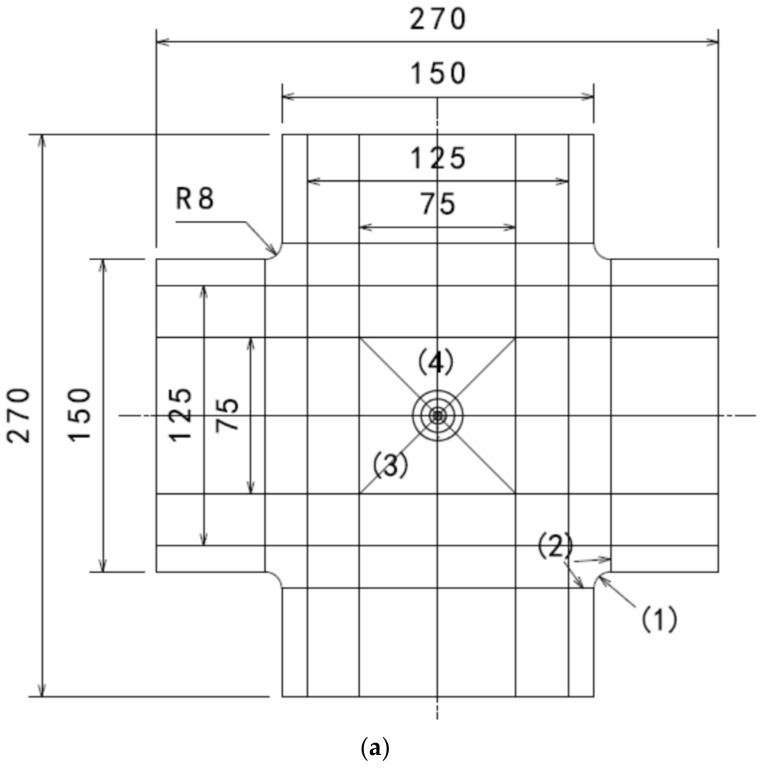
Seeding method of some edges in the composite laminate model: (**a**) Whole view; (**b**) detail view.

**Figure 10 materials-15-05029-f010:**
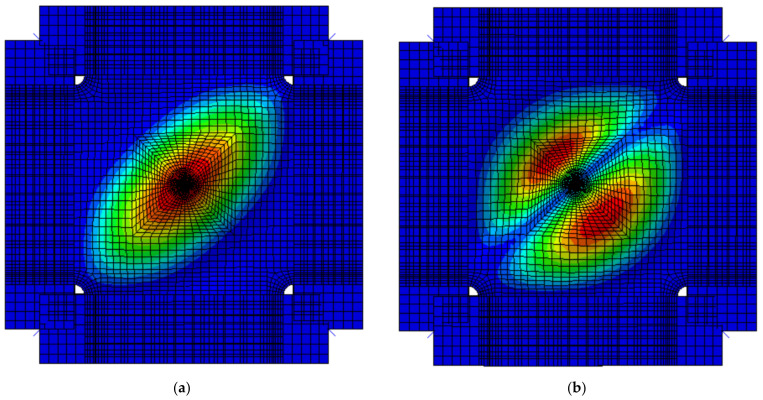
Buckling modes of the composite laminate under shearing load: (**a**) The first buckling mode; (**b**) the second buckling mode.

**Figure 11 materials-15-05029-f011:**
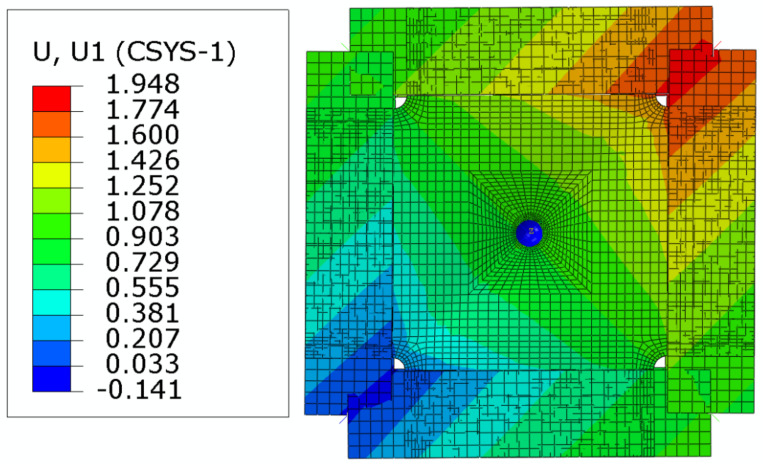
In-plane shear deformation of the composite laminate.

**Figure 12 materials-15-05029-f012:**
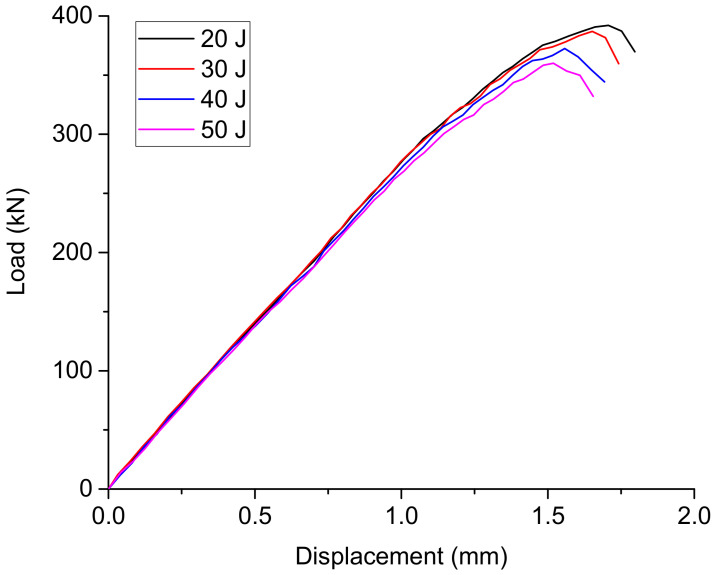
Simulated load-displacement curves of the composite laminates with different impact damages.

**Table 1 materials-15-05029-t001:** Mechanical properties of the unidirectional lamina.

Parameters	Values
Longitudinal modulus E11 (GPa)	163
Transverse modulus E22 (GPa)	9.9
Shear modulus G_12_ (GPa)	4.2
Transverse shear modulus G_23_ (GPa)	3.2
Poisson’s ratio ν_12_	0.31
Longitudinal tensile strength Xt (MPa)	3100
Longitudinal compressive strength Xc (MPa)	1487
Transverse tensile strength Yt (MPa)	69
Transverse compressive strength Yc (MPa)	277
Shear strength S_12_ (MPa)	92
Transverse shear strength S_12_ (MPa)	102

**Table 2 materials-15-05029-t002:** The seeding at special locations the composite laminate plate.

Edge No.	Location Description	Seeding Method
1	The four fillets at the four corners of the specimens	8 seeds without bias
2	The eight lines that connected with the four fillets	6 seeds without bias
3	The four inclined lines in the center of the specimen	15 seeds with bias ratio of 1.5 towards the center
4	The two horizontal and vertical lines in the center of the specimen	12 seeds with bias ratio of 2 towards the center
5	The eight lines in radial direction between the two outermost circles	4 seeds with bias ratio of 1.5 towards the center
6	The eight lines in radial direction between the two circles with diameters of 8 mm and 16 mm	5 seeds with bias ratio of 1.2 towards the center
7	The eight lines in radial direction between the two circles with diameters of 4 mm and 8 mm	5 seeds without bias
8	The eight lines in radial direction inside the innermost circle	5 seeds without bias
9	The circumferences of the four circles	48 seeds without bias

**Table 3 materials-15-05029-t003:** Hashin’s failure criteria [35].

$\mathrm{Fiber}\;\mathrm{tension}\;(\sigma_{11}\geq 0$)	${\left(\frac{\sigma_{11}}{X_{T}}\right)}^{2}+\alpha {\left(\frac{\sigma_{12}}{S_{12}}\right)}^{2}\geq 1$
$\mathrm{Fiber}\;\mathrm{compression}\;(\sigma_{11}<0$)	${\left(\frac{\sigma_{11}}{X_{C}}\right)}^{2}\geq 1$
$\mathrm{Matrix}\;\mathrm{tension}\;(\sigma_{22}\geq 0$)	${\left(\frac{\sigma_{22}}{Y_{T}}\right)}^{2}+{\left(\frac{\sigma_{12}}{S_{12}}\right)}^{2}\geq 1$
$\mathrm{Matrix}\;\mathrm{compression}\;(\sigma_{22}<0$)	${\left(\frac{\sigma_{22}}{2Y_{13}}\right)}^{2}+\left[{\left(\frac{Y_{C}}{2S_{13}}\right)}^{2}-1\right]\frac{\sigma_{22}}{Y_{C}}+{\left(\frac{\sigma_{12}}{S_{12}}\right)}^{2}\geq 1$

where $\sigma_{ij}$ (*i*, *j* = 1, 2, 3) are the stress components, and the 1-coordinate direction is aligned with the fibers, the 2-coordinate direction is transverse to the fibers, while the 3-coordinate direction is the thickness direction of the lamina; the α is a coefficient that determines the contribution of the shear stress to the fiber tensile initiation criterion.

**Table 4 materials-15-05029-t004:** Comparisons of the simulated and test results.

Impact energies of the laminate	20 J	30 J	40 J	50 J
Residual effective shear failure strain γ (/)	0.0153	0.0146	0.0137	0.0132
Simulated failure load Nxy (N/mm)	1848.2	1824.2	1755.6	1697.1
Tested failure load Nxy (N/mm)	1786.1	1721.4	1700.8	1645.9
Error (%)	3.48	5.97	3.22	3.11

**Table 5 materials-15-05029-t005:** Cohesive failure status of the composite laminate with an impact energy of 20 J.

Layer No.	Impacted Results	Shear Results	Layer No.	Impacted Results	Shear Results
1	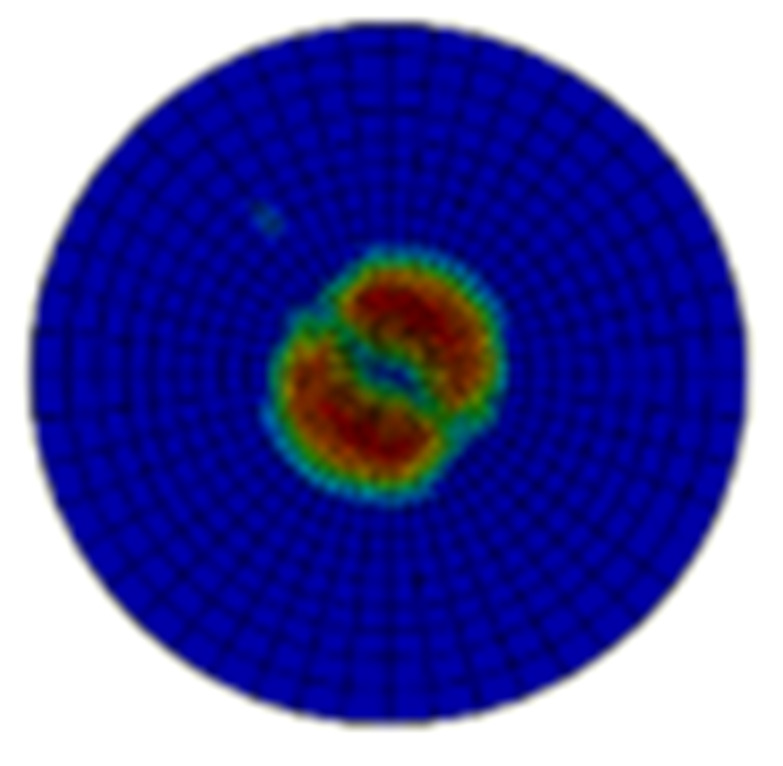	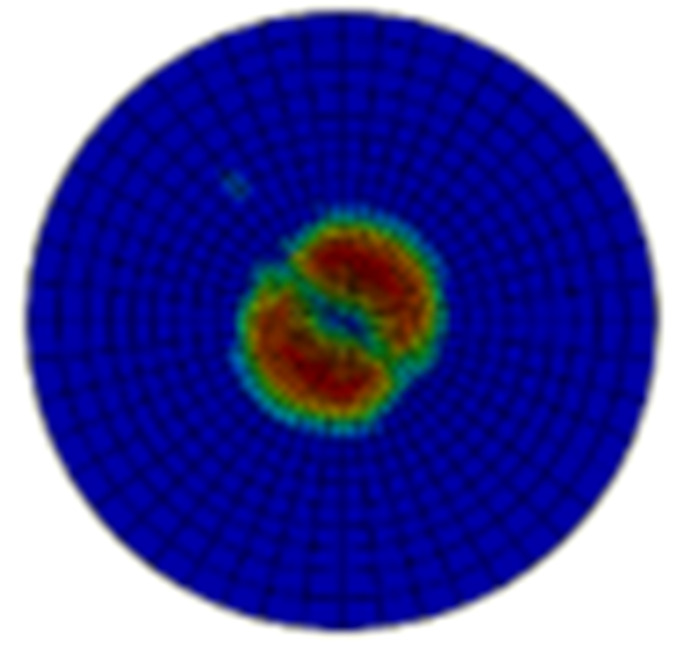	13	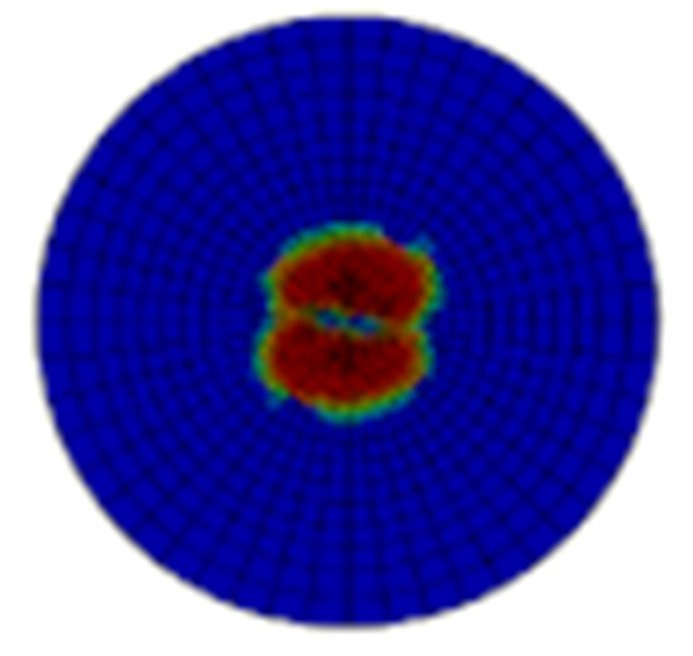	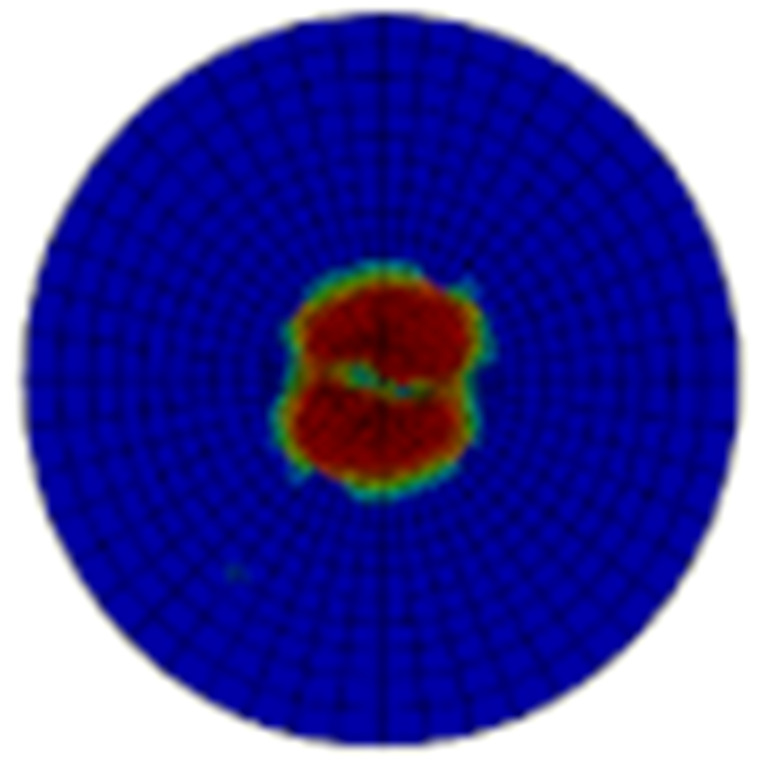
2	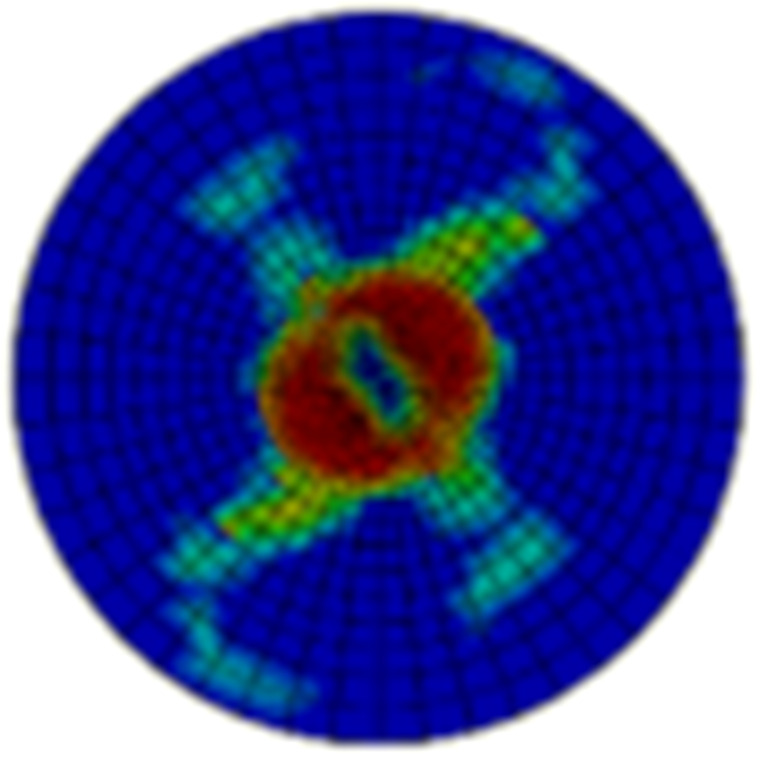	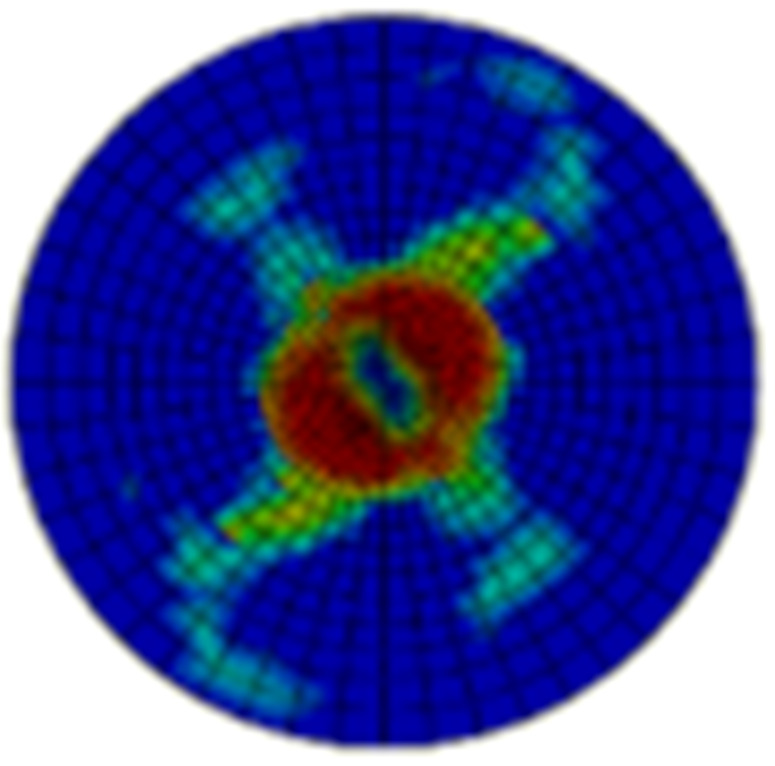	14	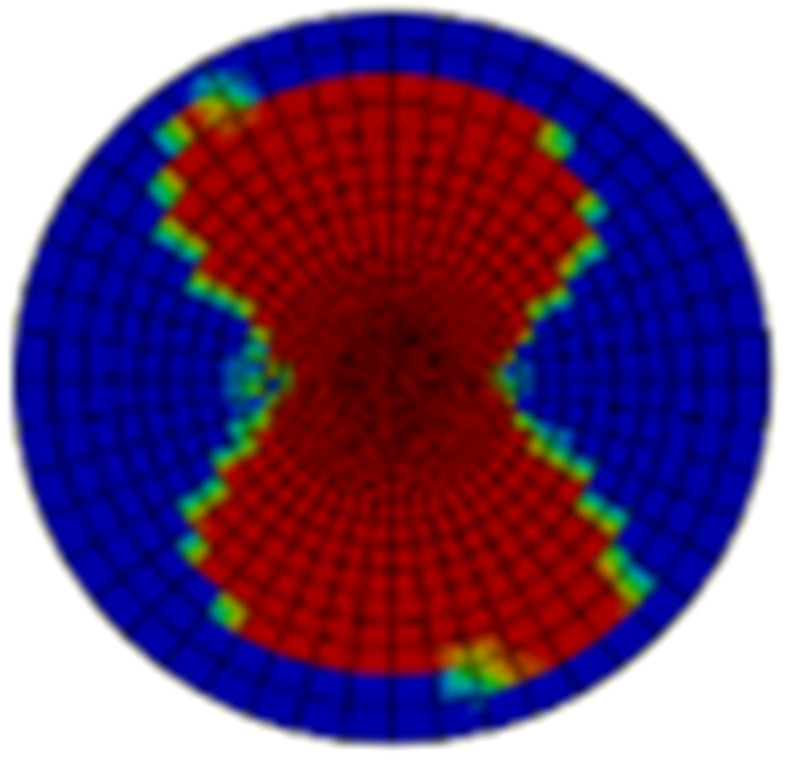	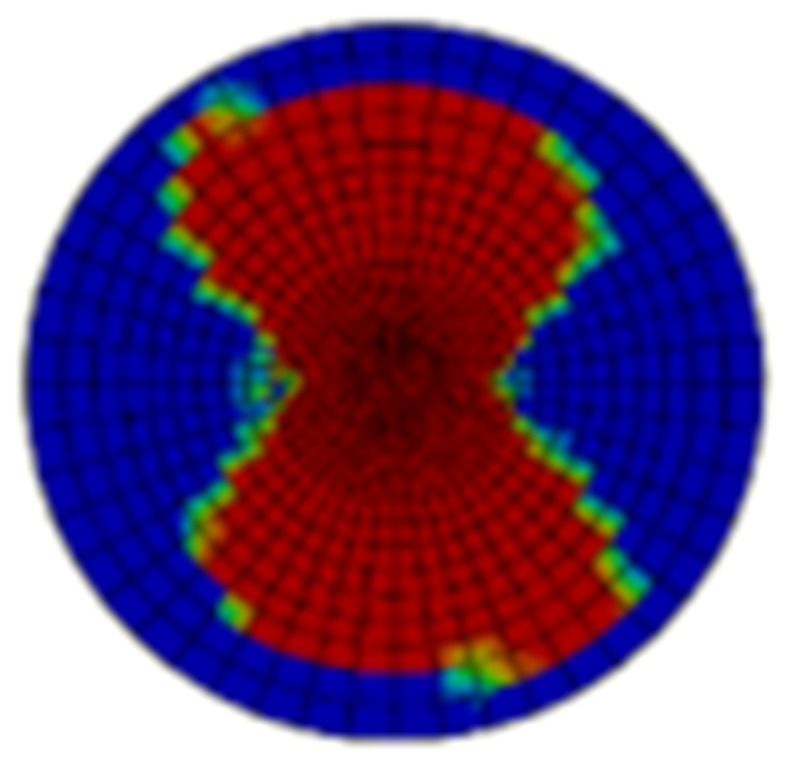
3	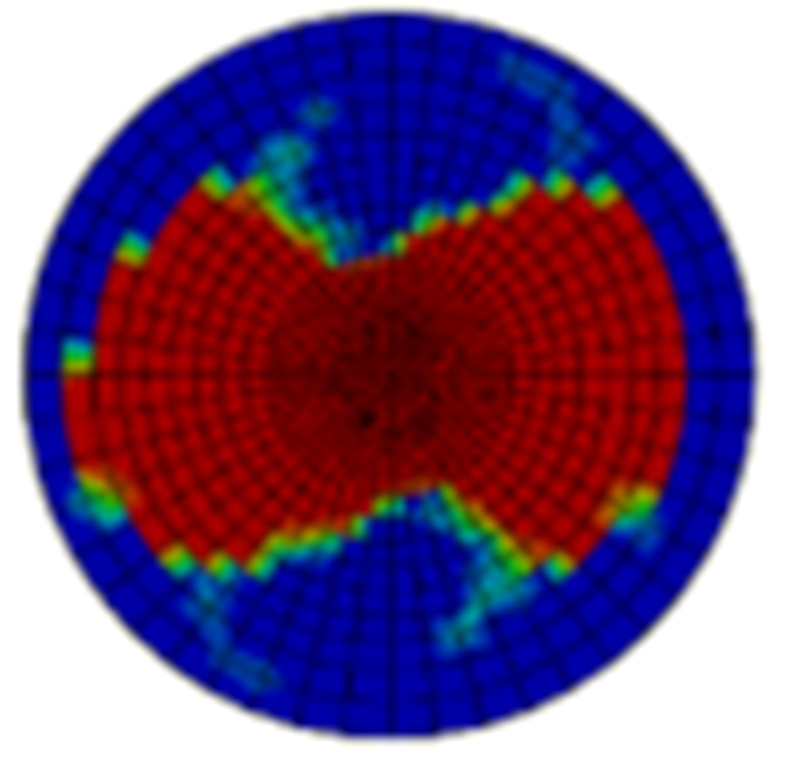	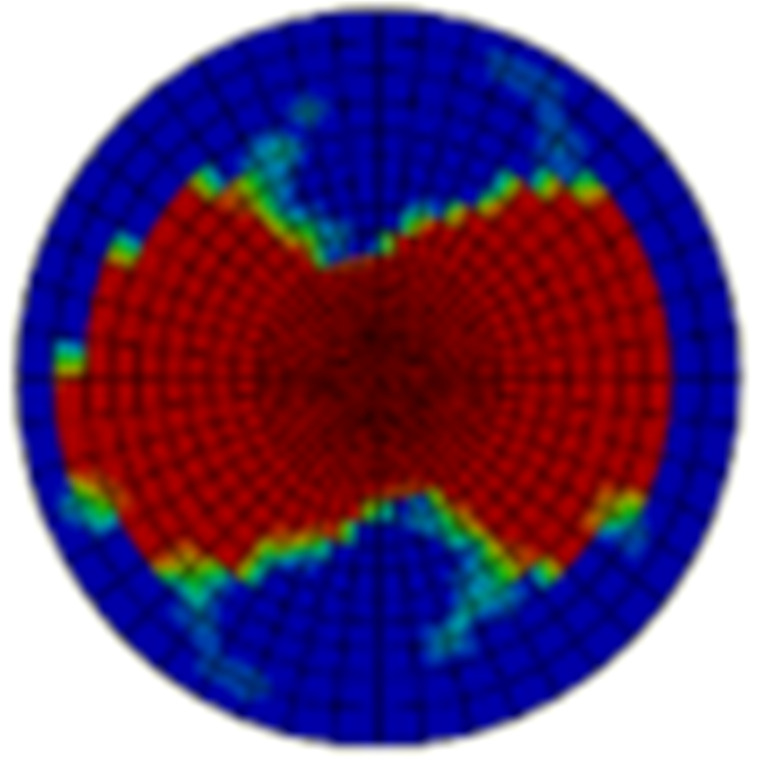	15	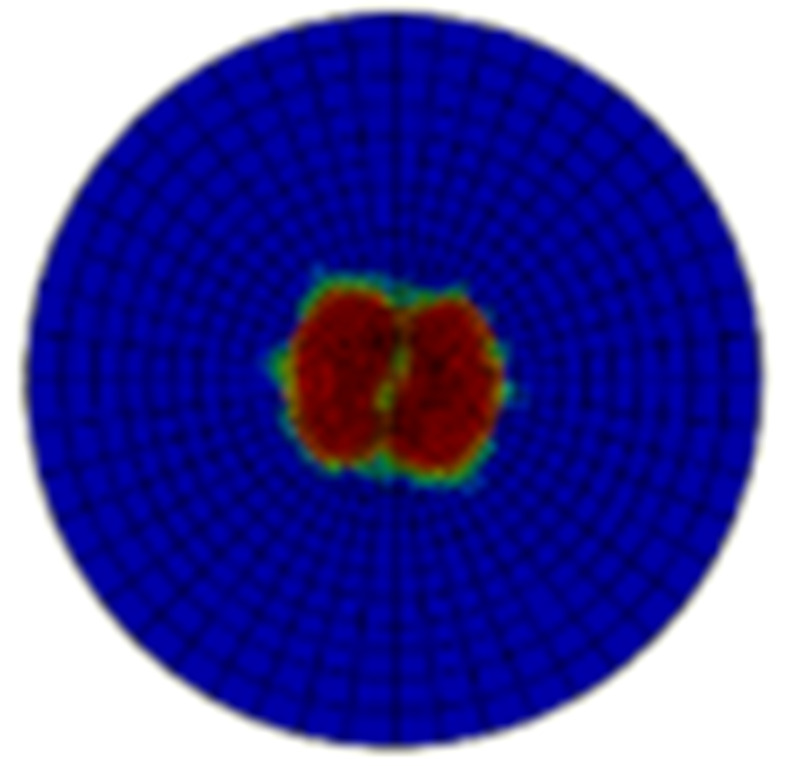	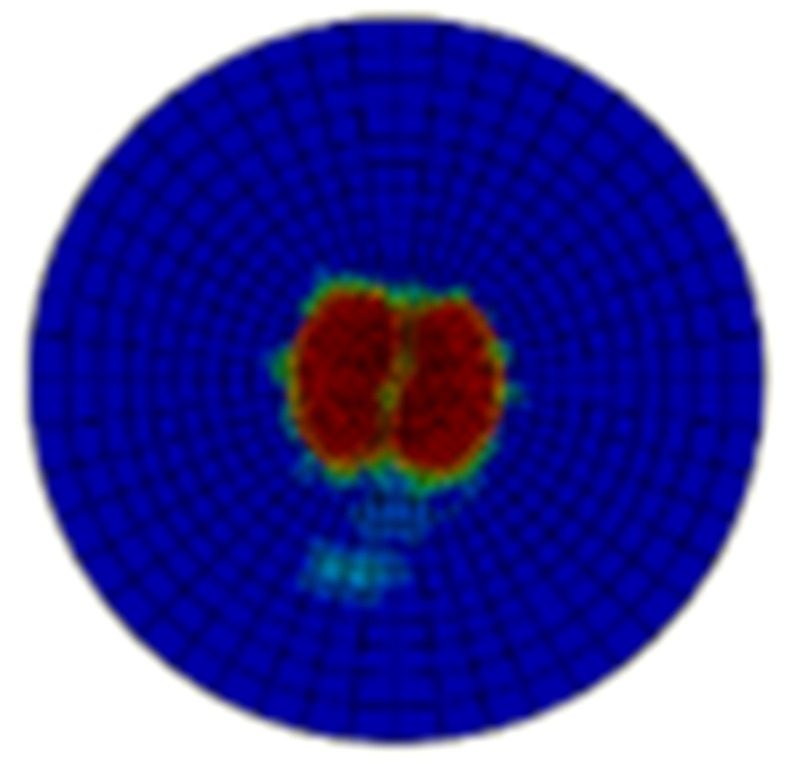
4	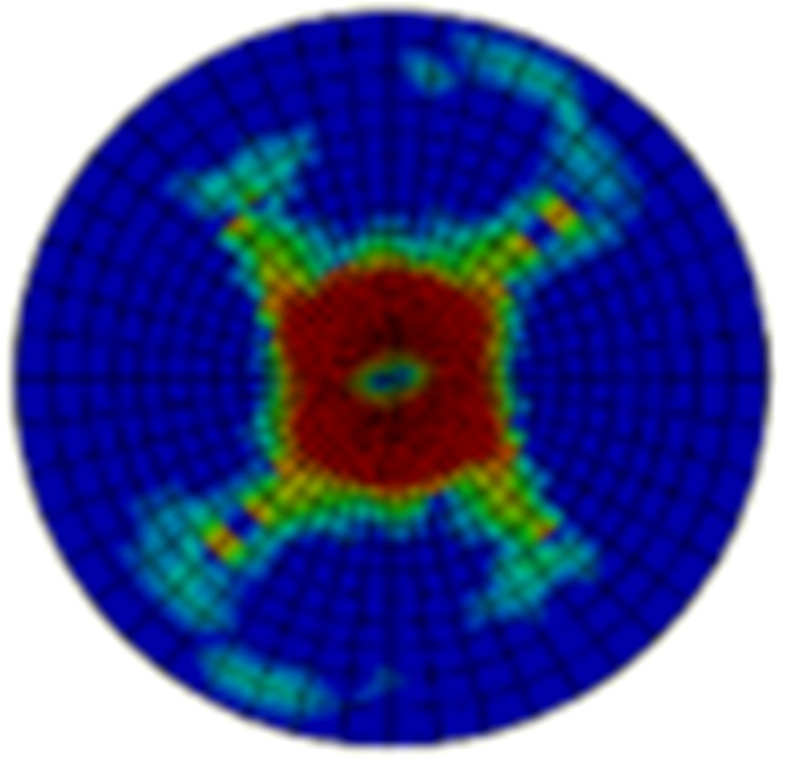	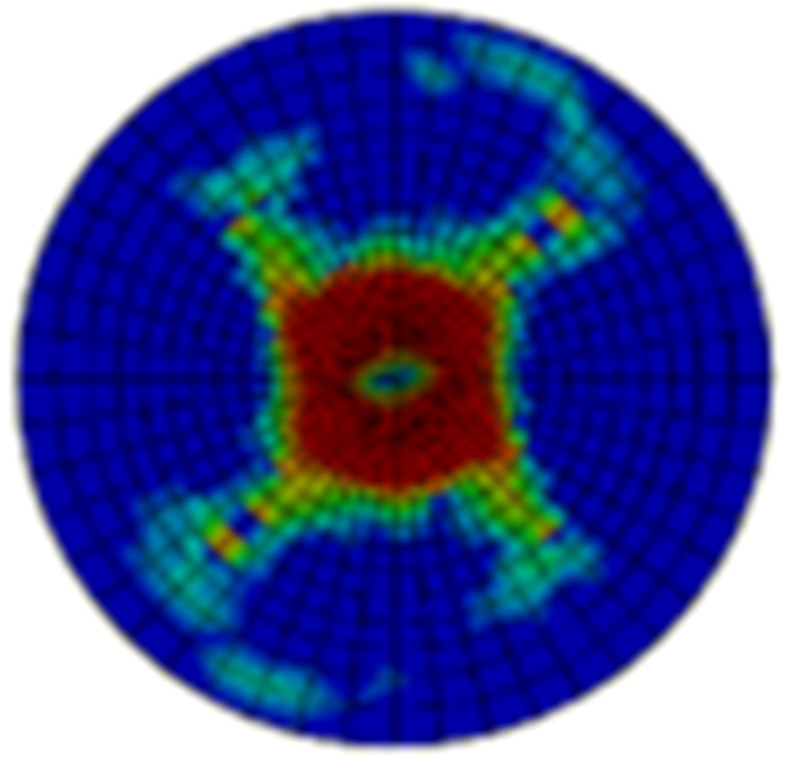	16	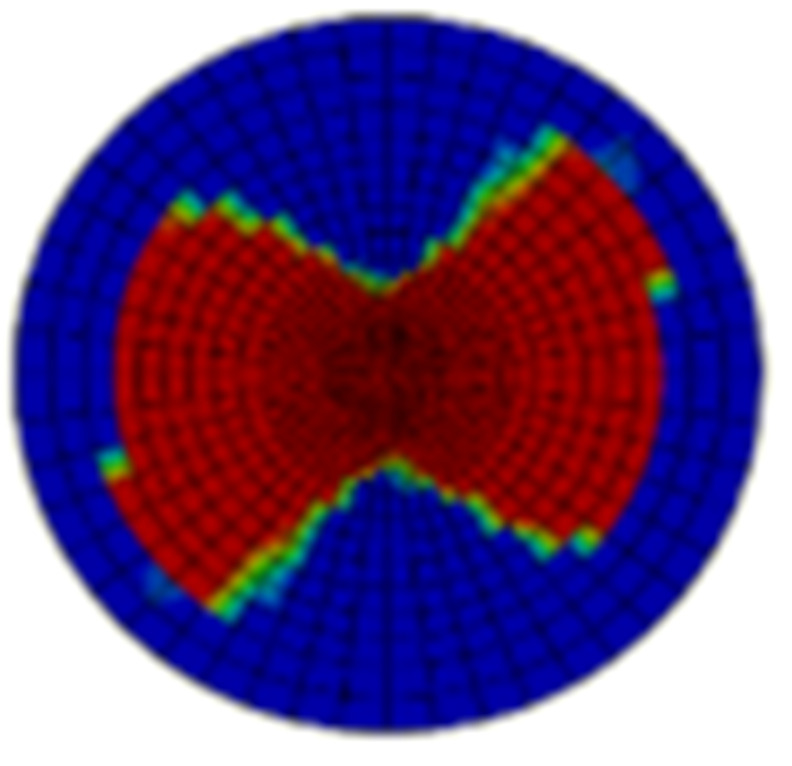	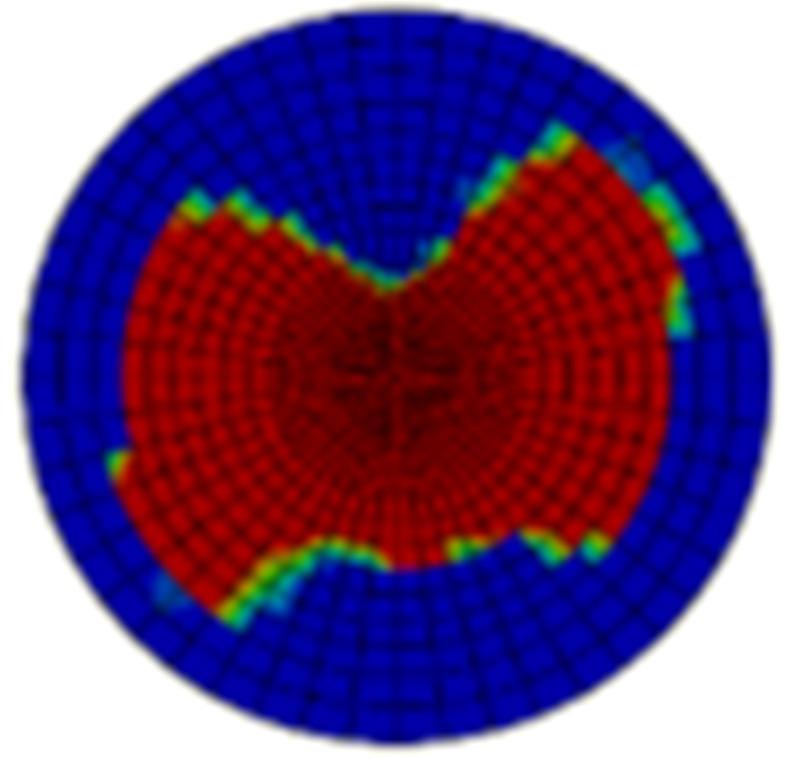
5	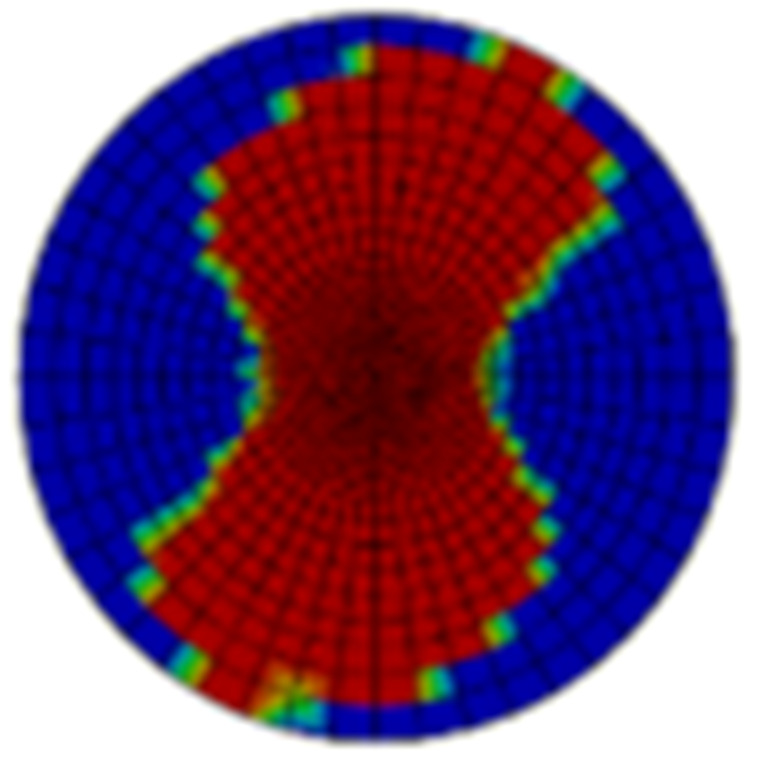	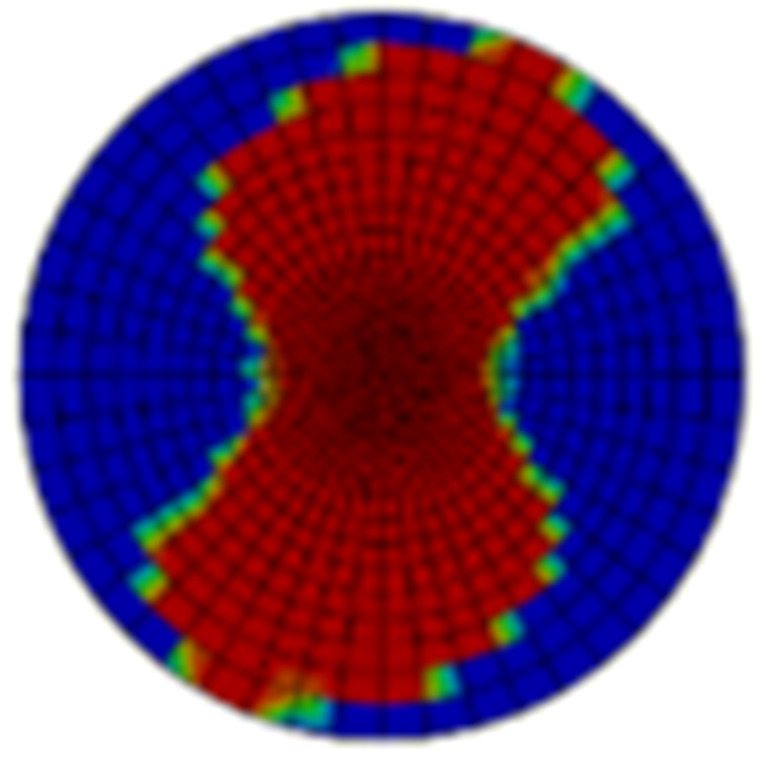	17	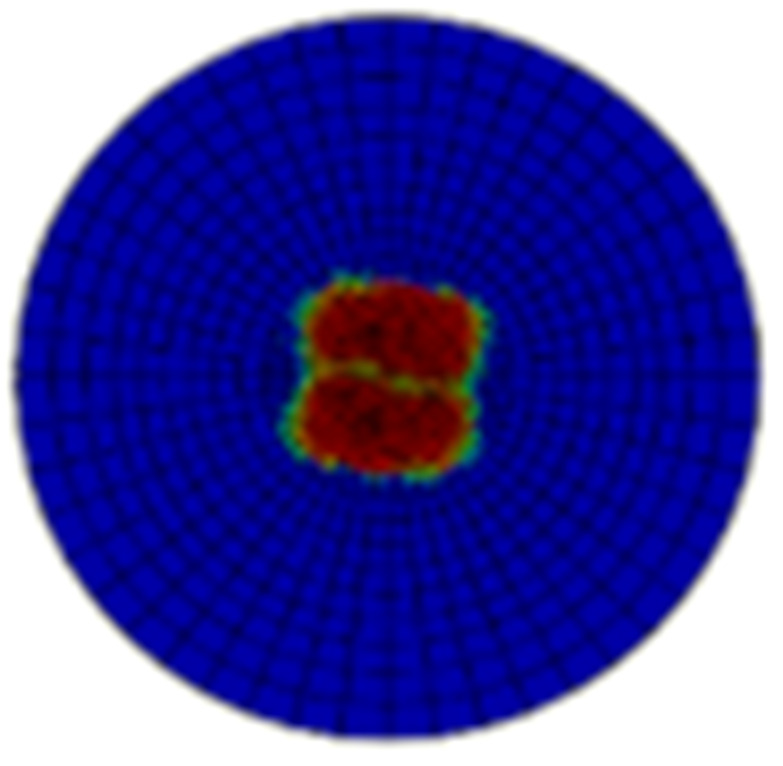	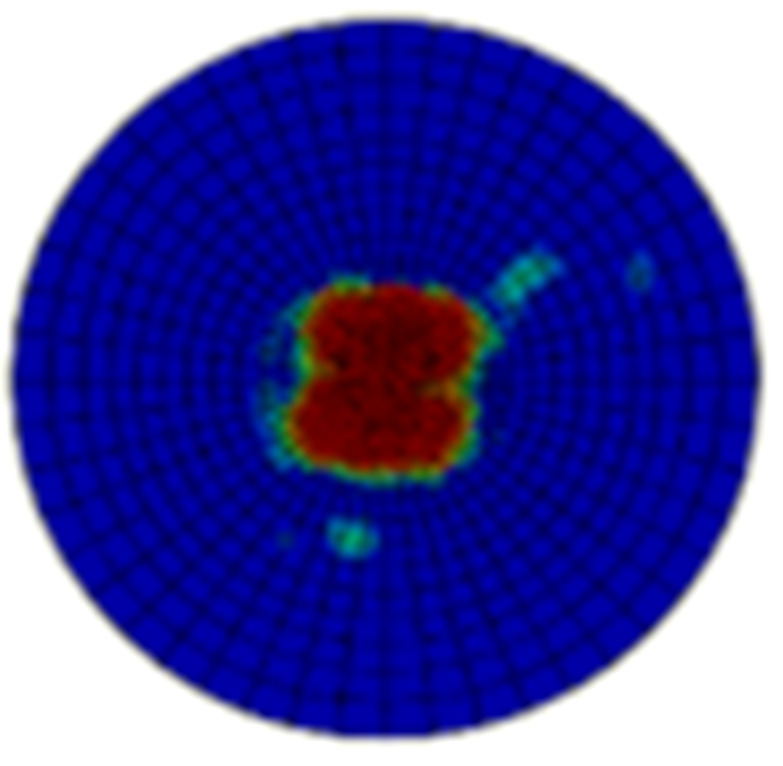
6	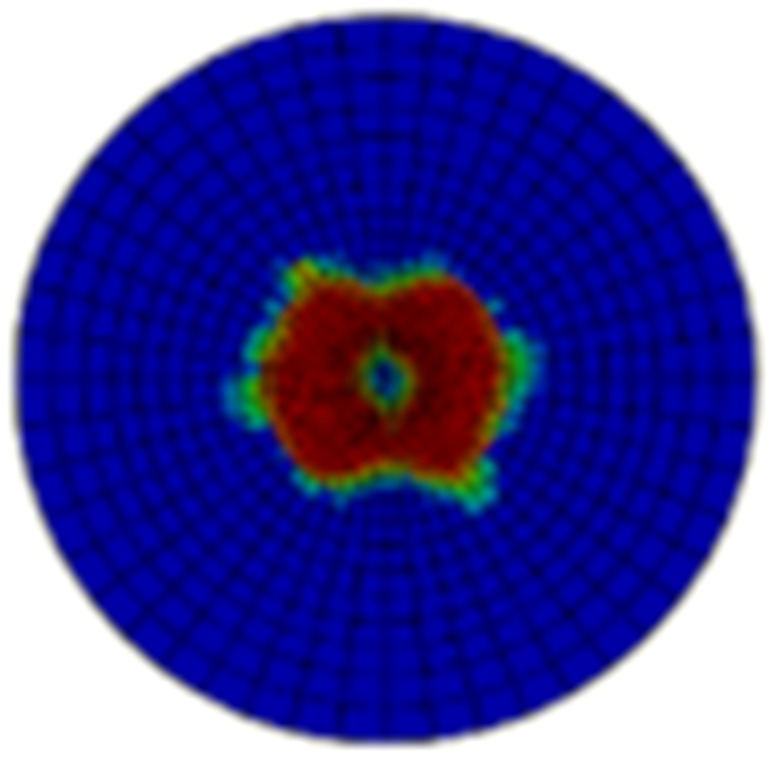	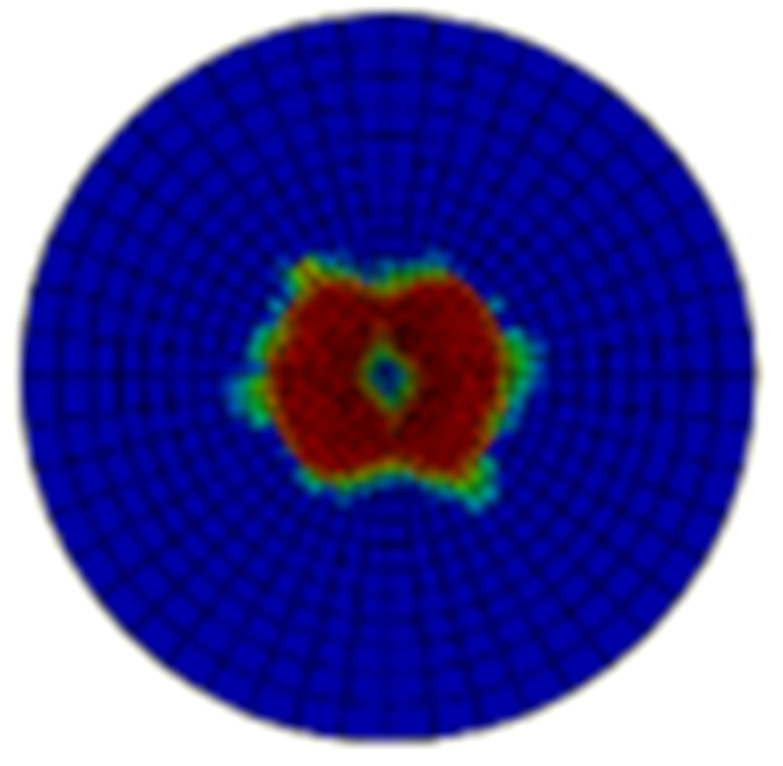	18	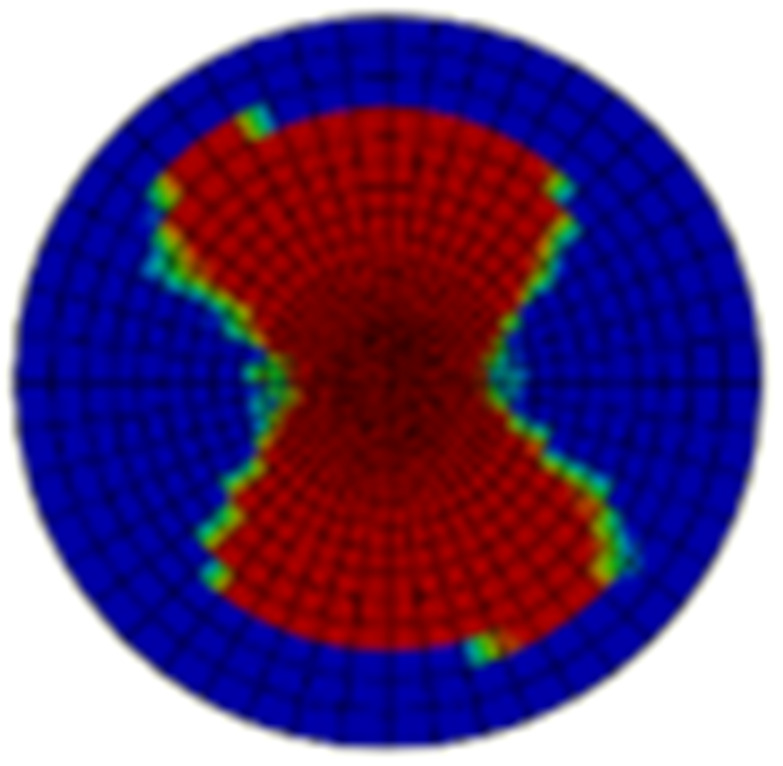	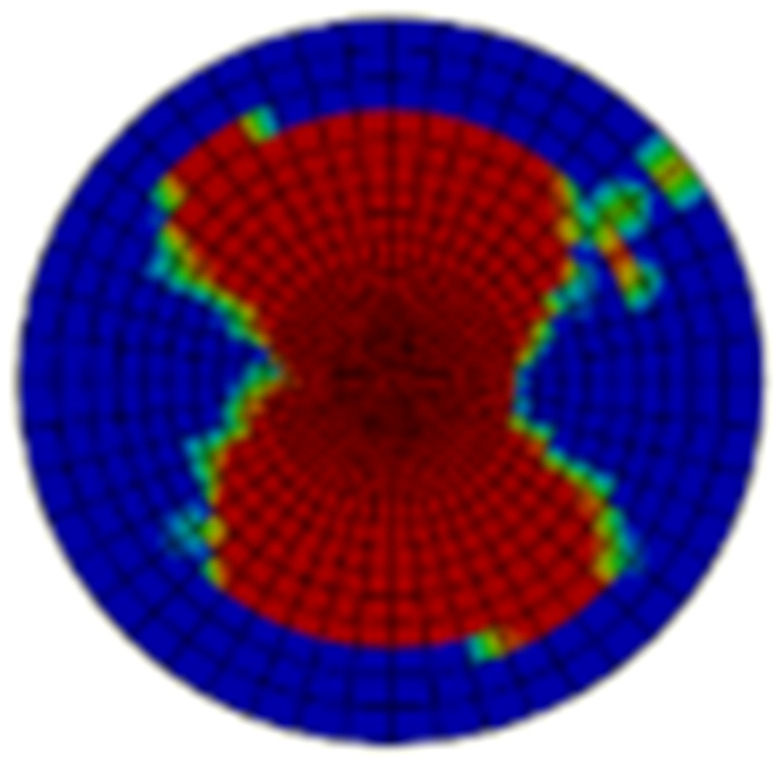
7	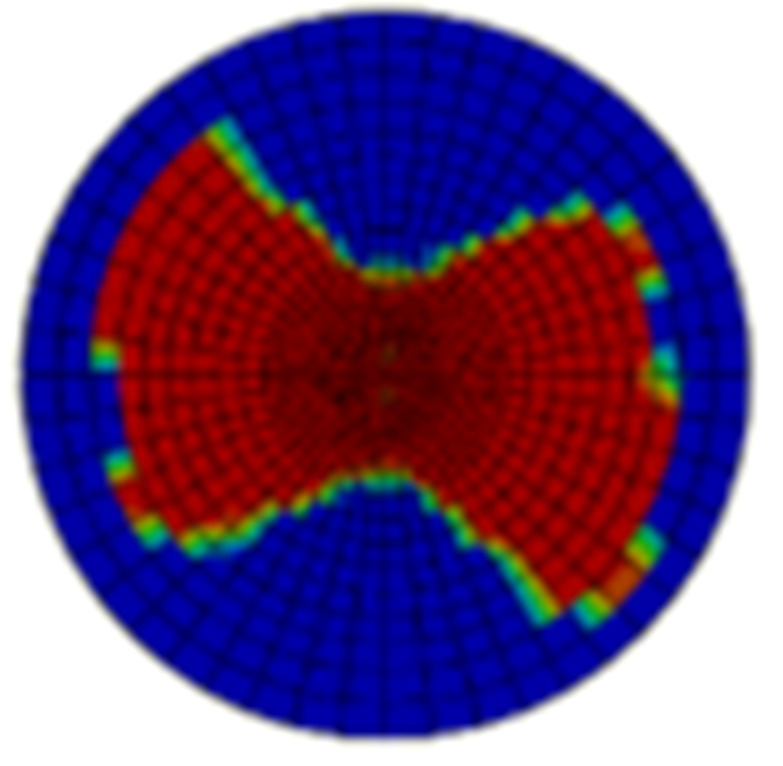	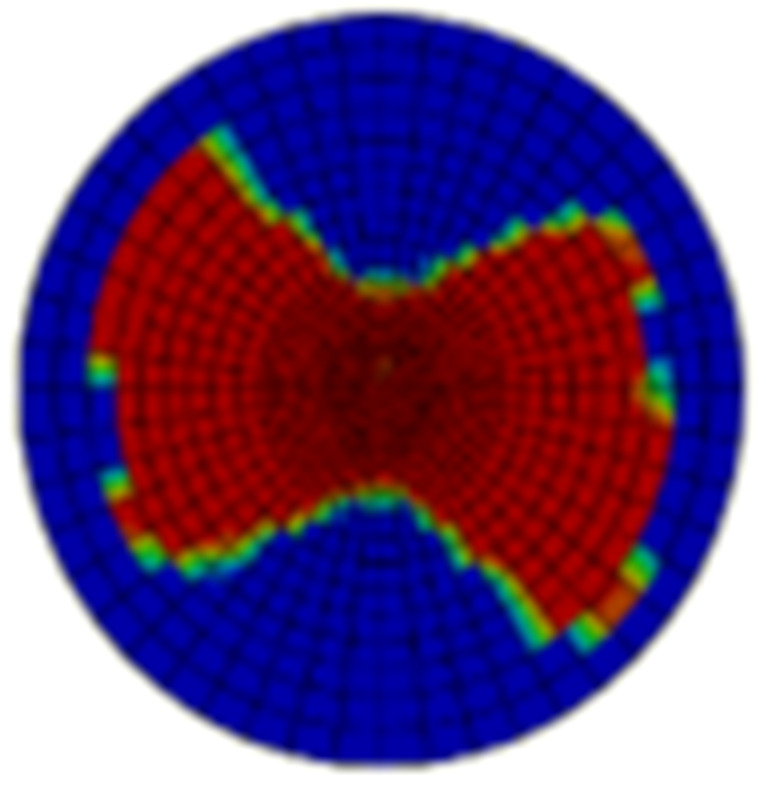	19	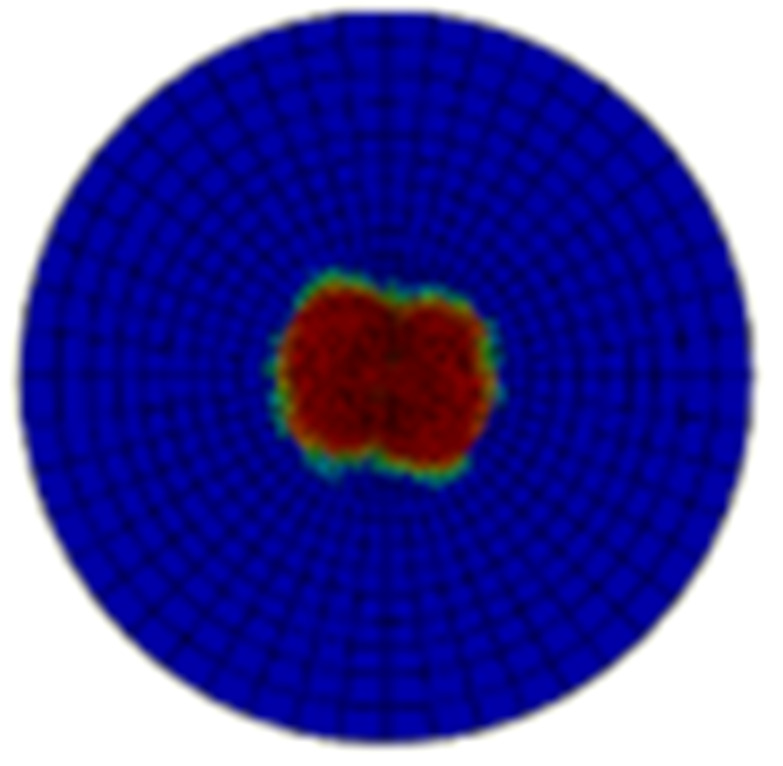	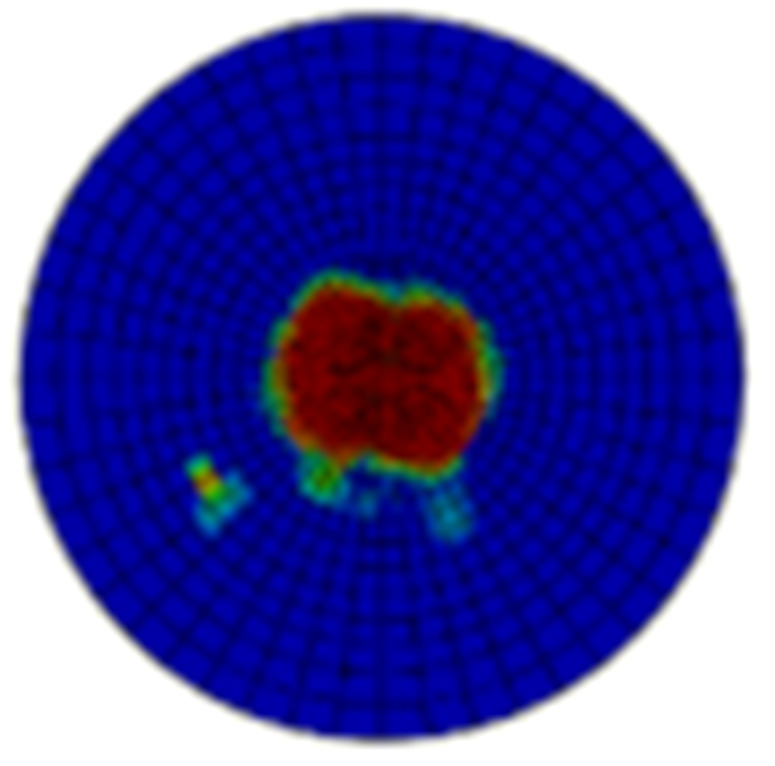
8	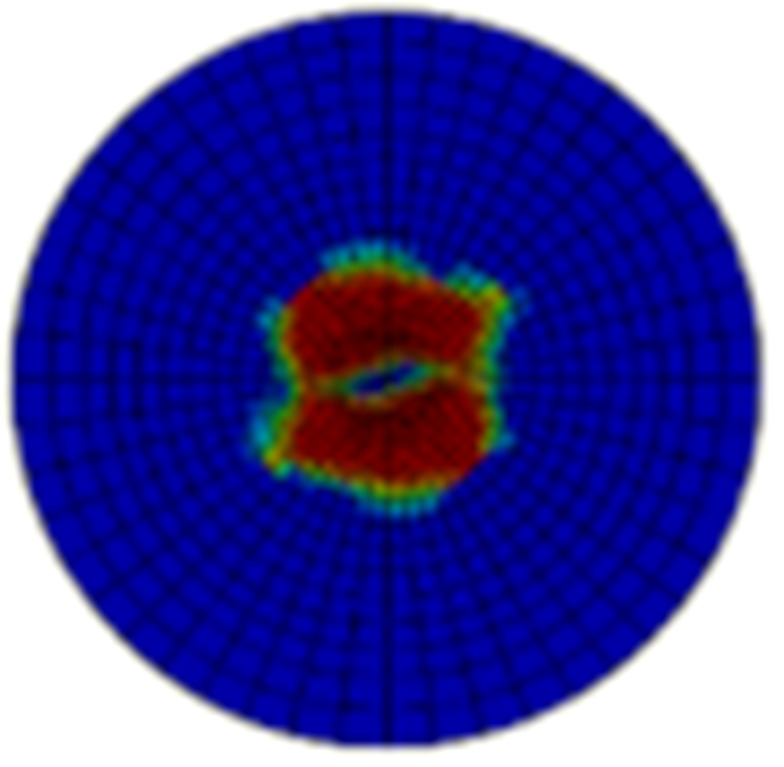	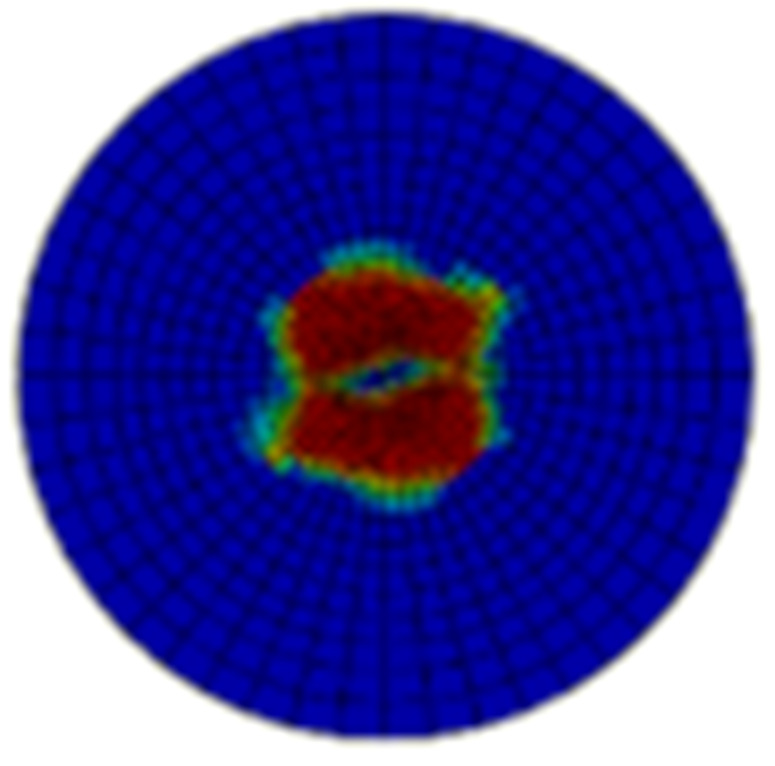	20	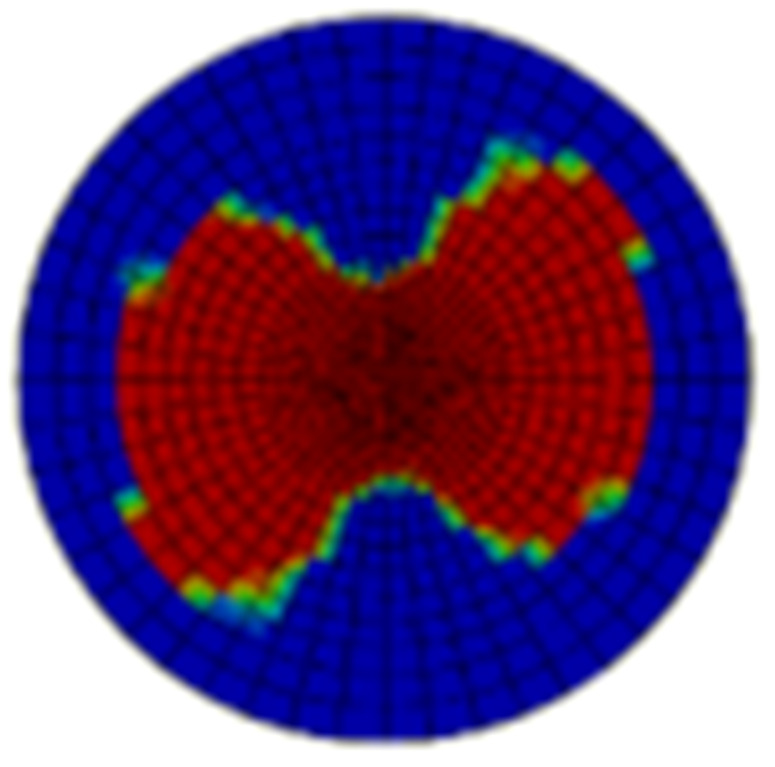	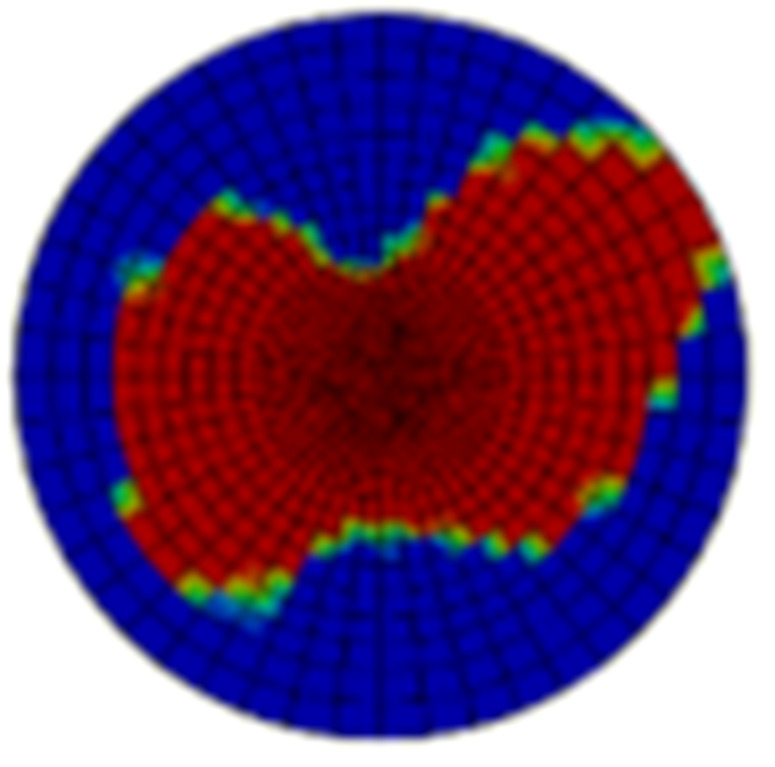
9	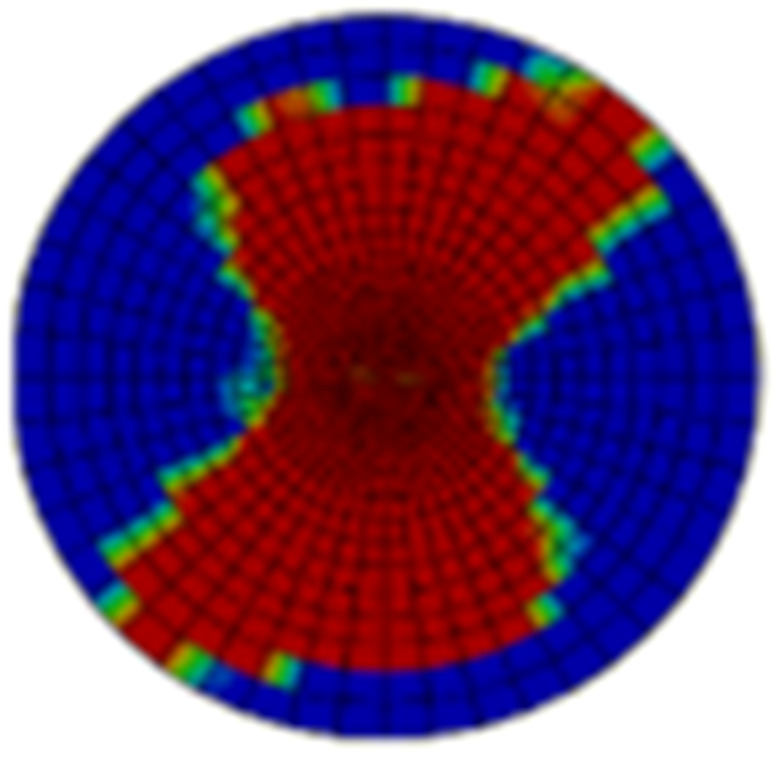	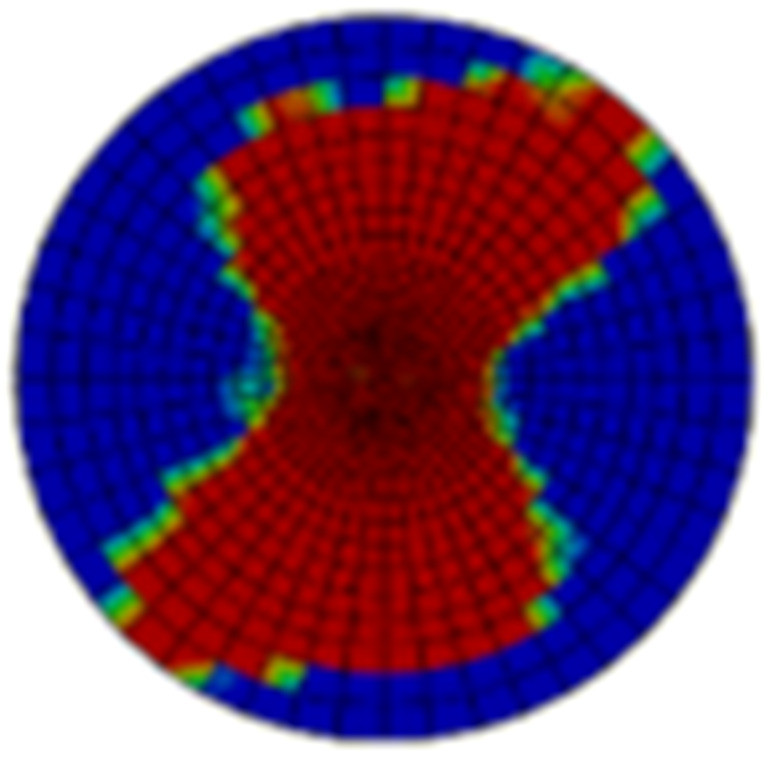	21	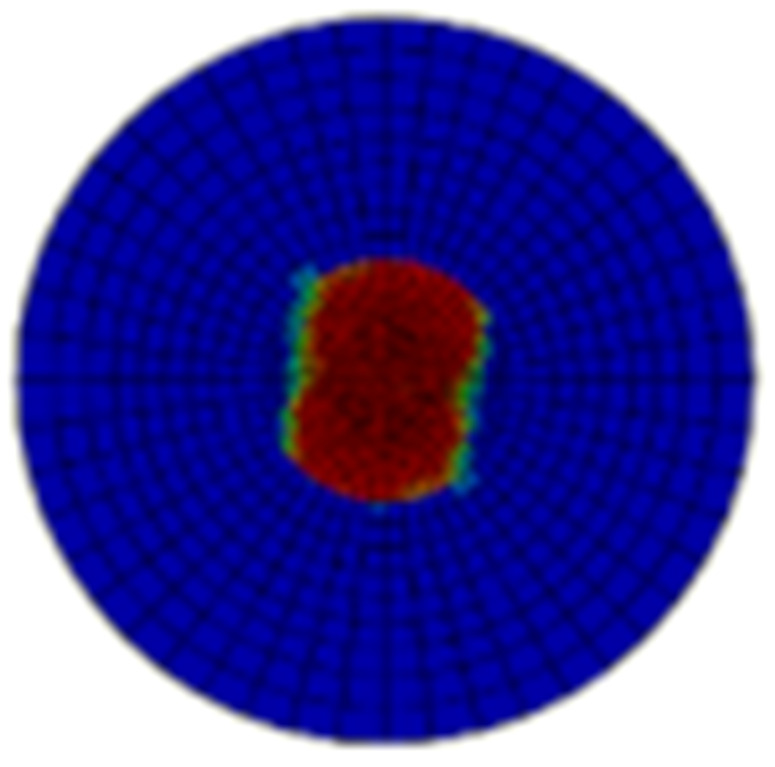	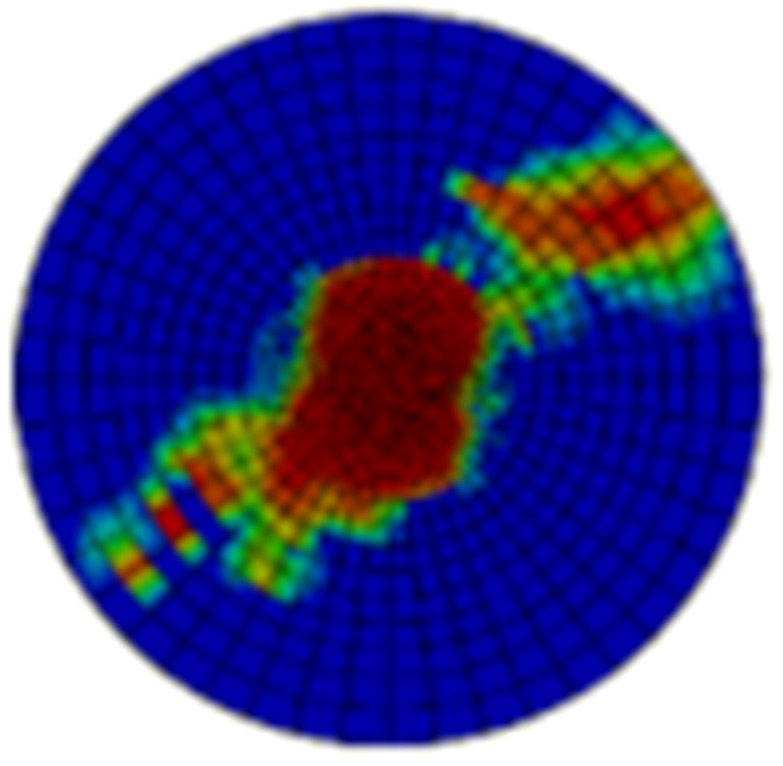
10	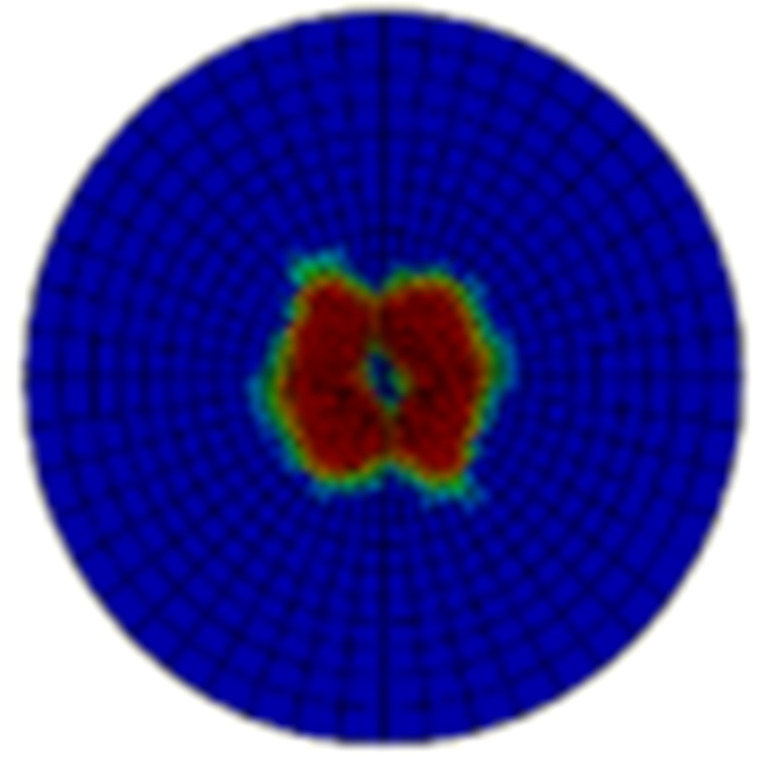	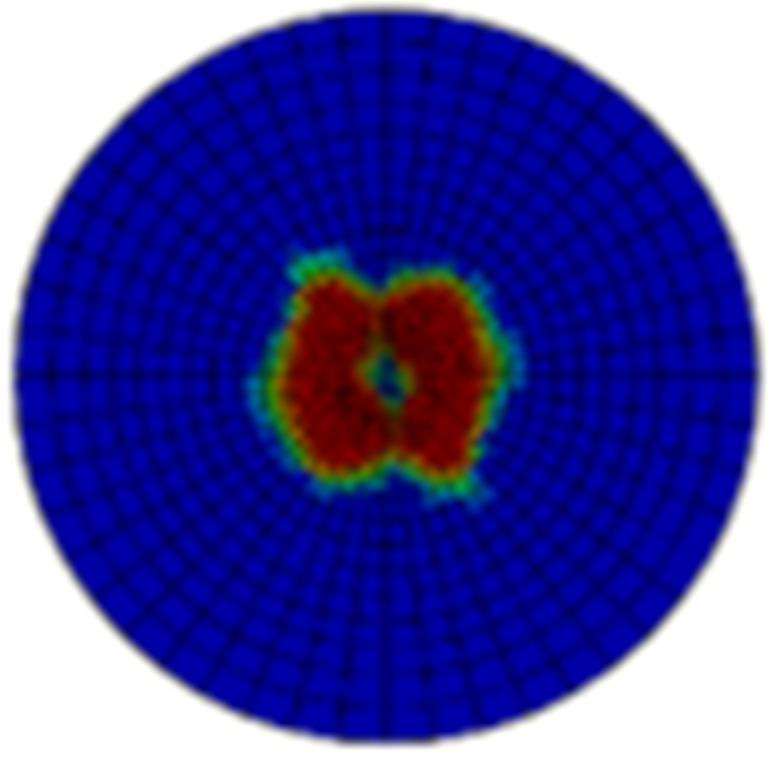	22	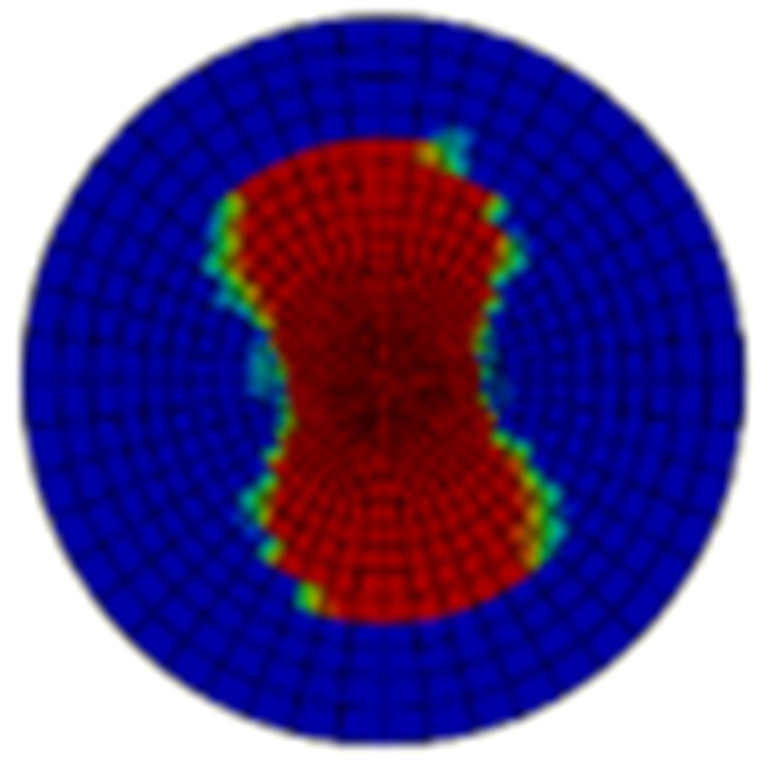	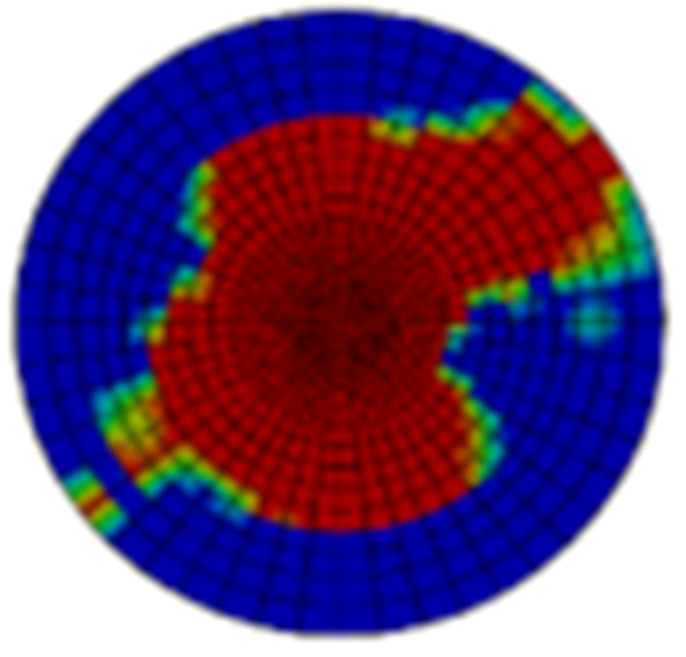
11	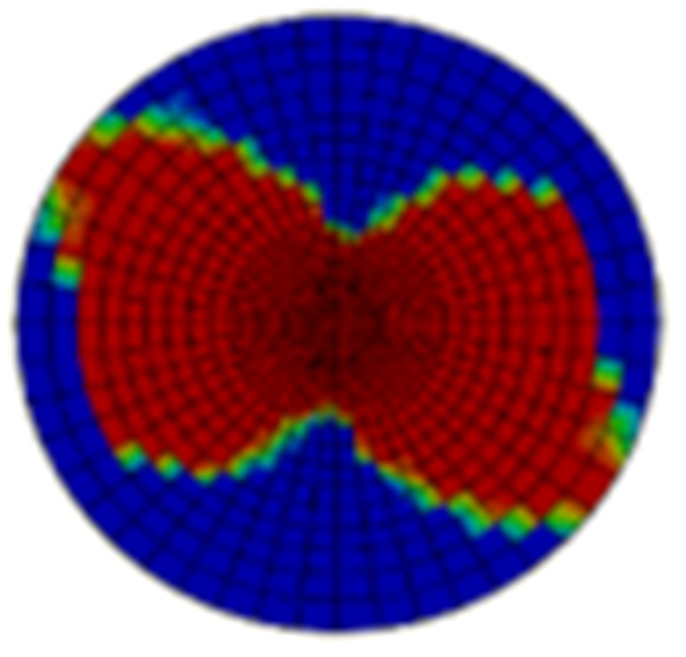	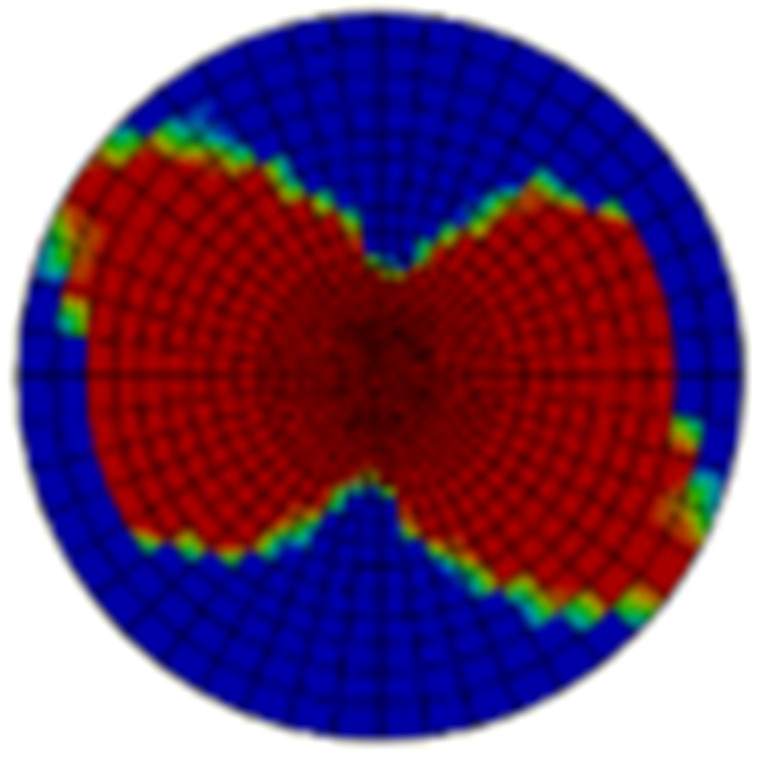	23	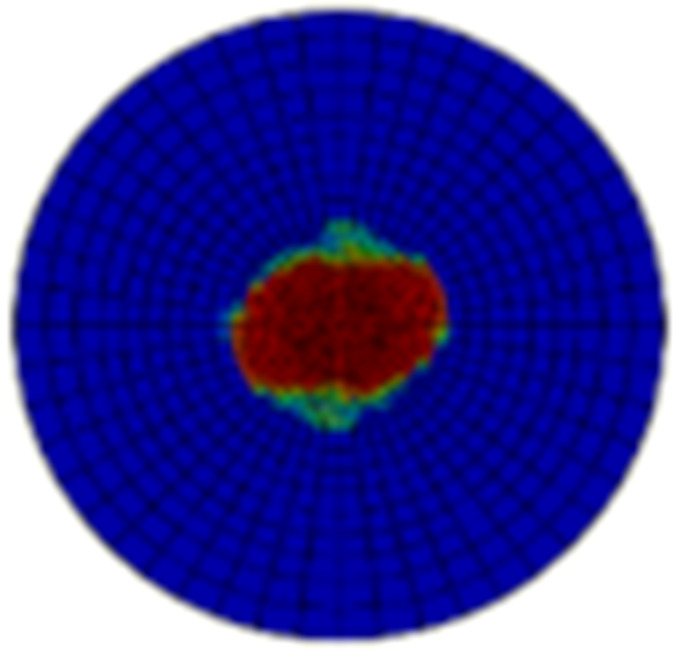	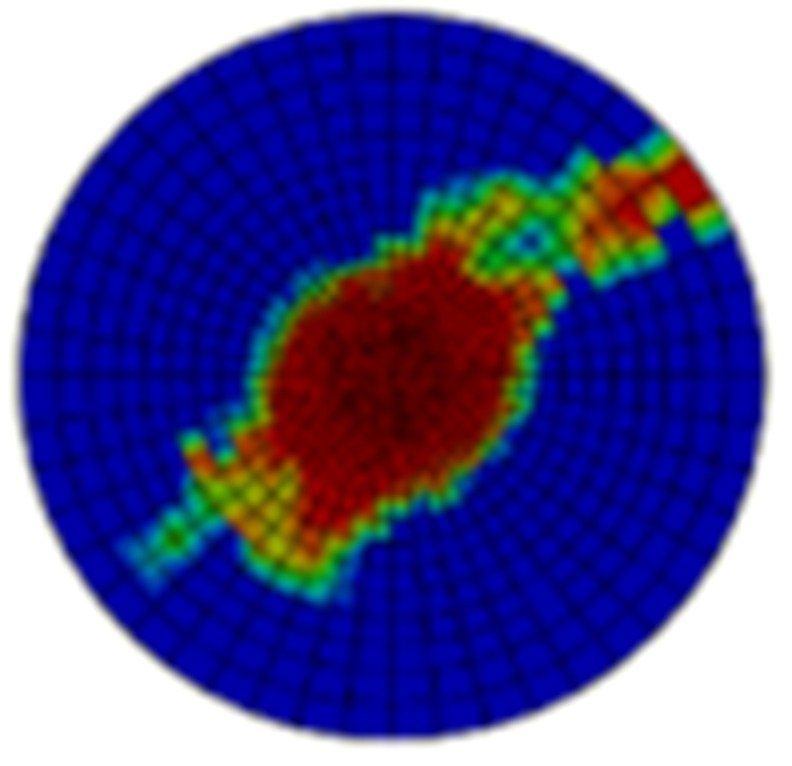
12	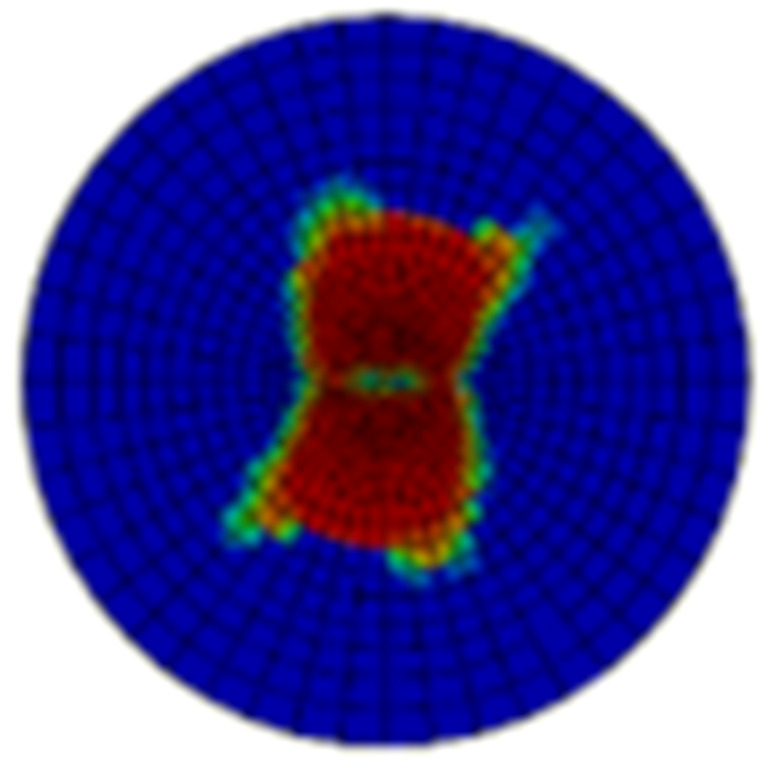	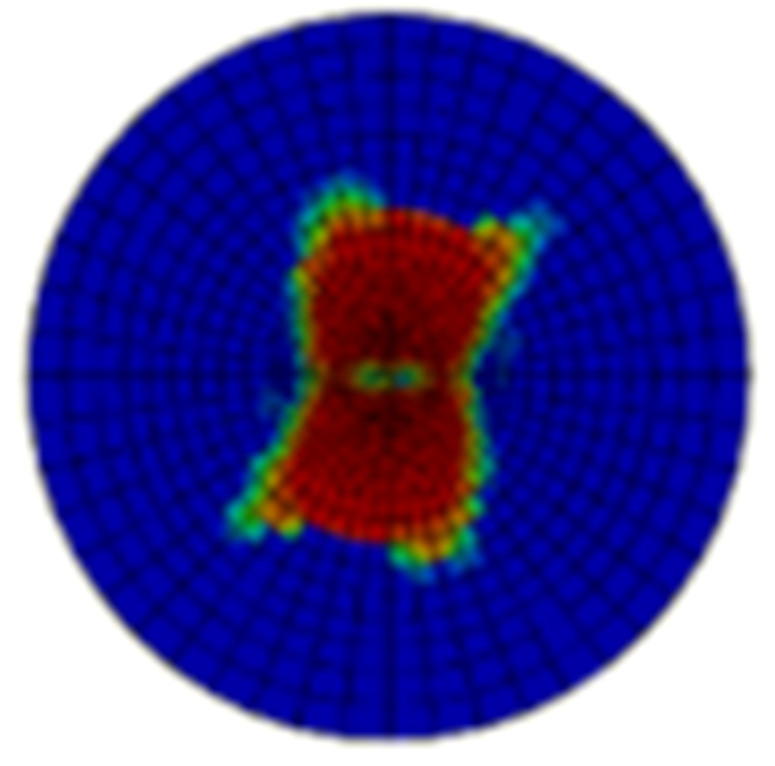	-		

**Table 6 materials-15-05029-t006:** Cohesive failure status of the composite laminate with an impact energy of 50 J.

Layer No.	Impacted Results	Shear Results	Layer No.	Impacted Results	Shear Results
1	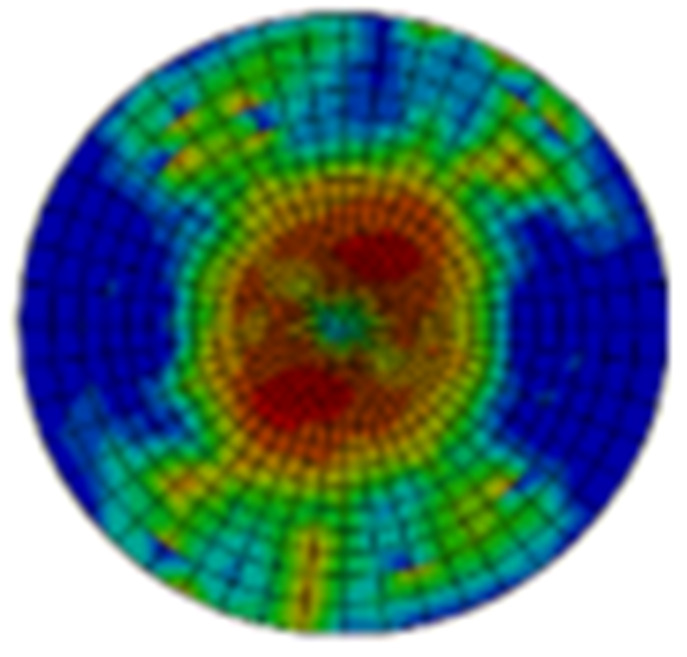	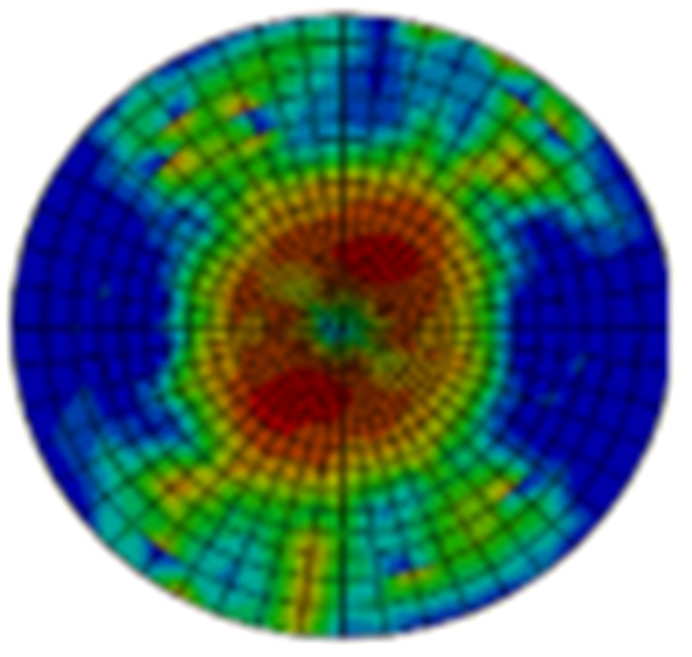	13	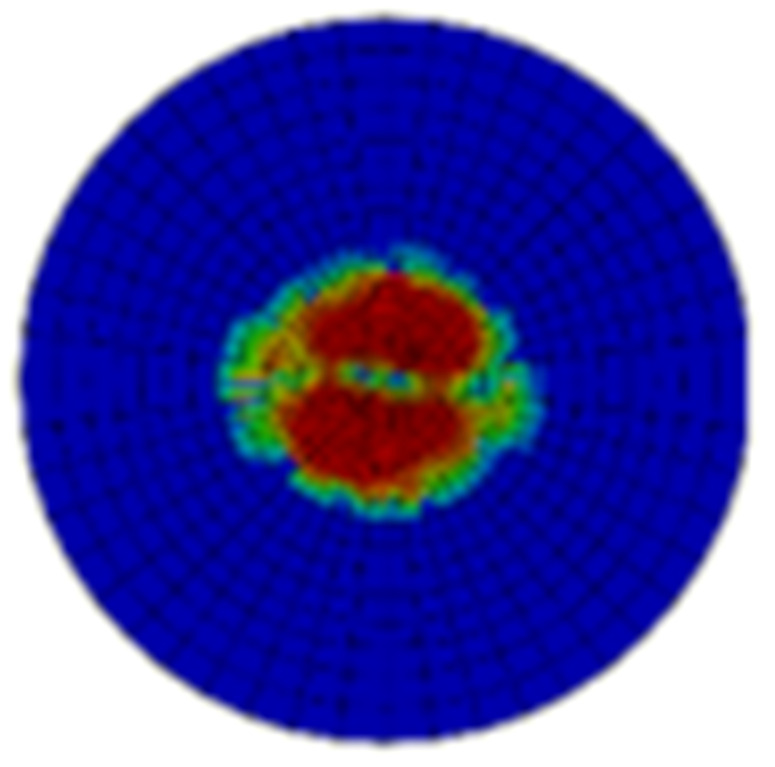	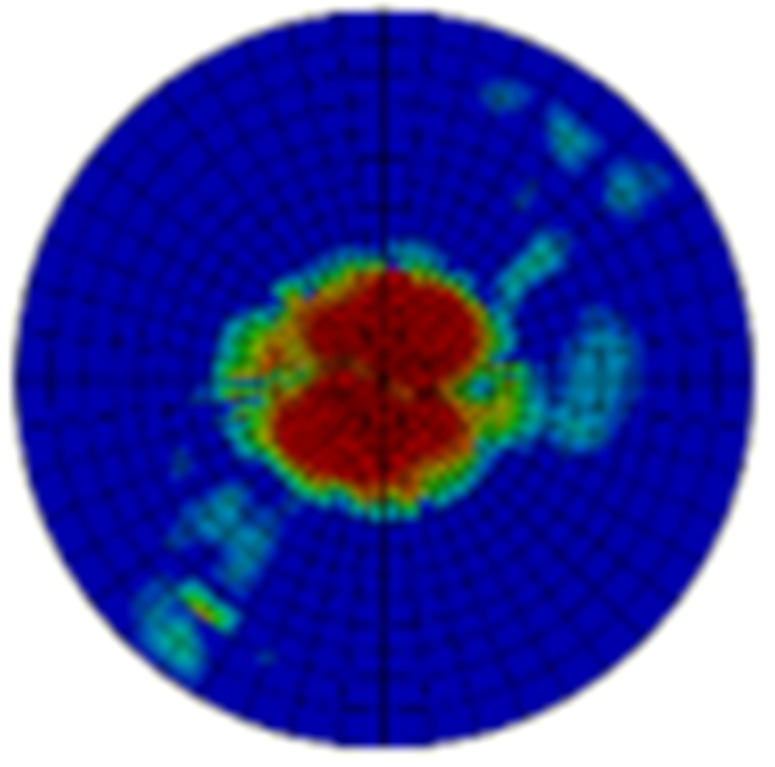
2	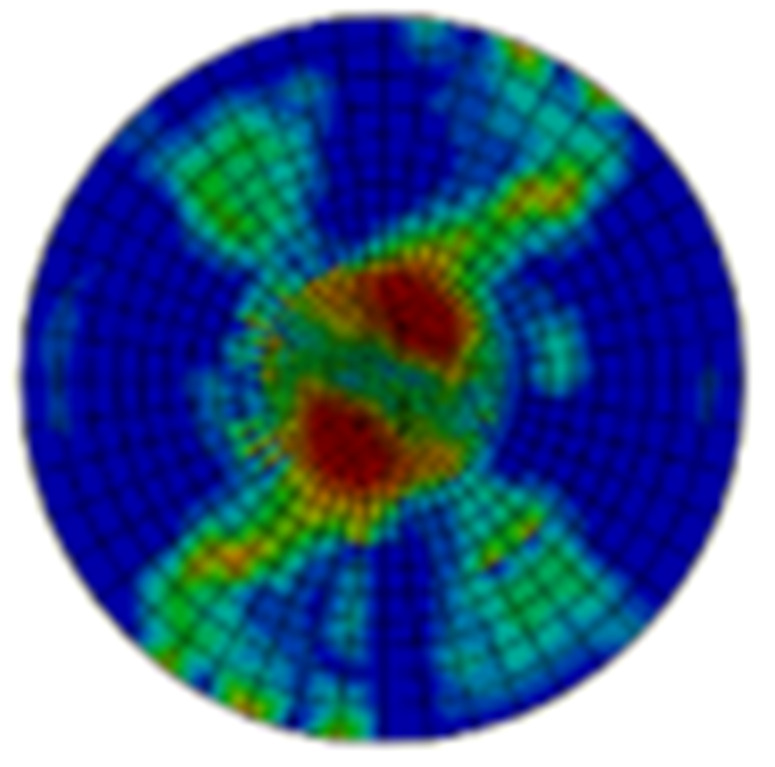	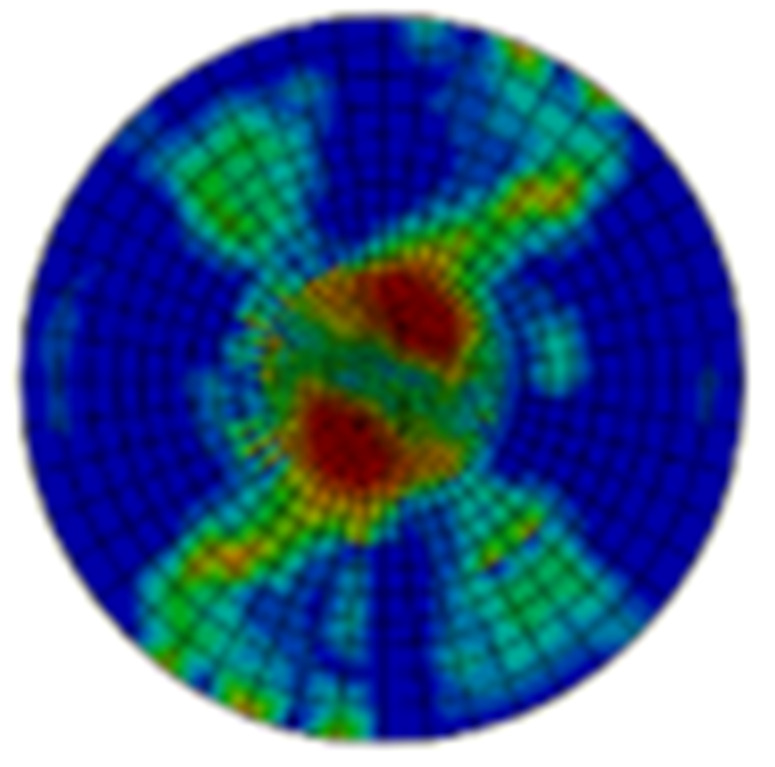	14	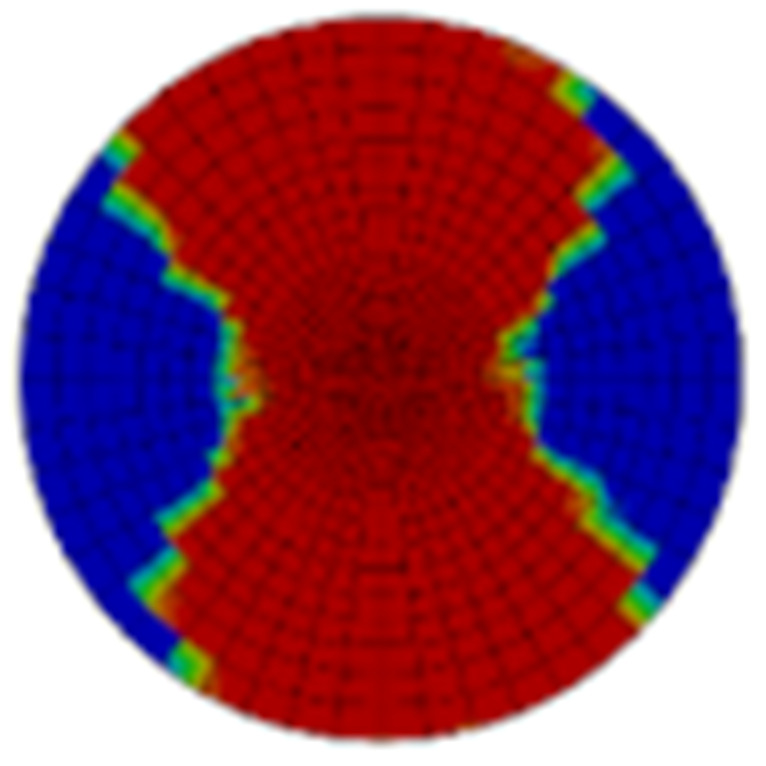	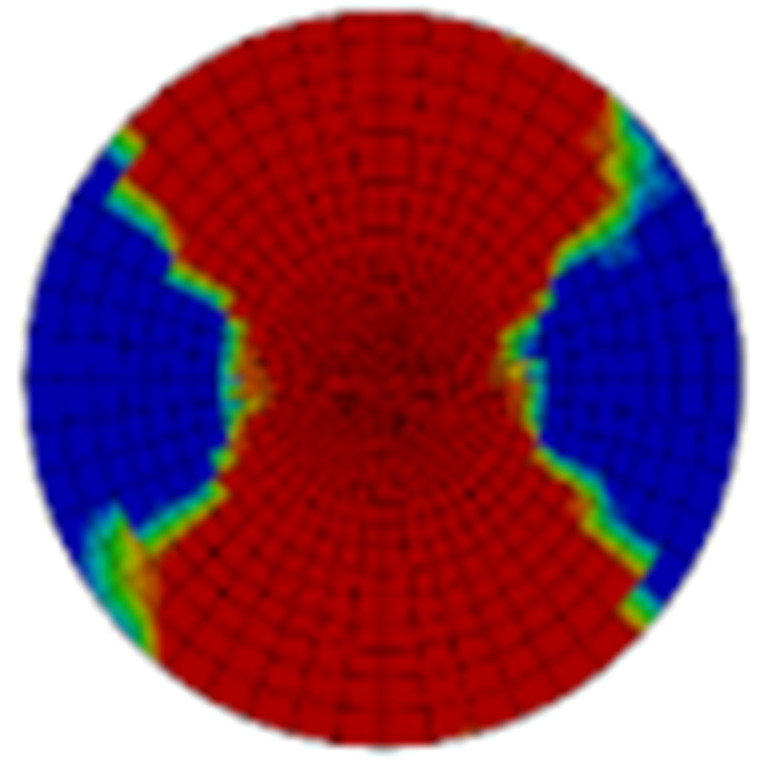
3	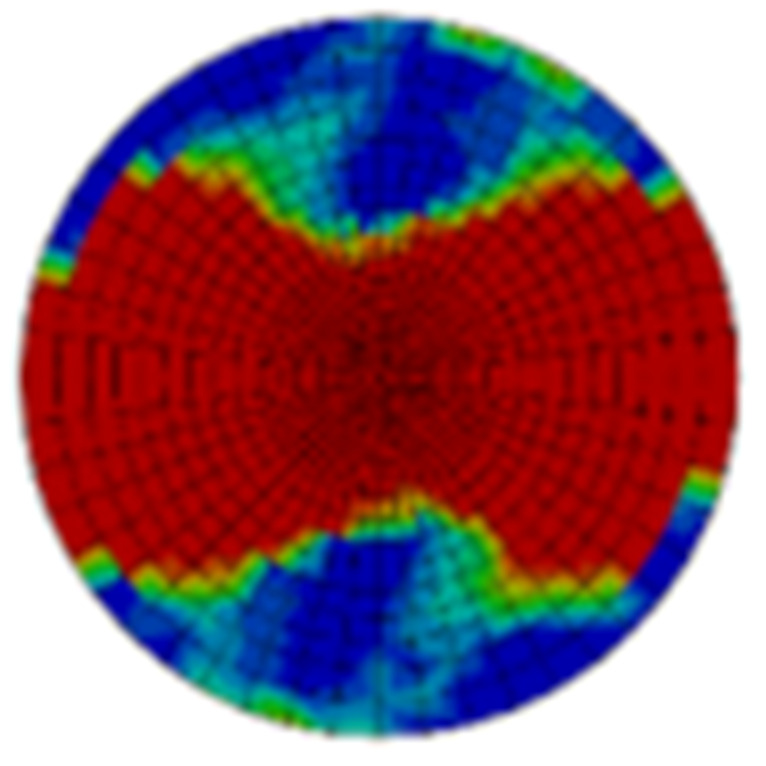	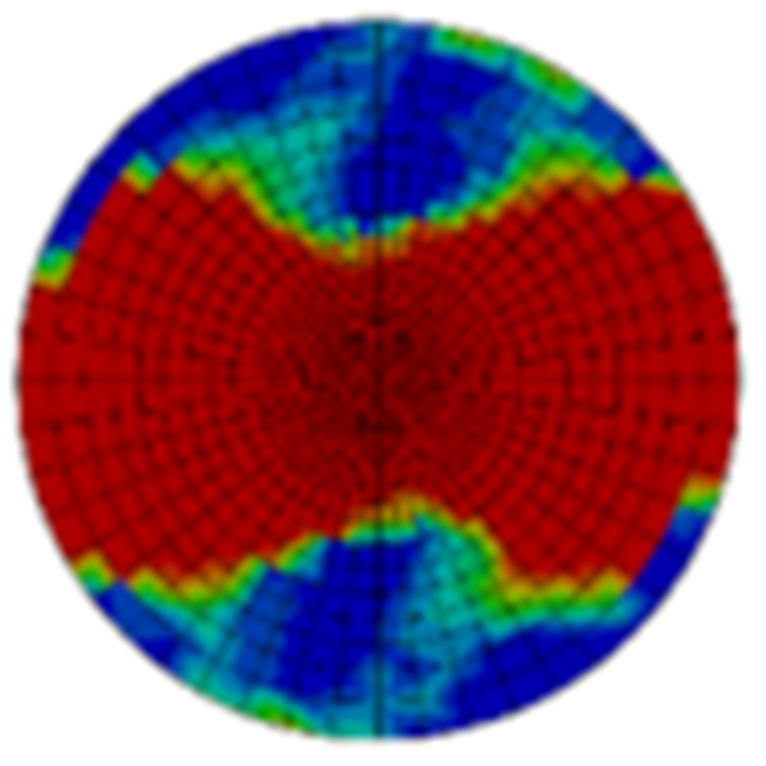	15	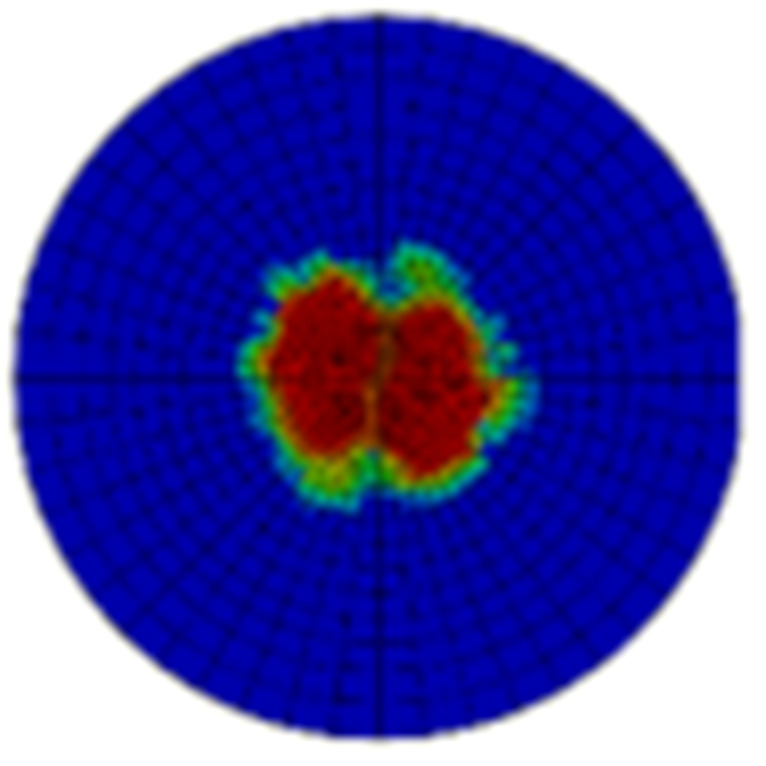	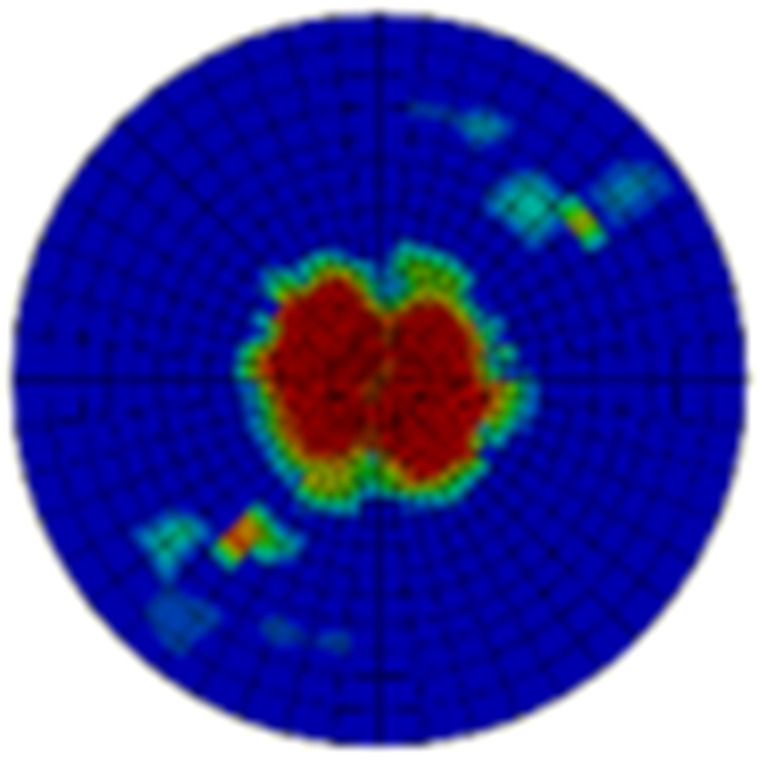
4	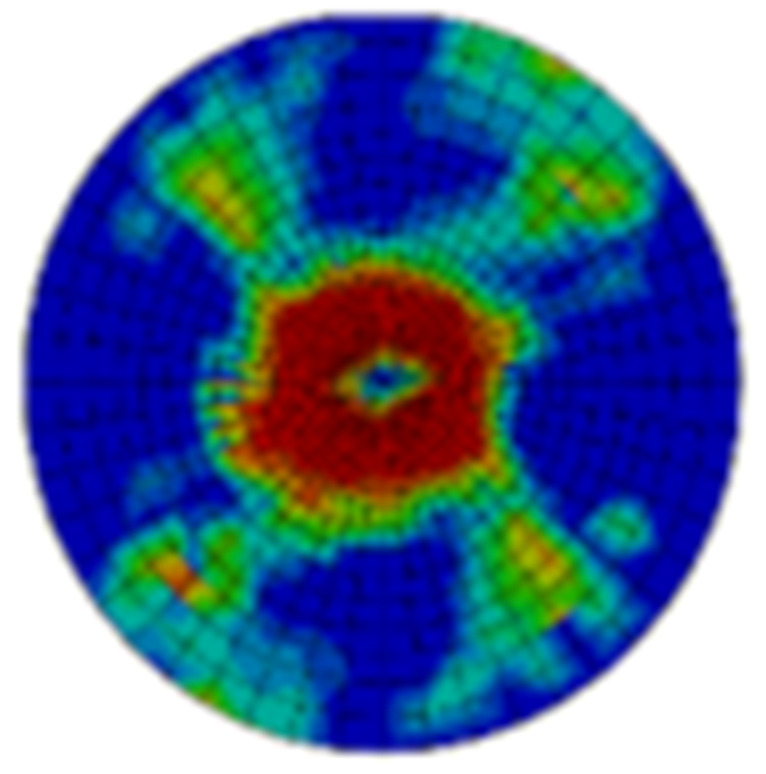	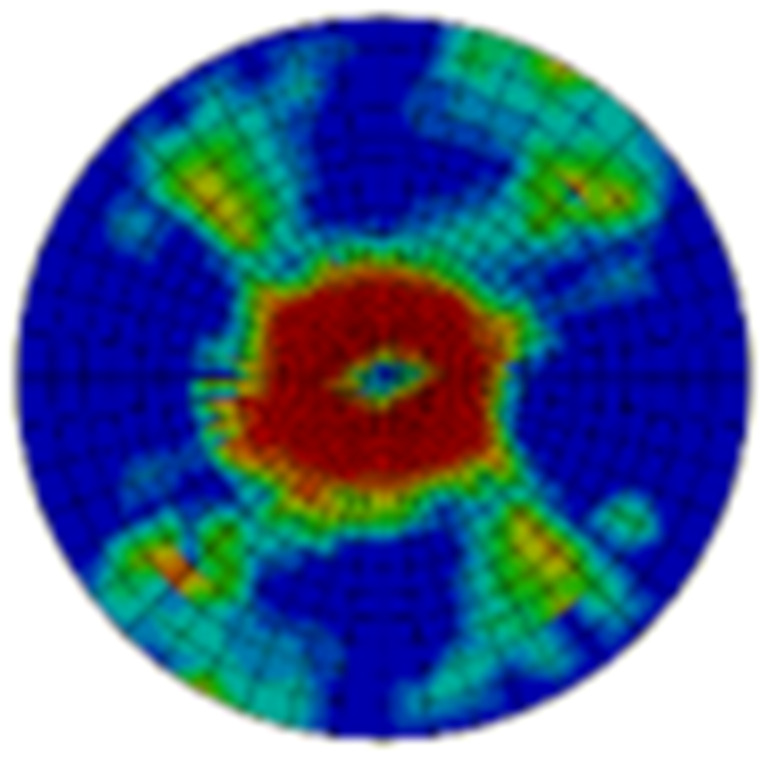	16	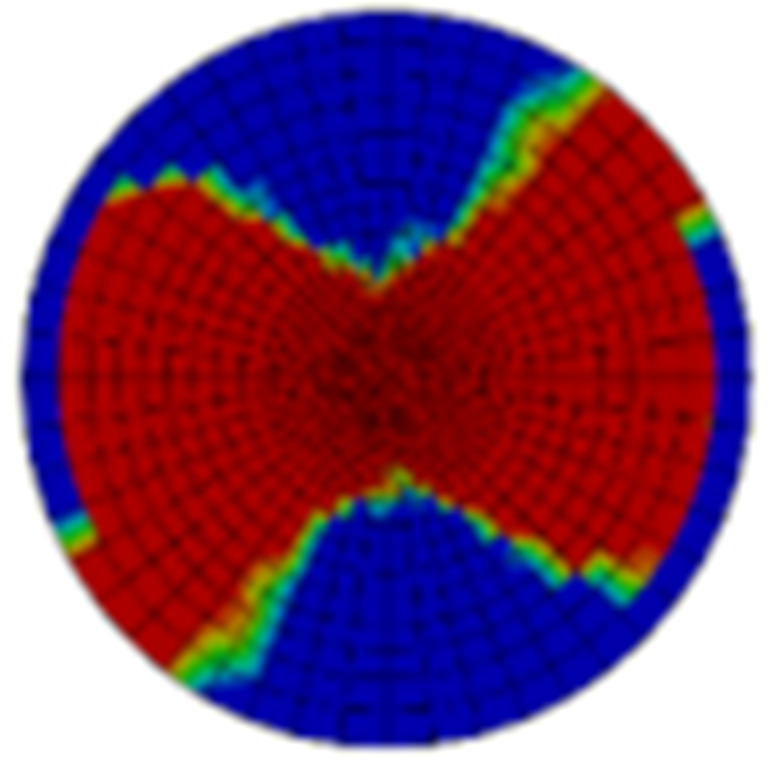	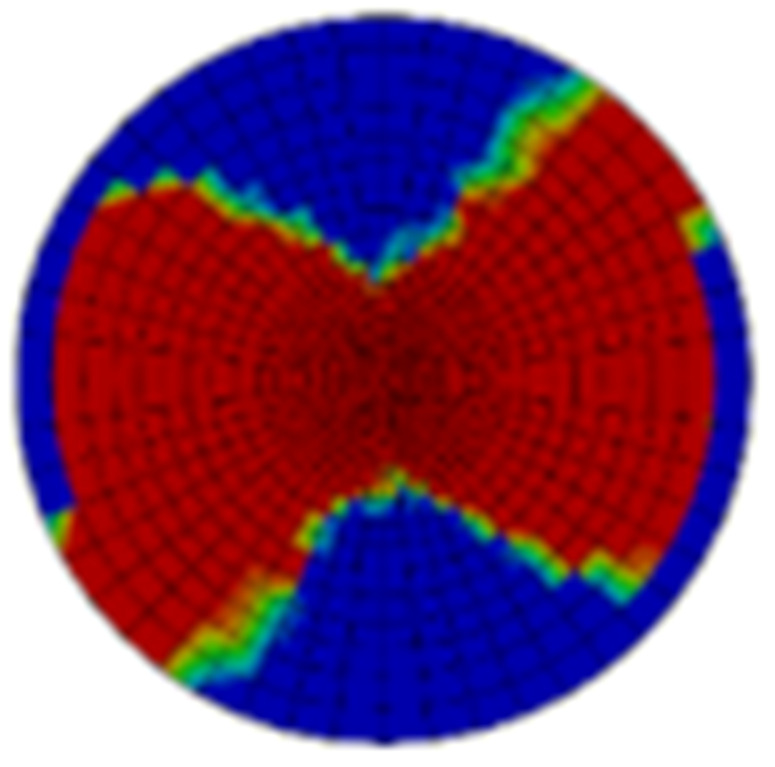
5	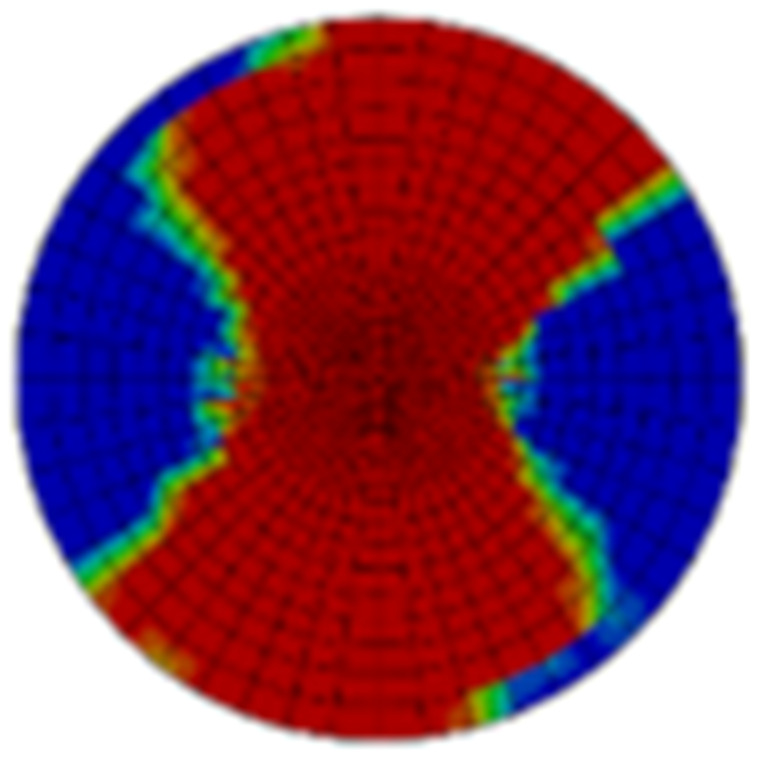	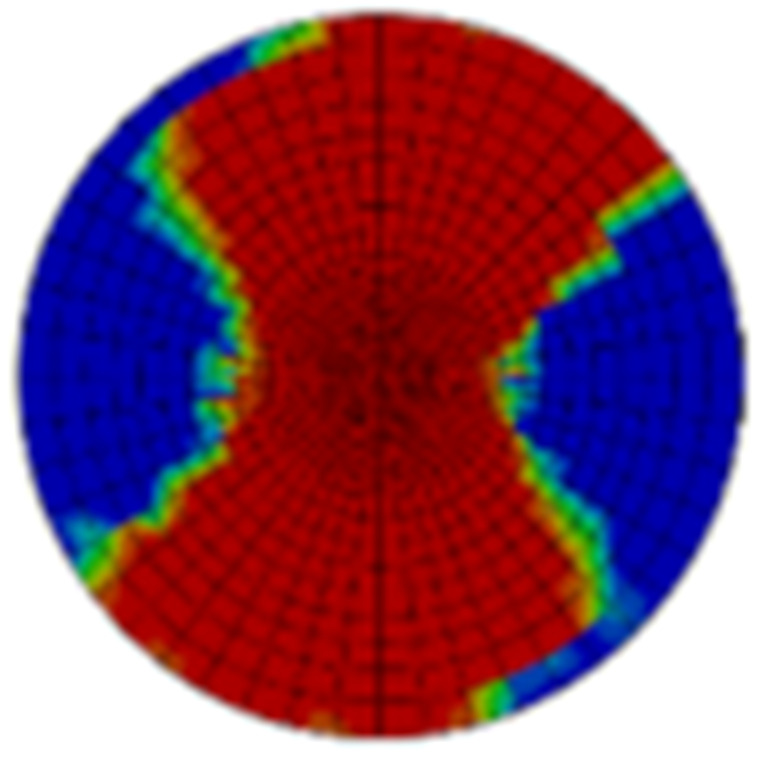	17	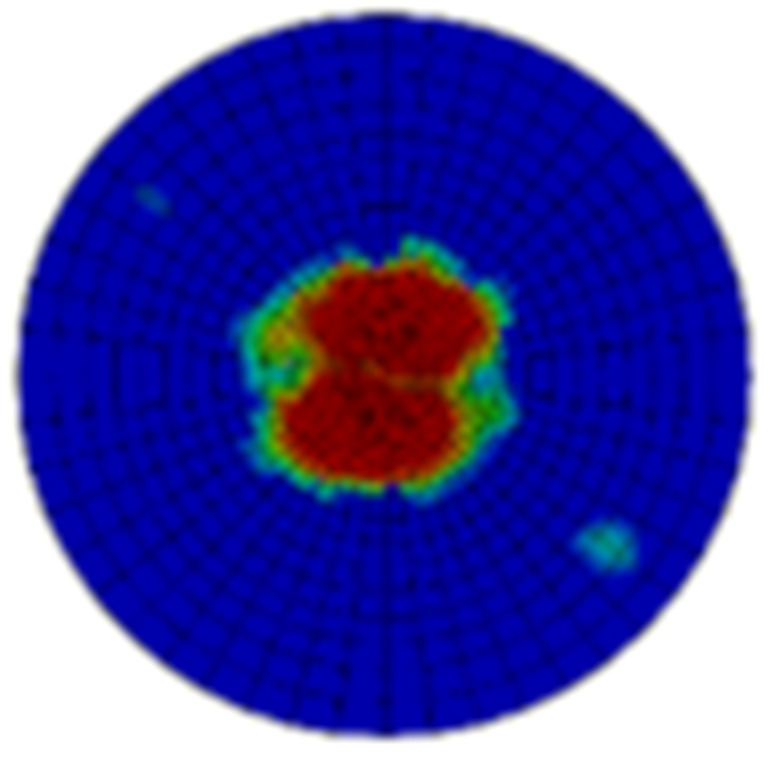	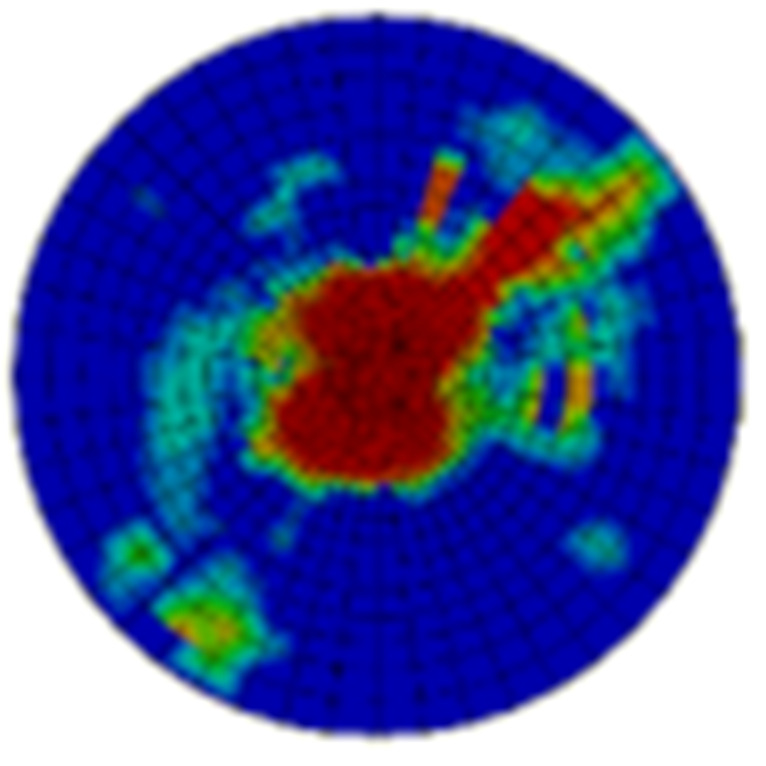
6	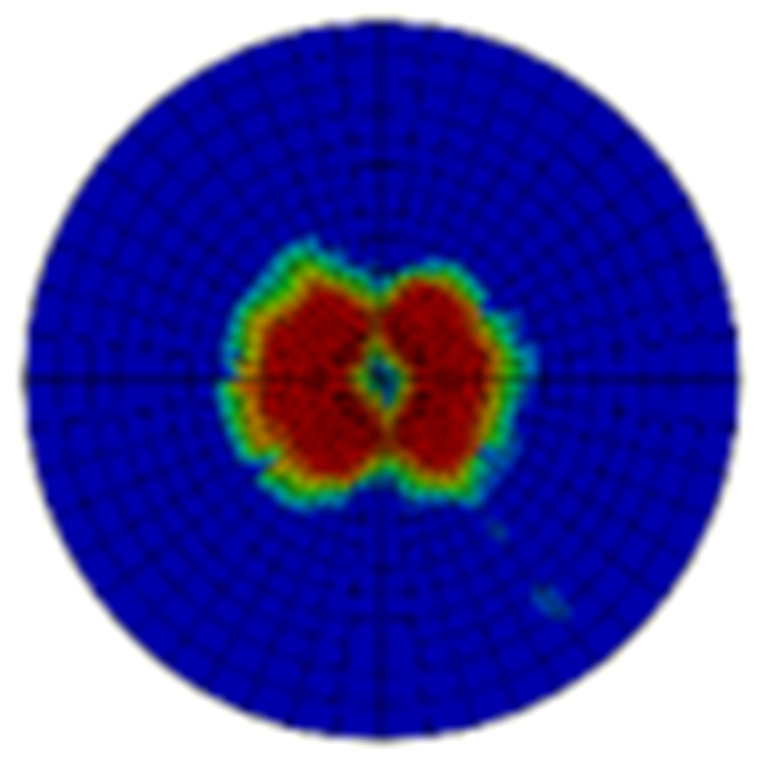	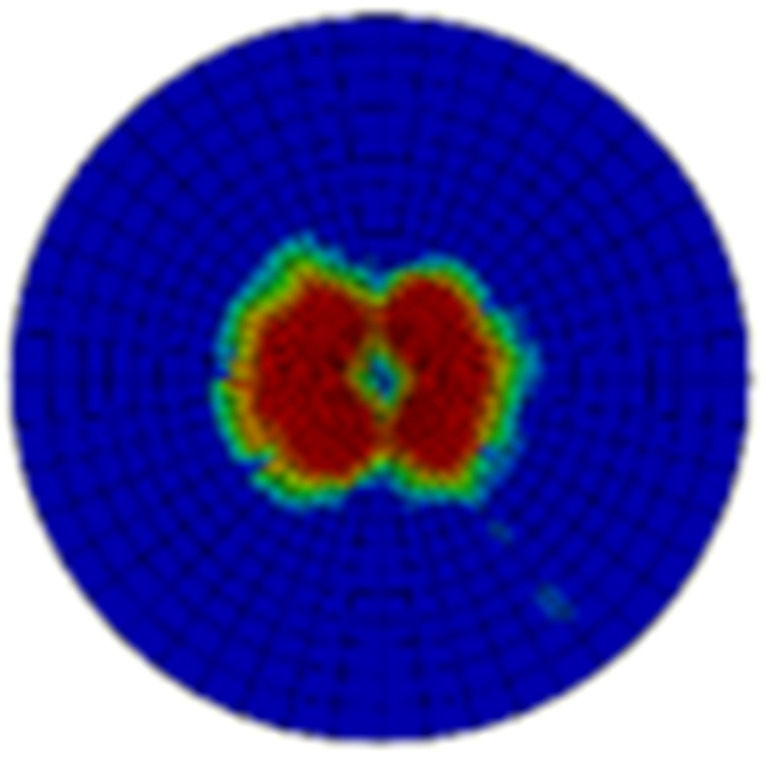	18	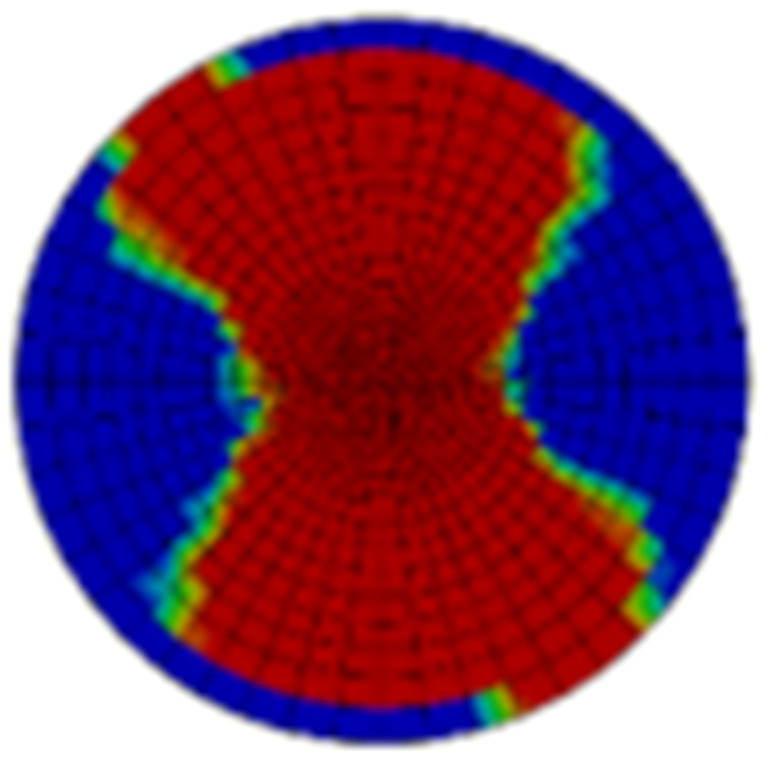	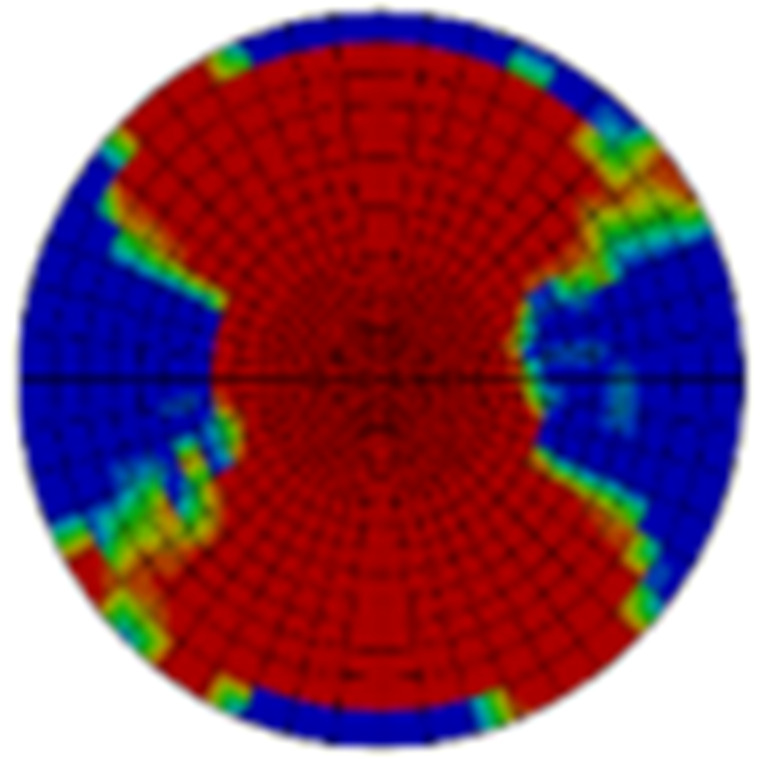
7	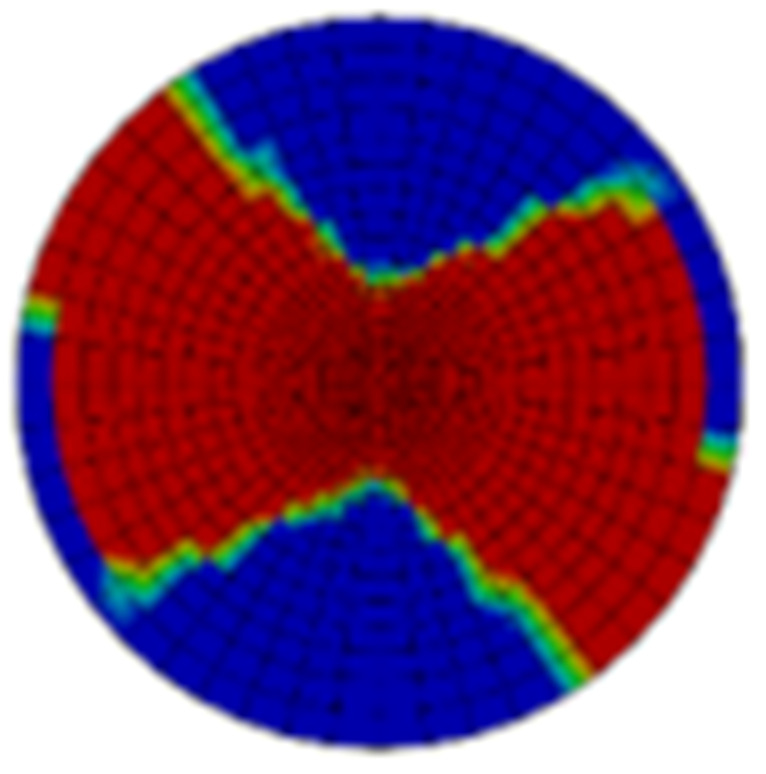	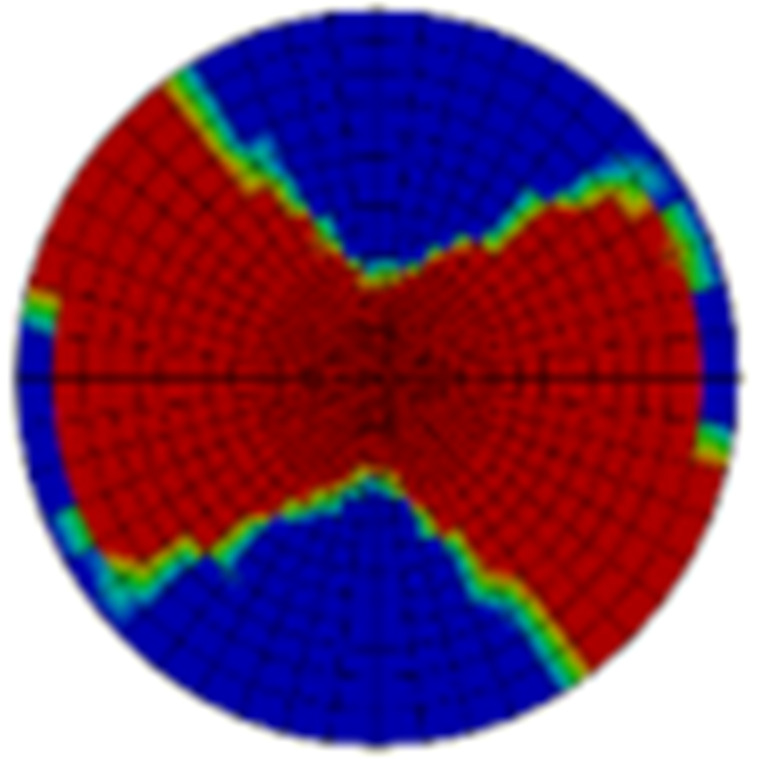	19	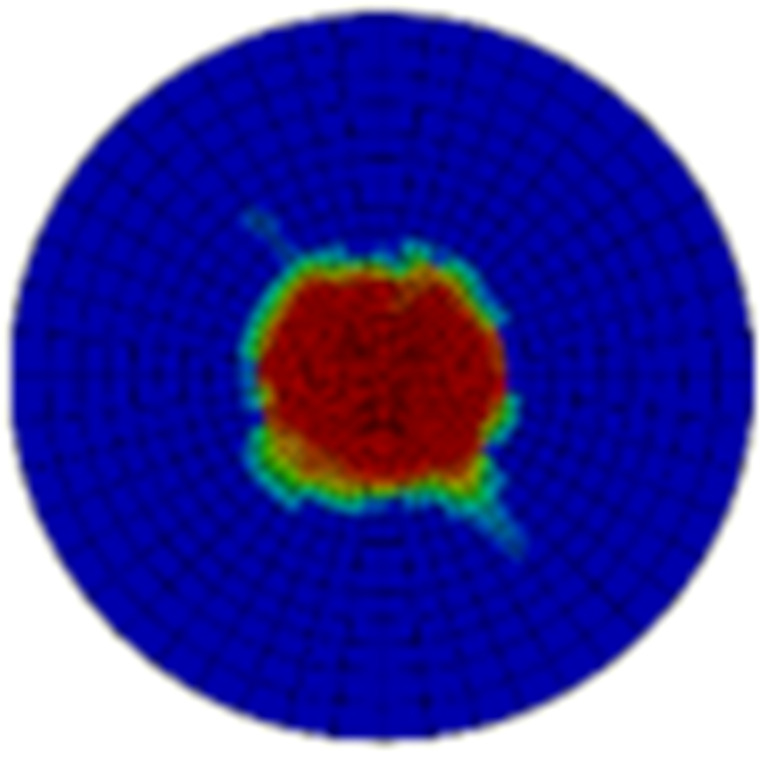	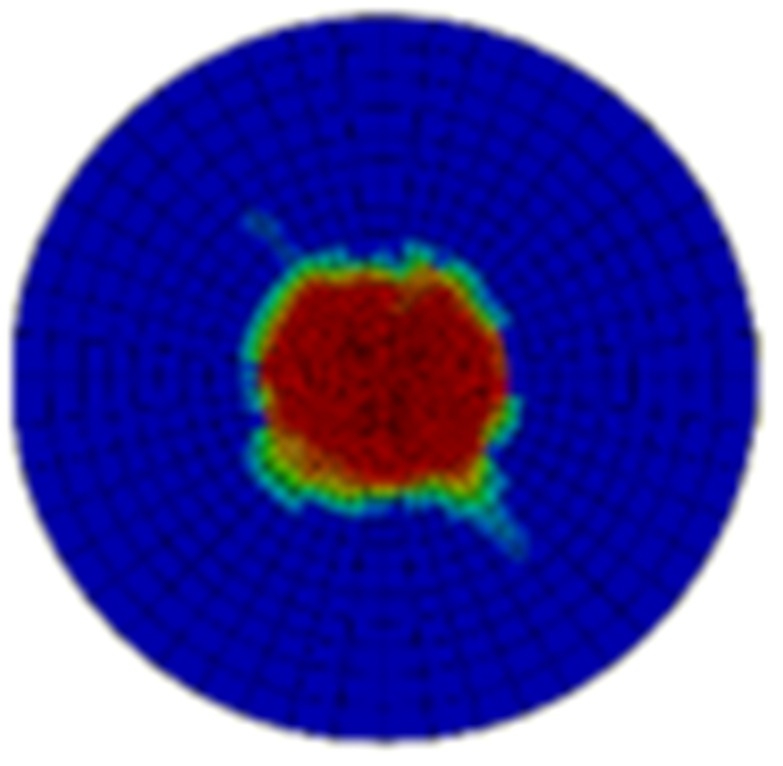
8	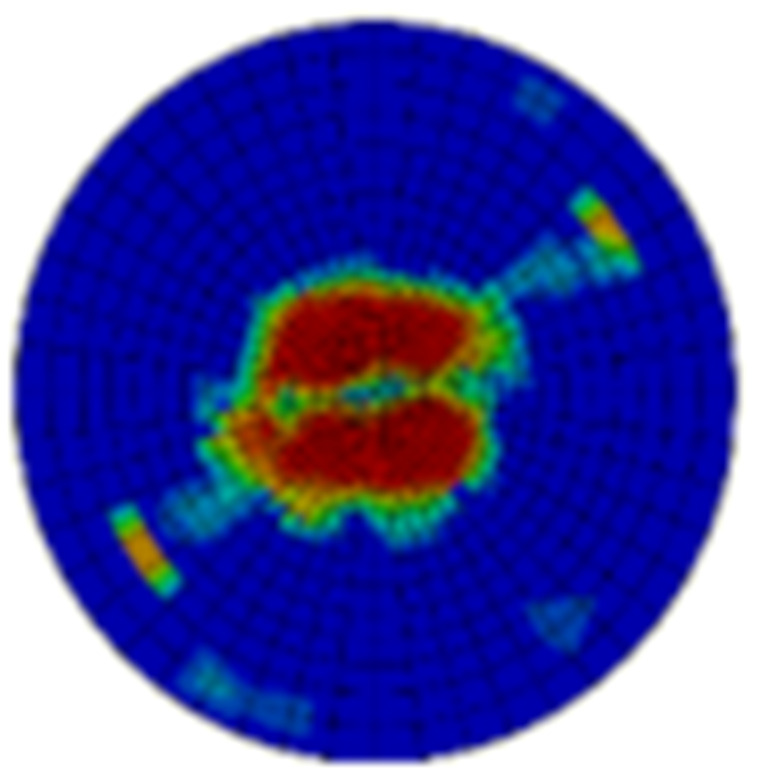	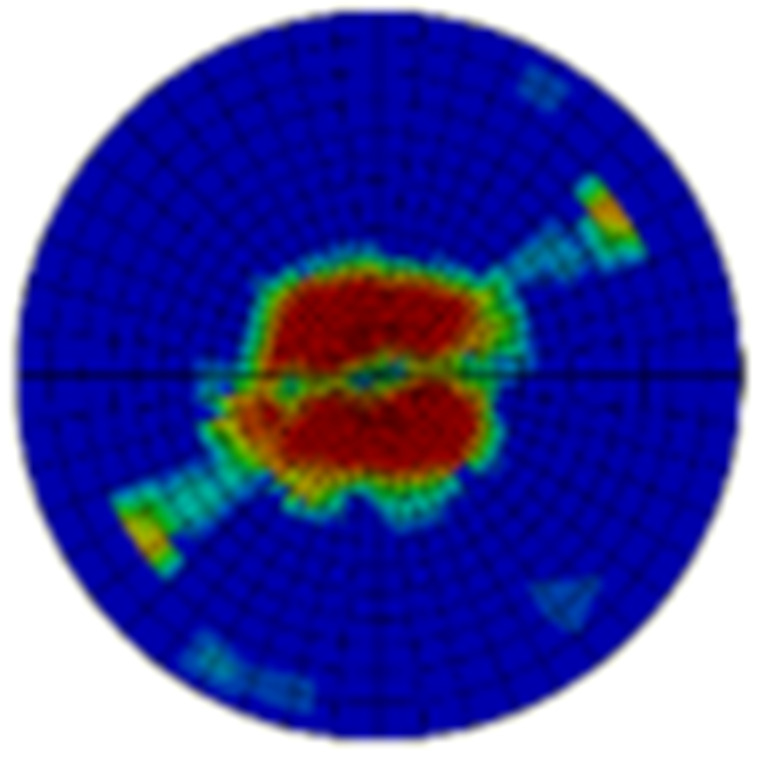	20	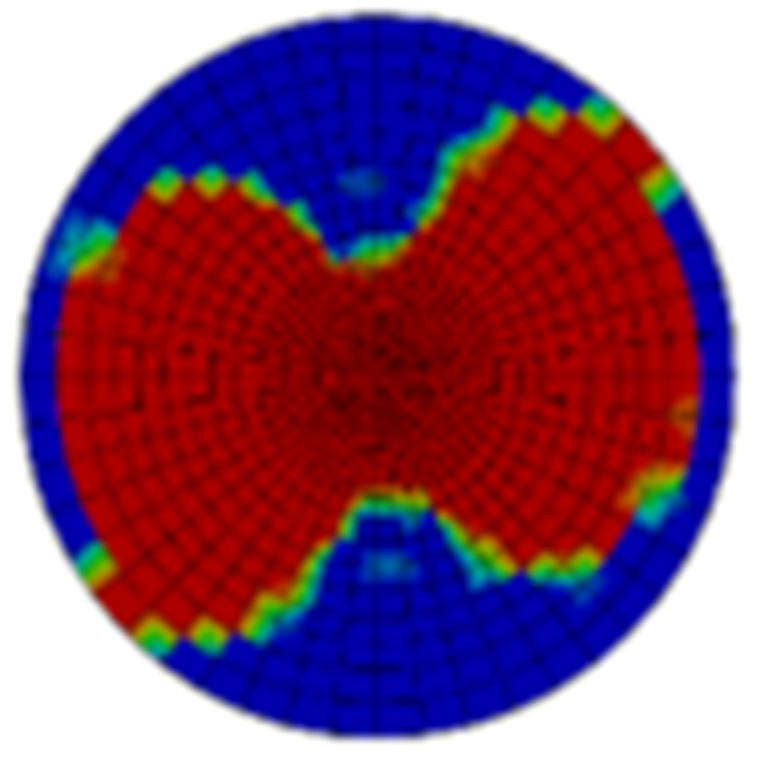	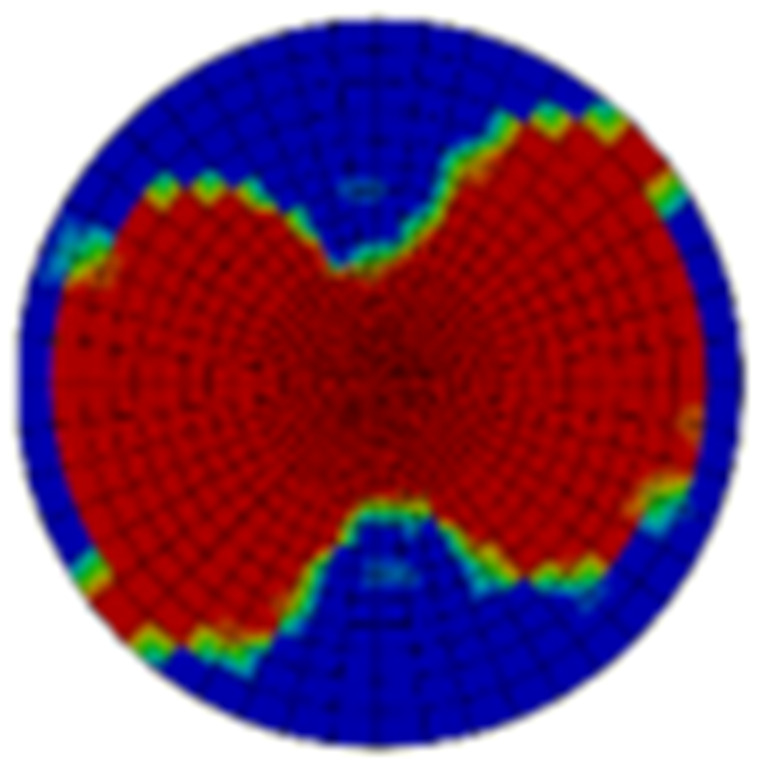
9	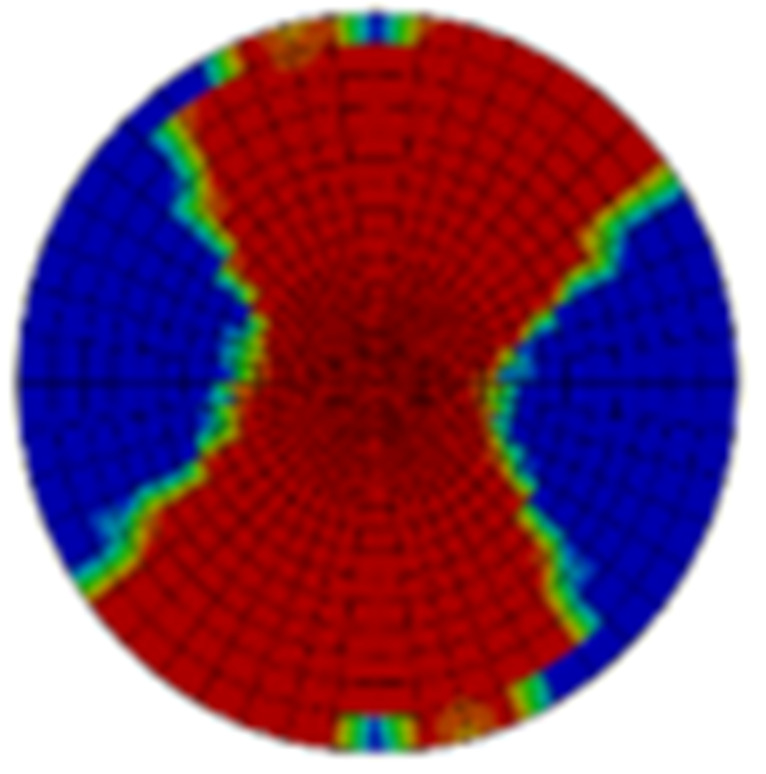	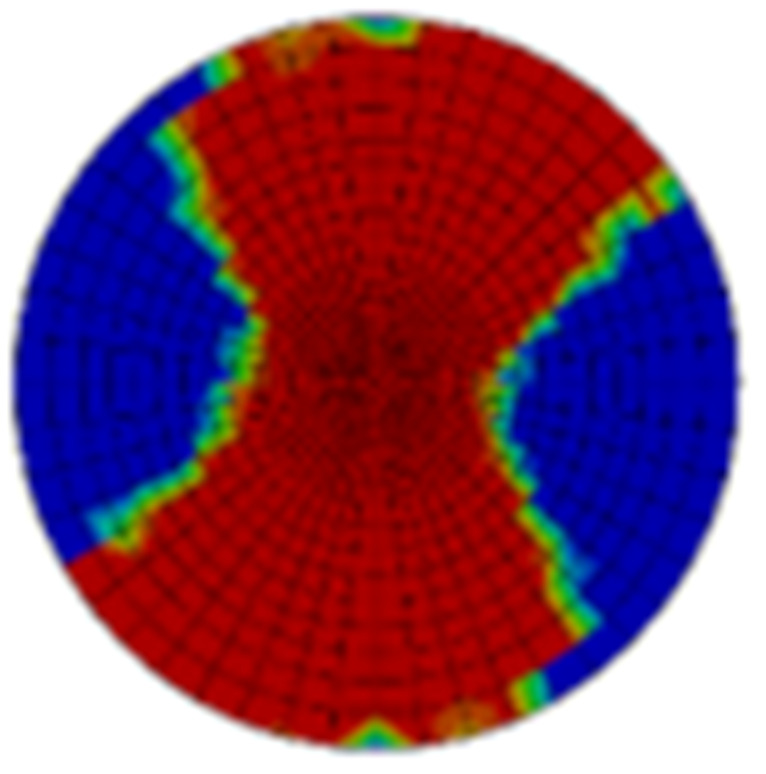	21	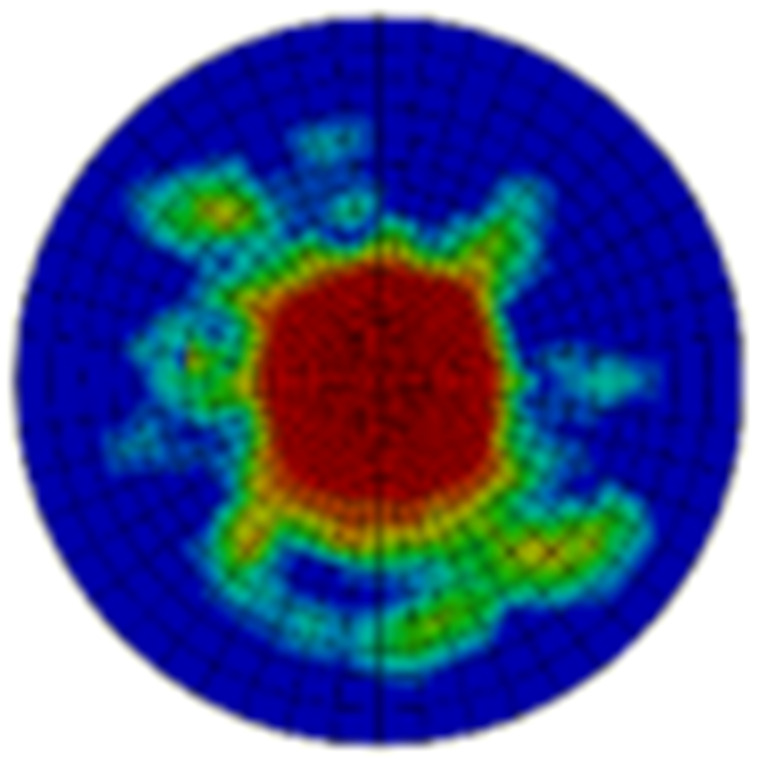	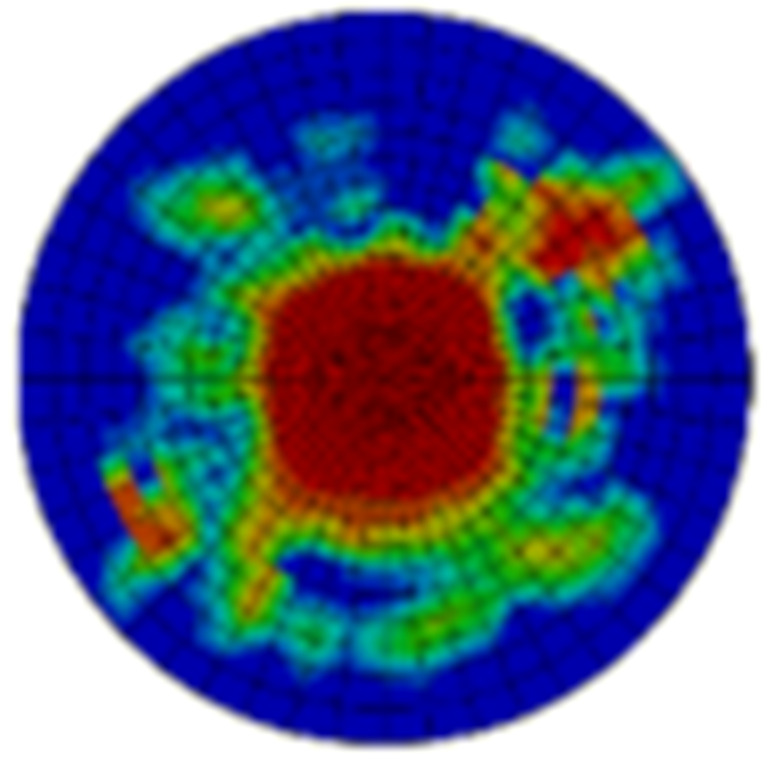
10	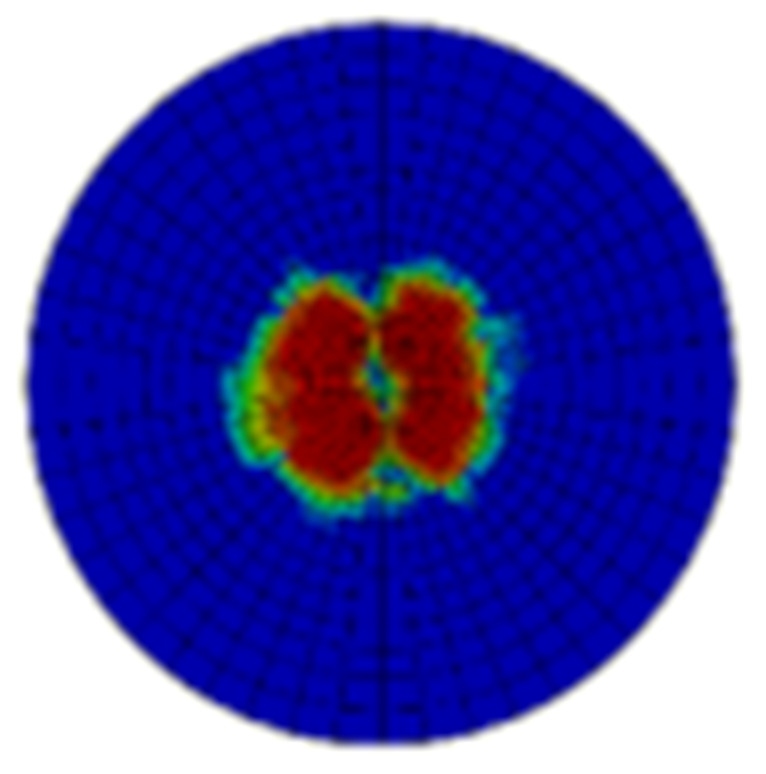	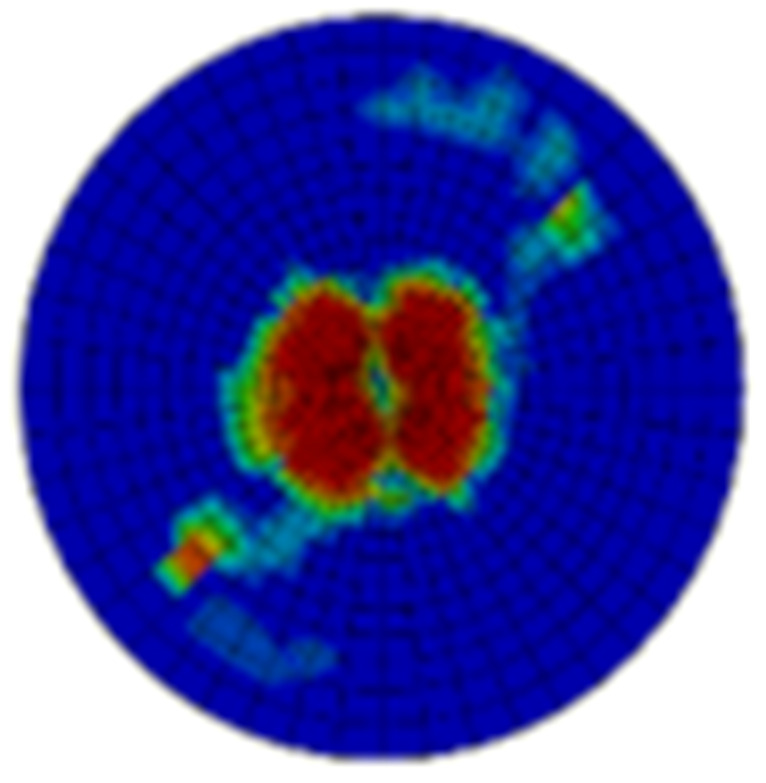	22	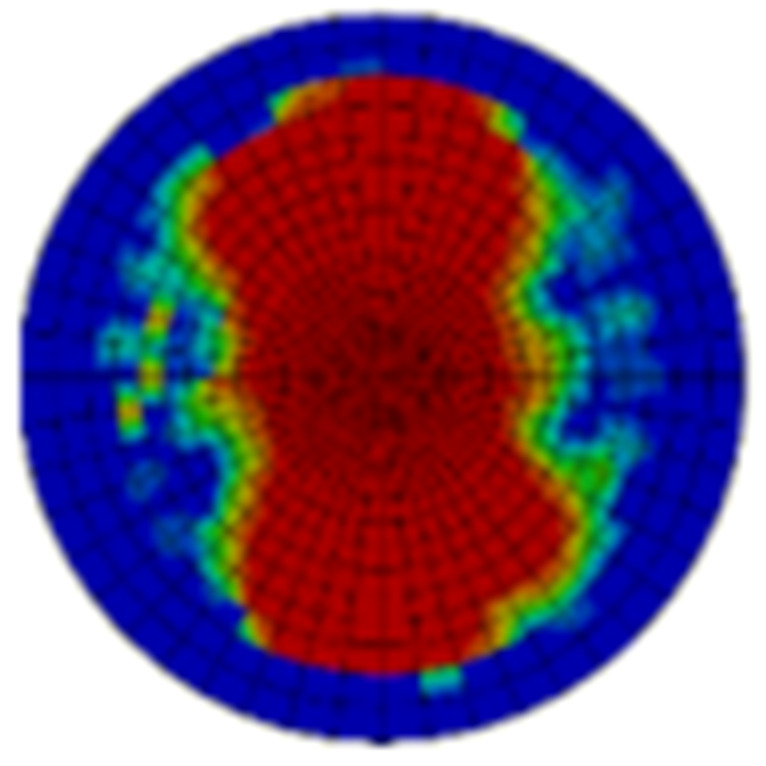	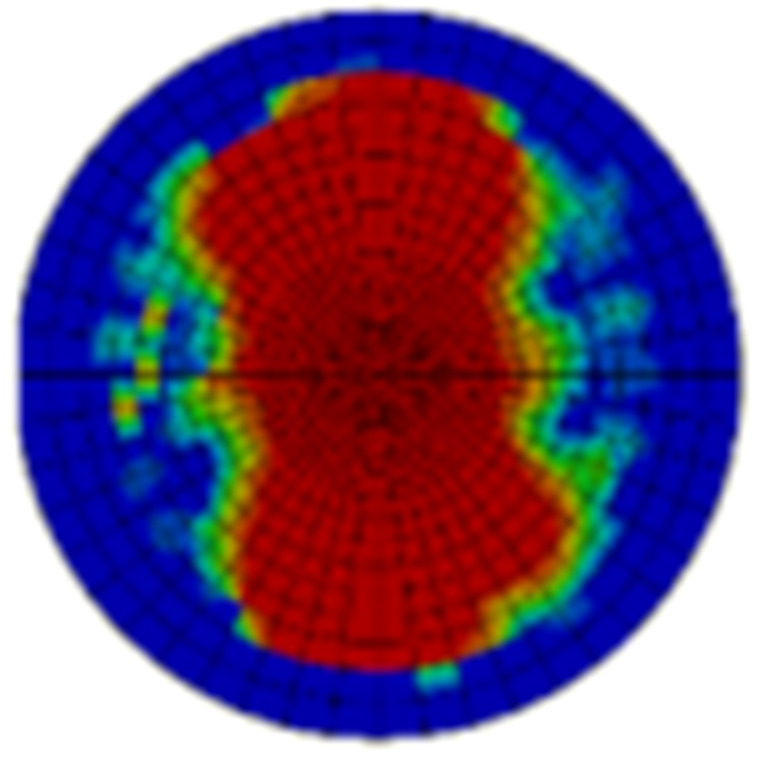
11	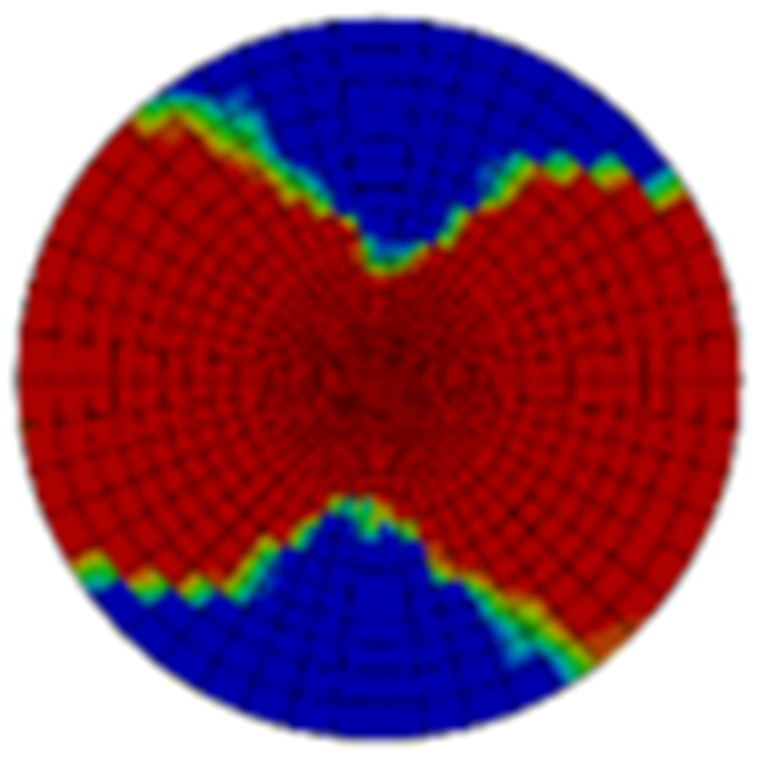	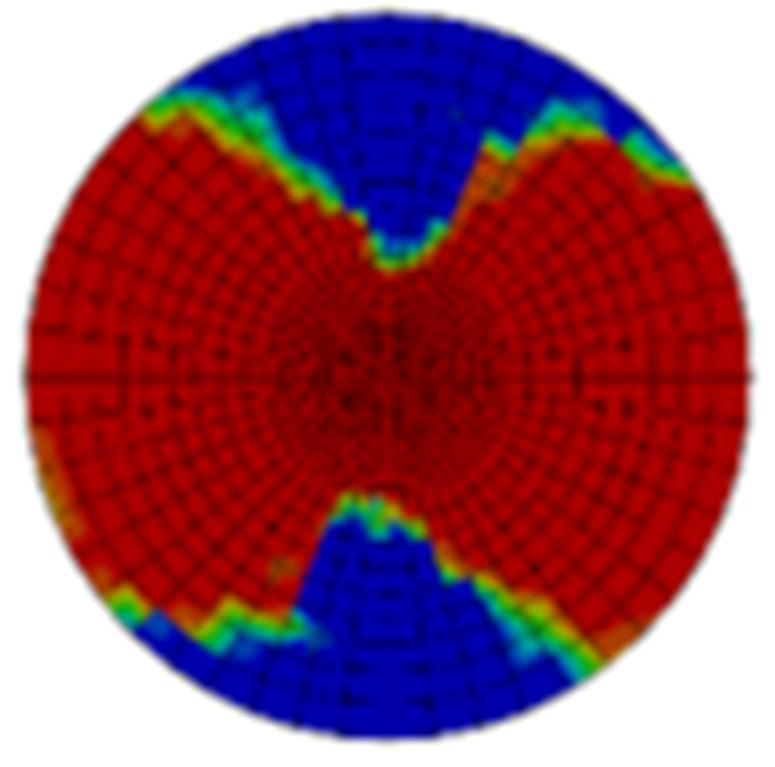	23	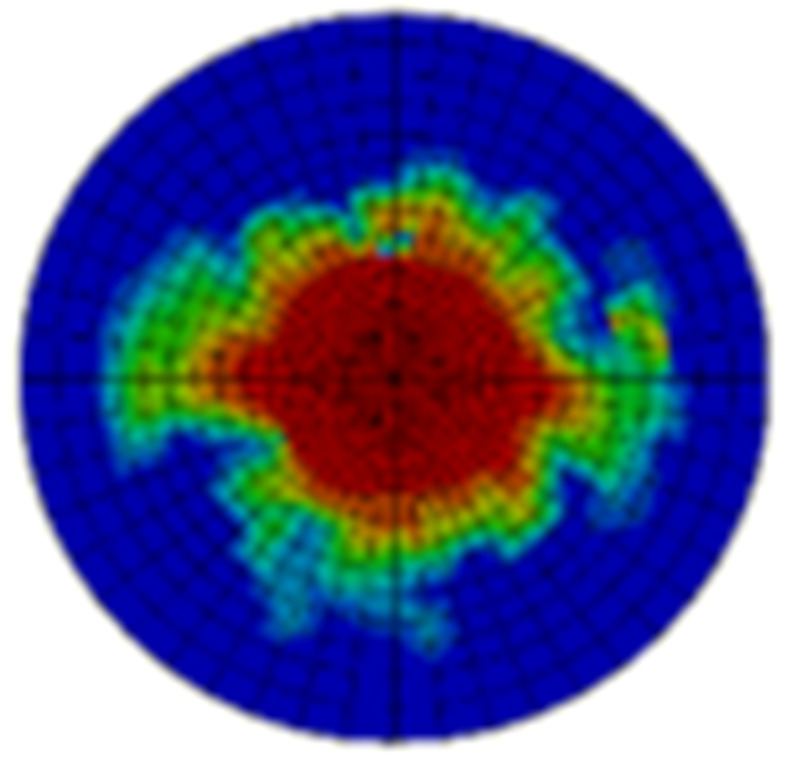	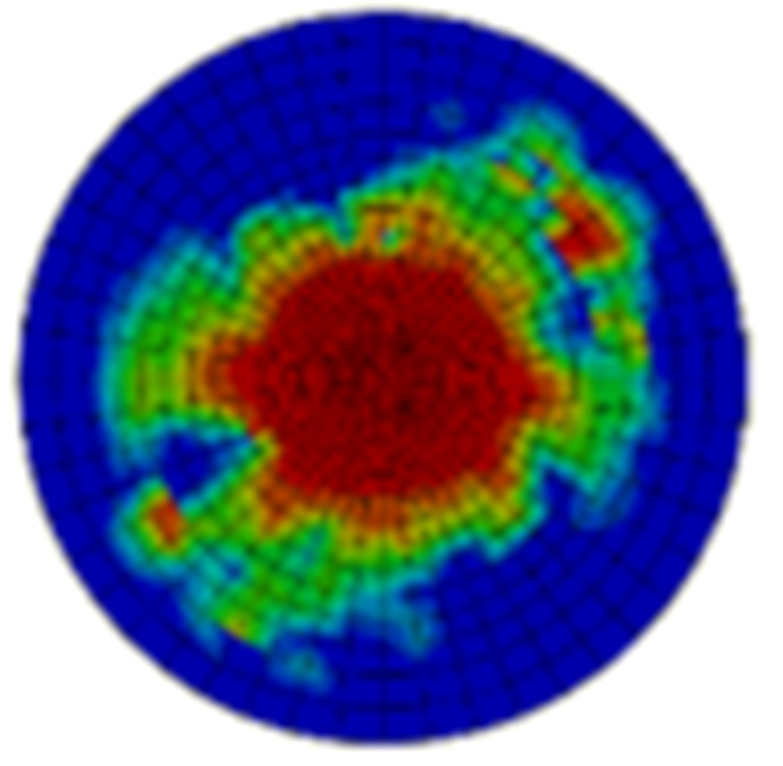
12	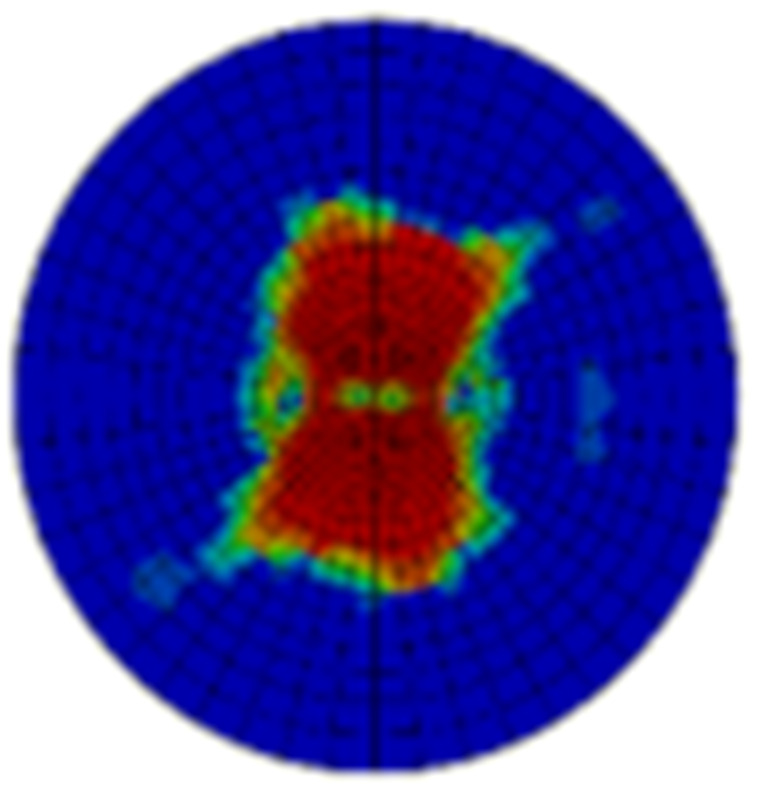	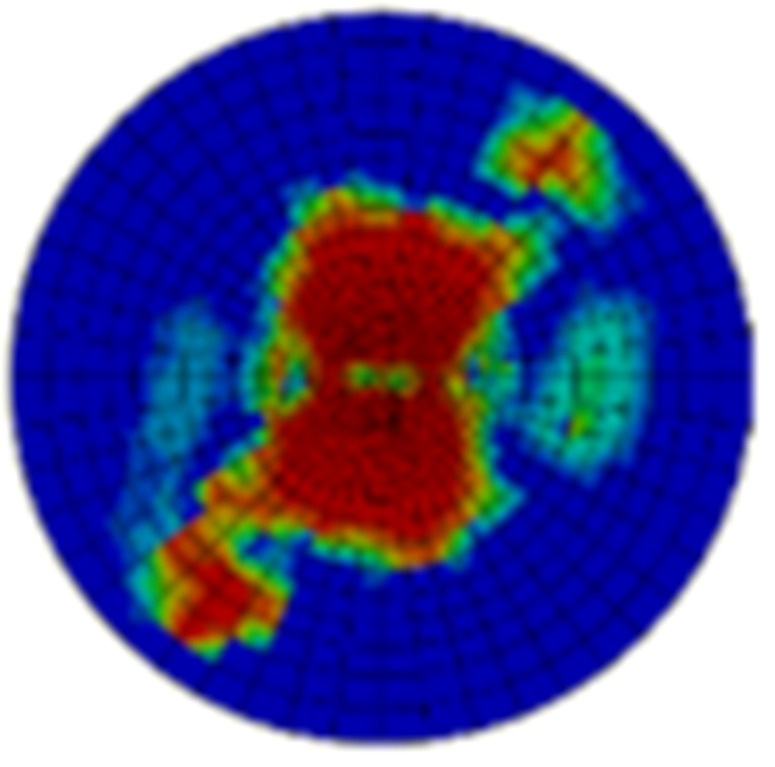	-		

**Table 7 materials-15-05029-t007:** Matrix failure status of the composite laminate with an impact energy of 20 J.

Layer No.	Impacted Results	Shear Results	Layer No.	Impacted Results	Shear Results
1	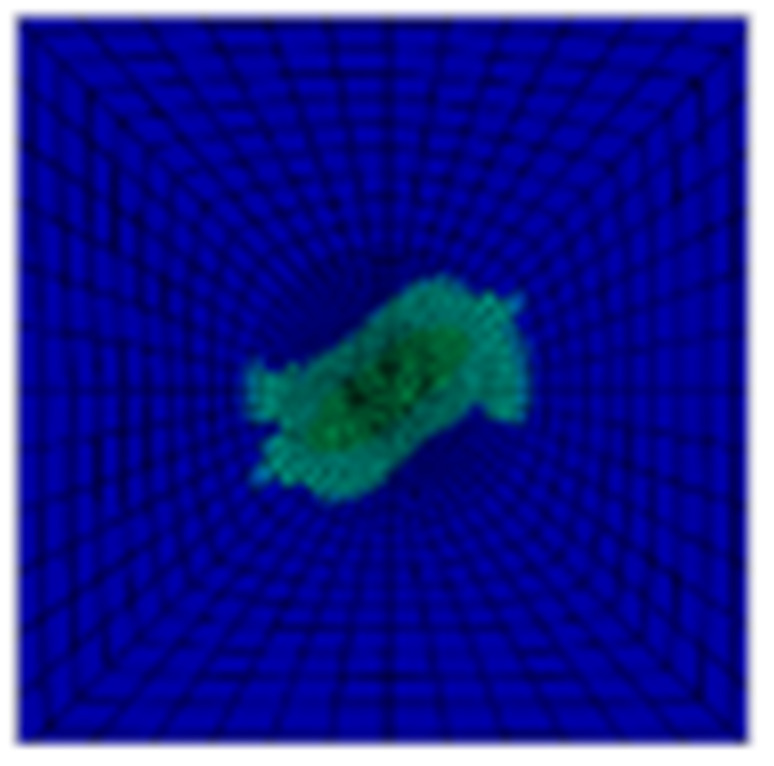	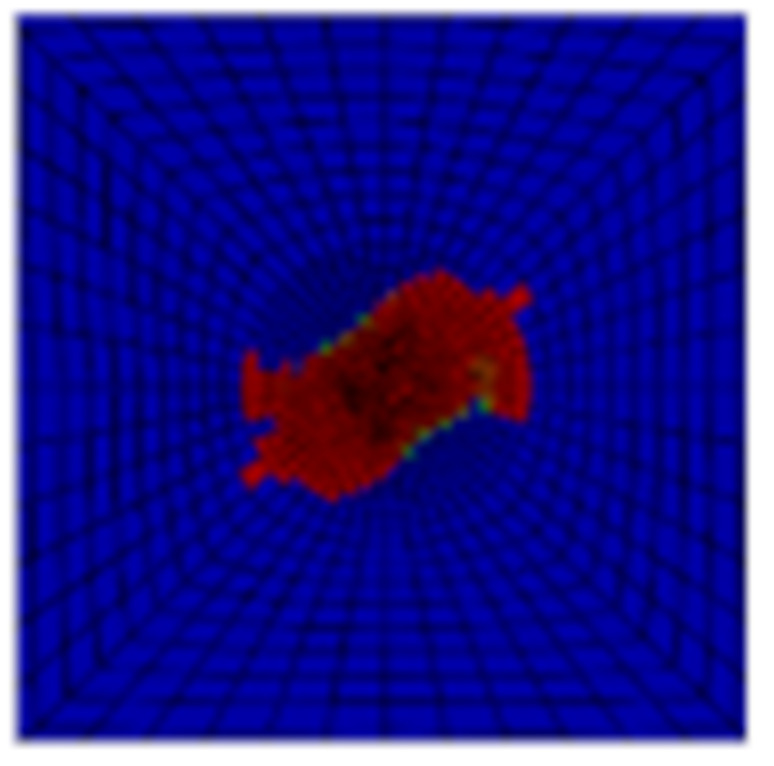	13	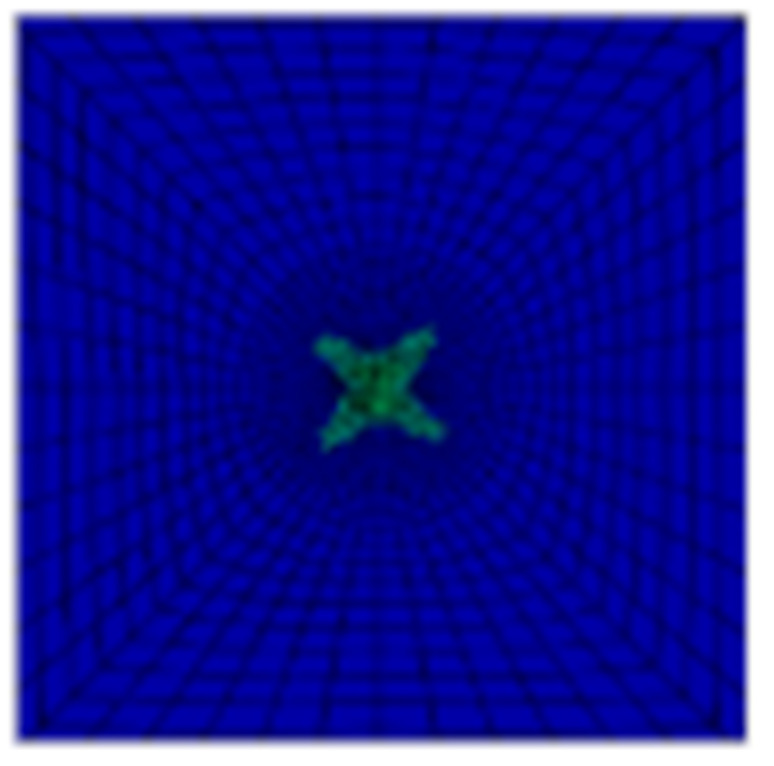	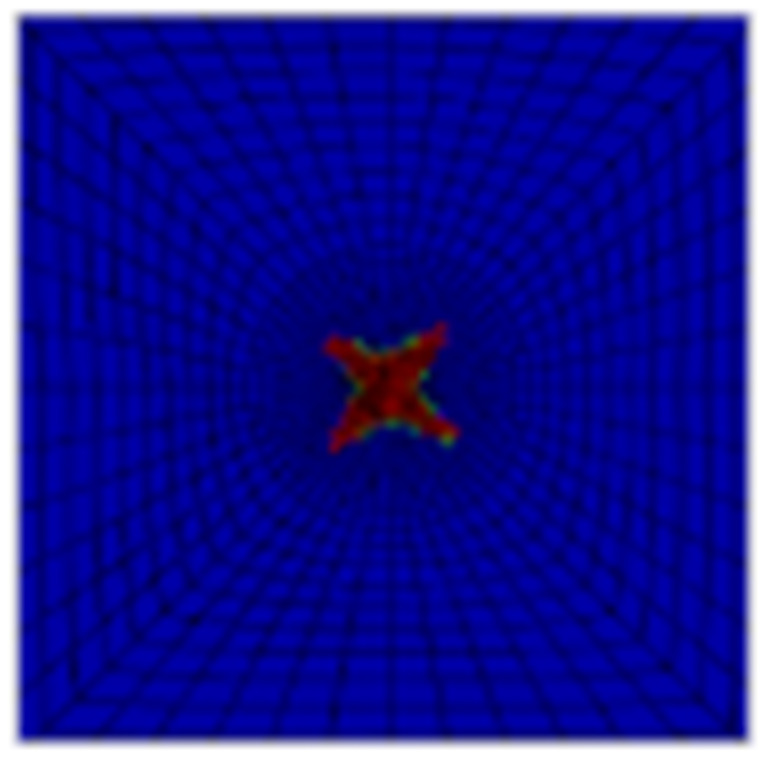
2	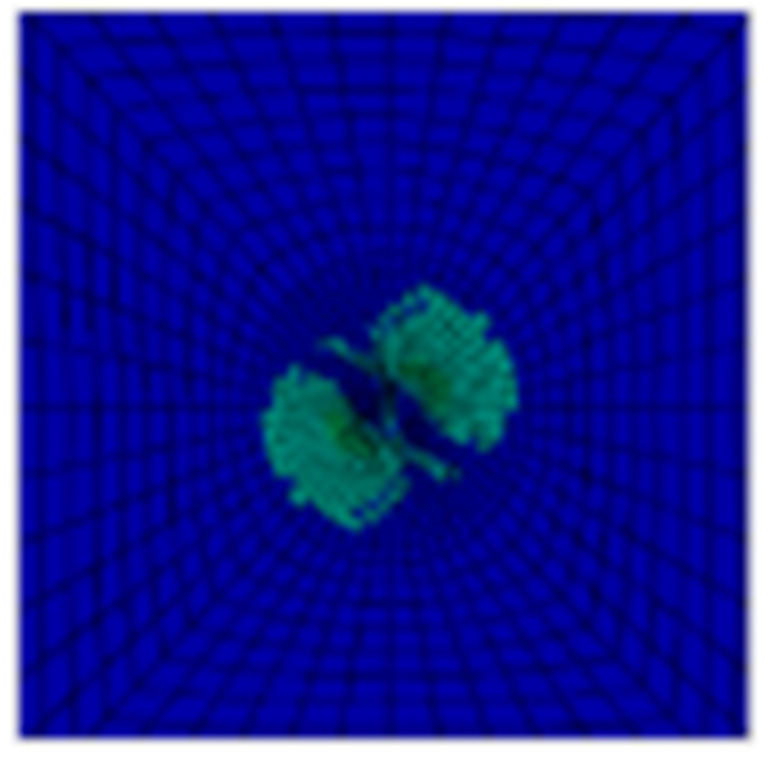	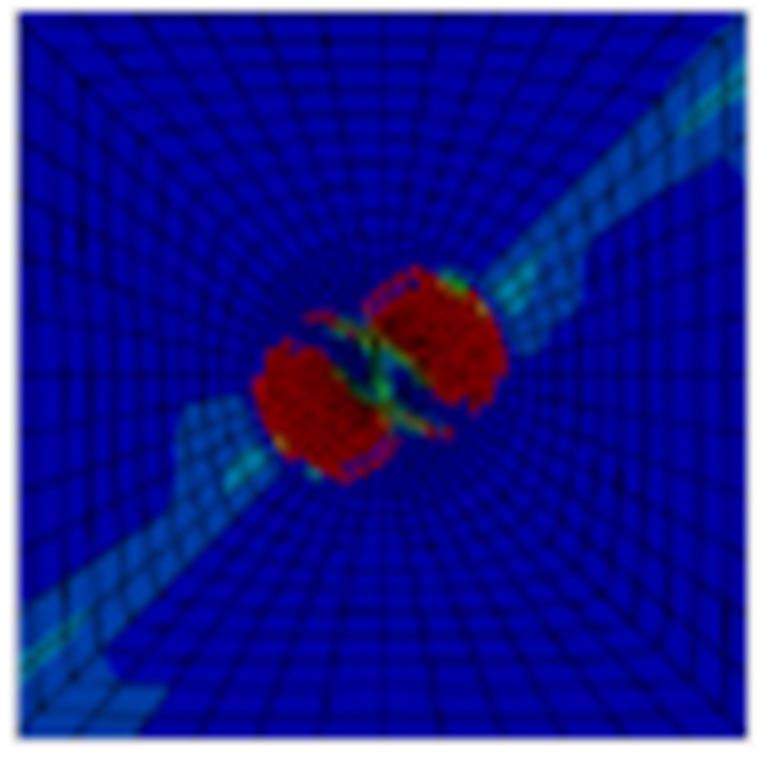	14	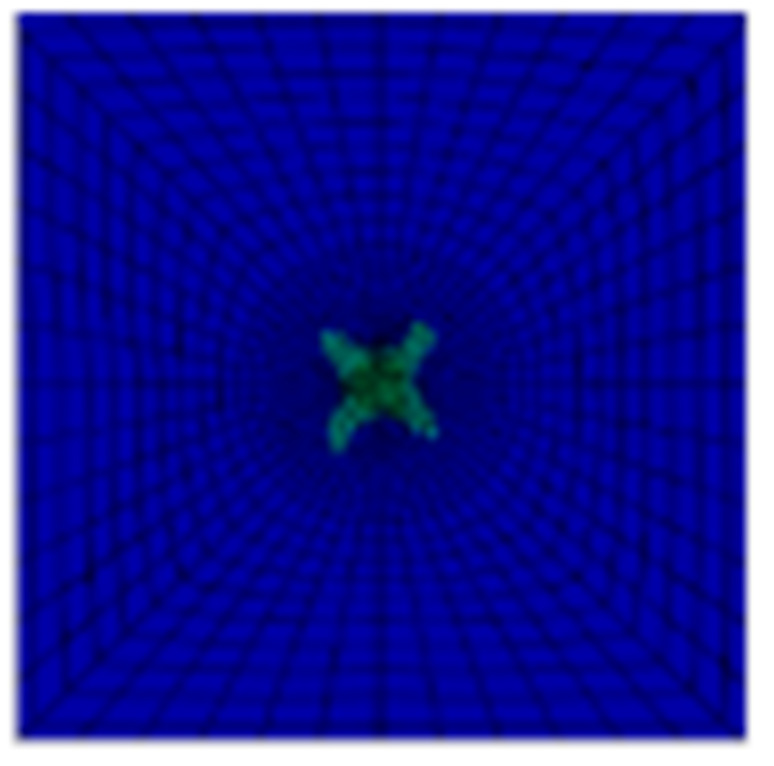	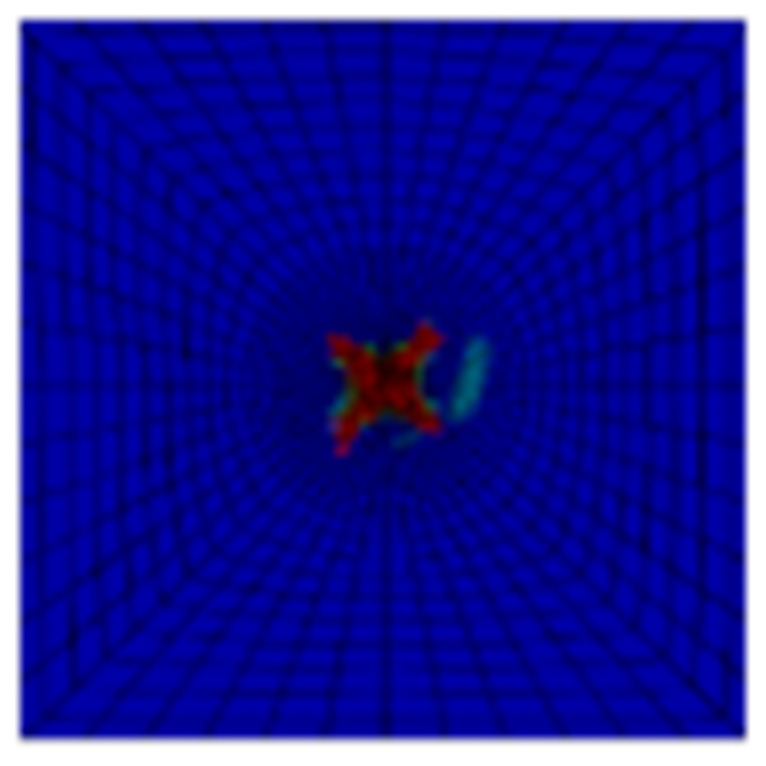
3	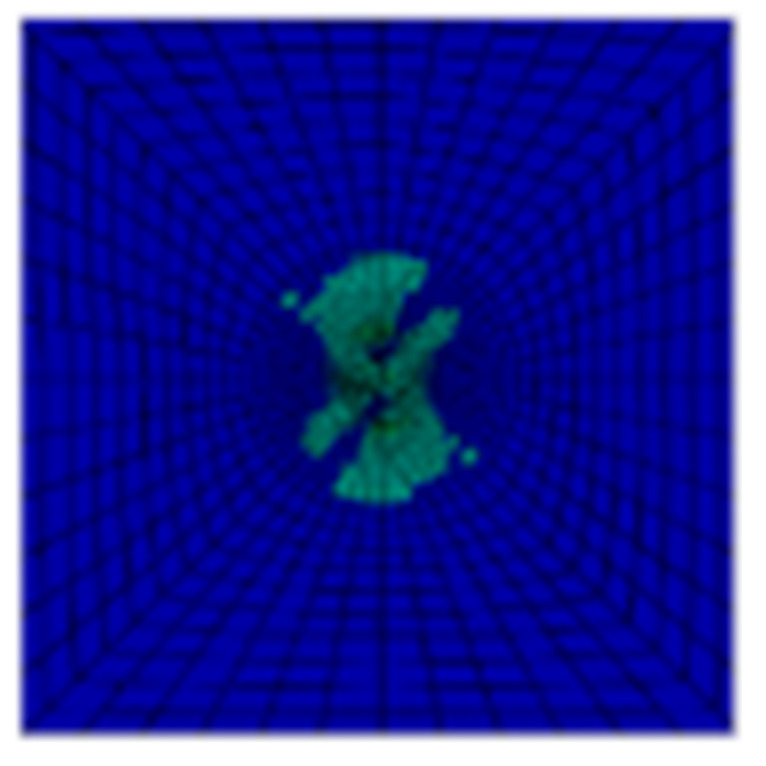	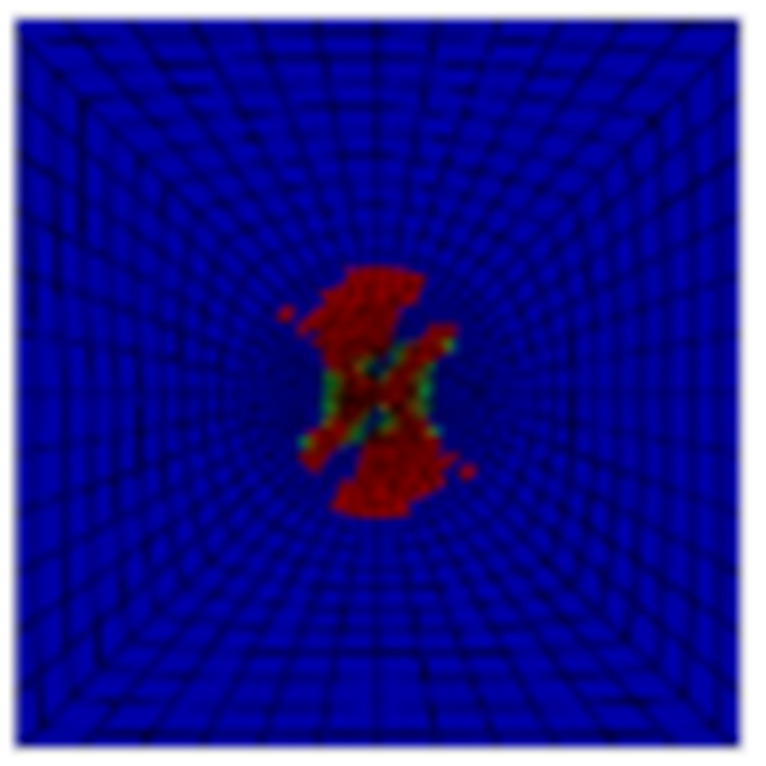	15	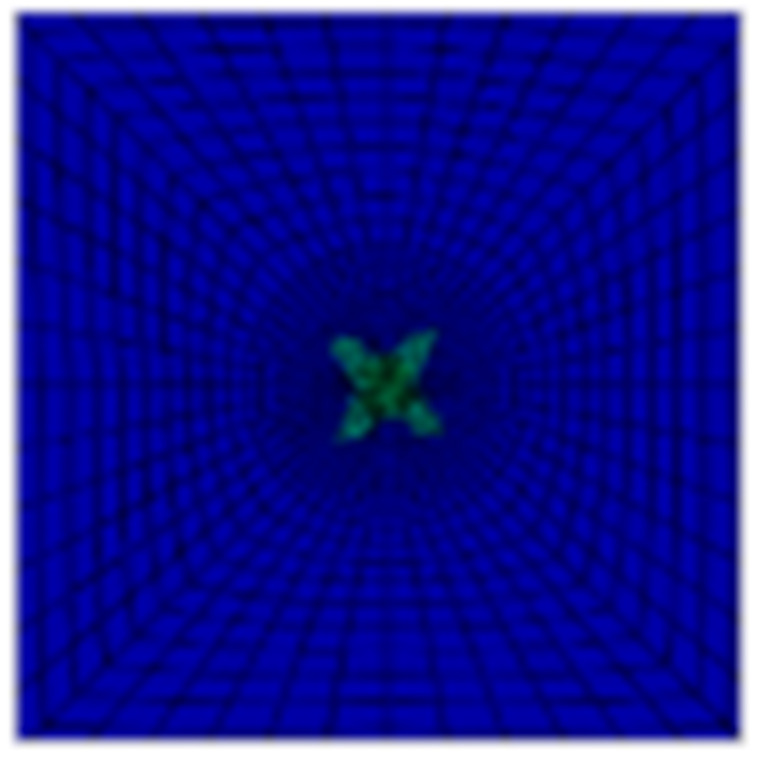	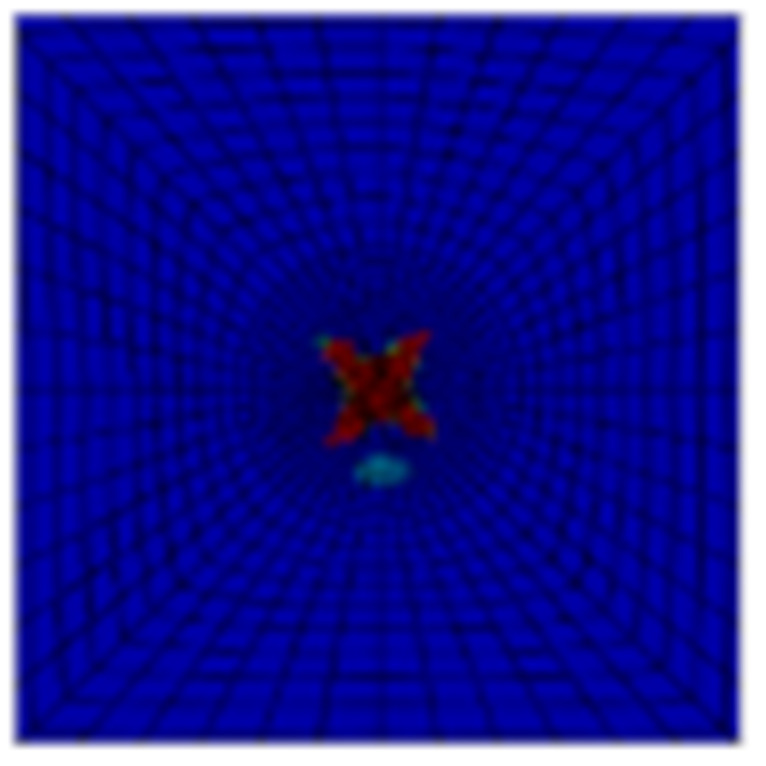
4	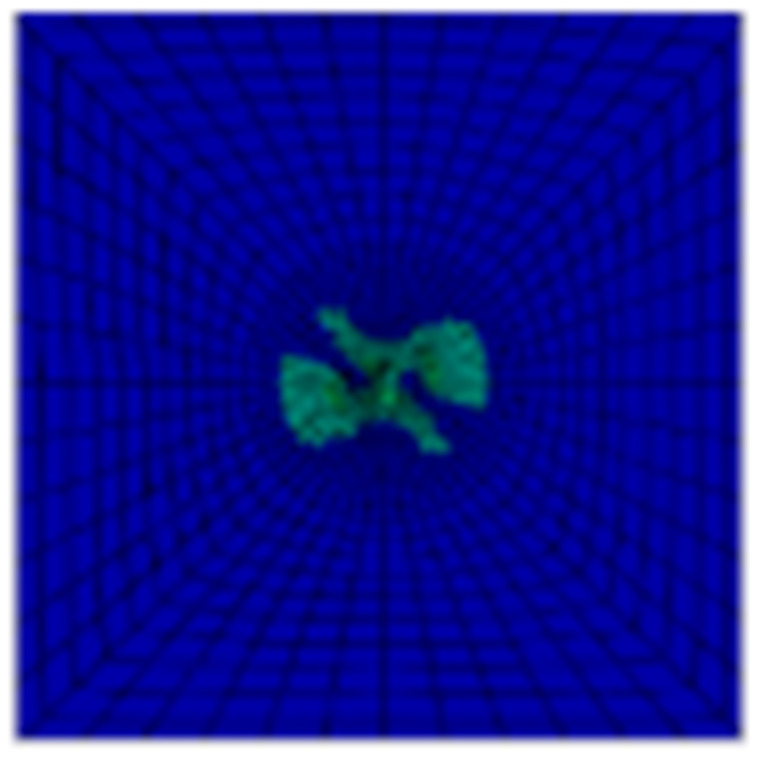	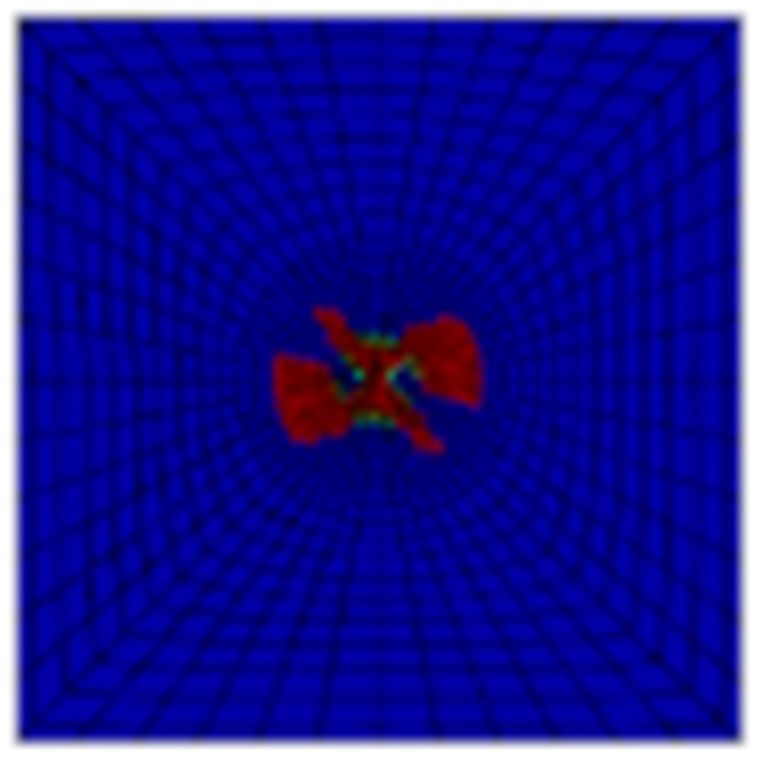	16	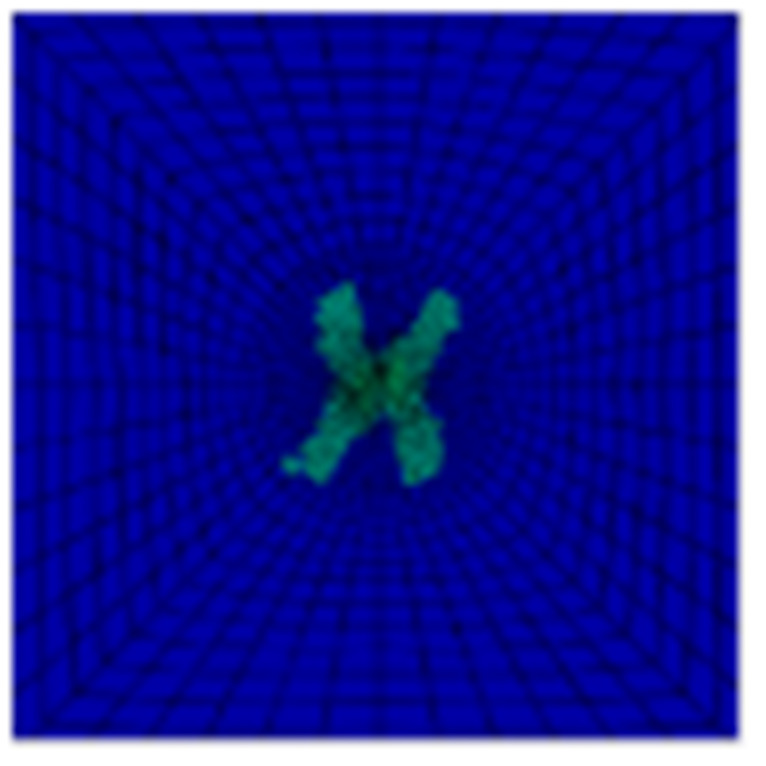	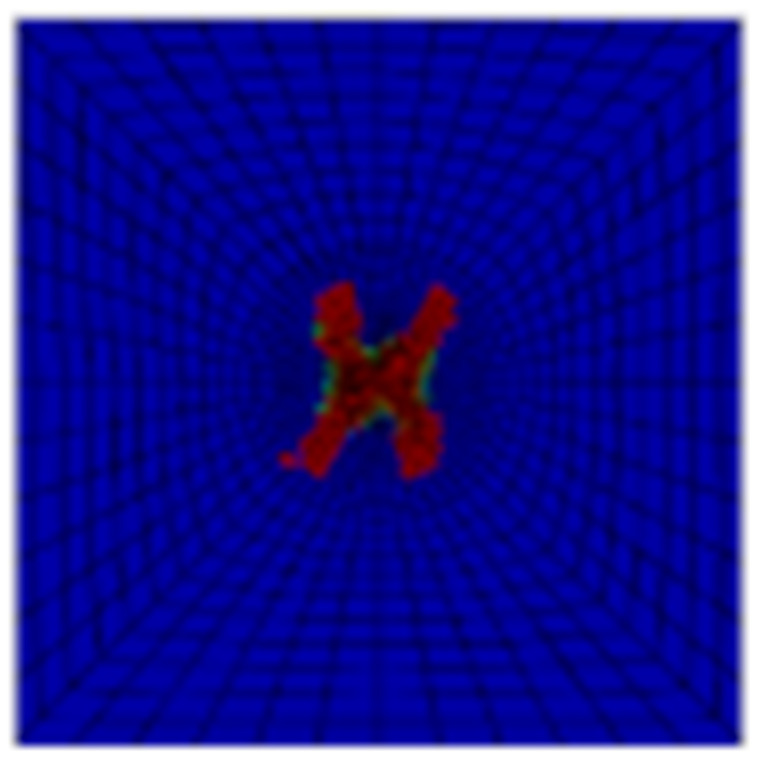
5	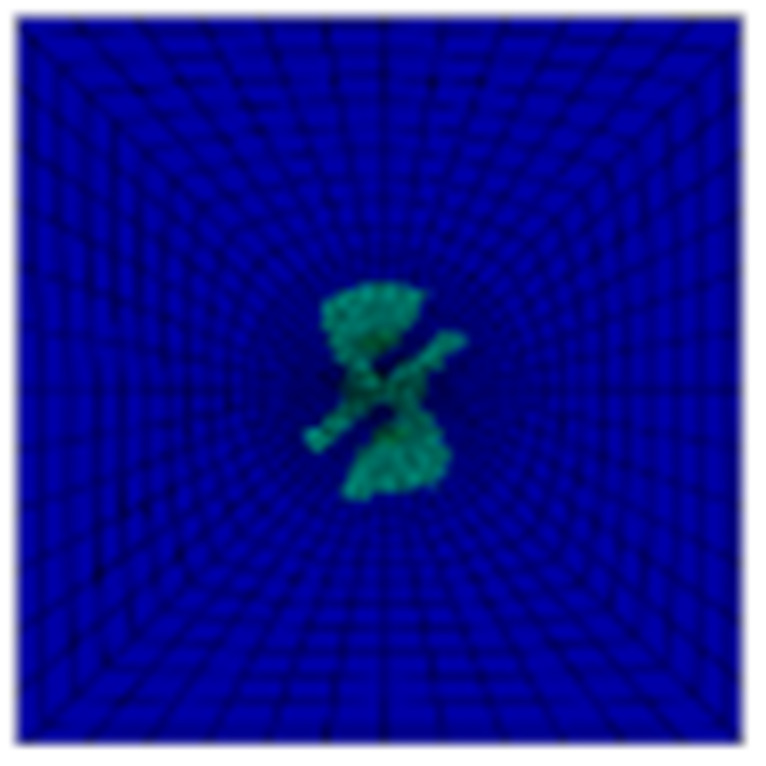	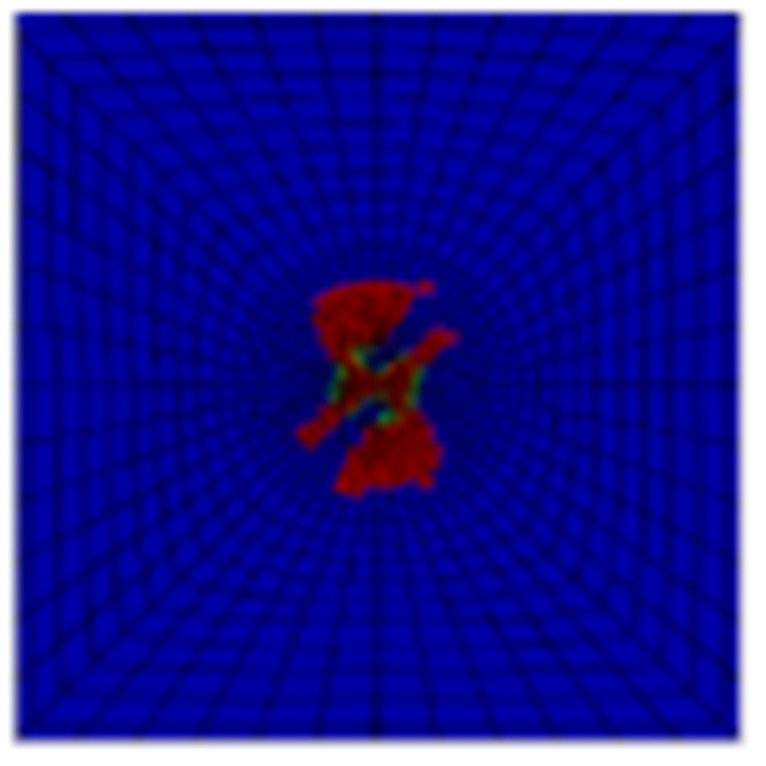	17	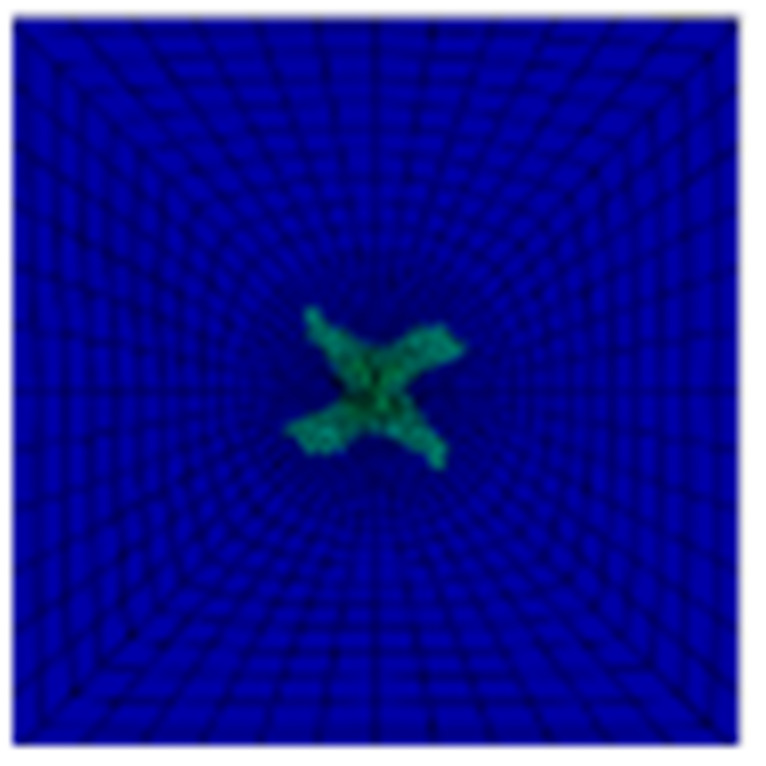	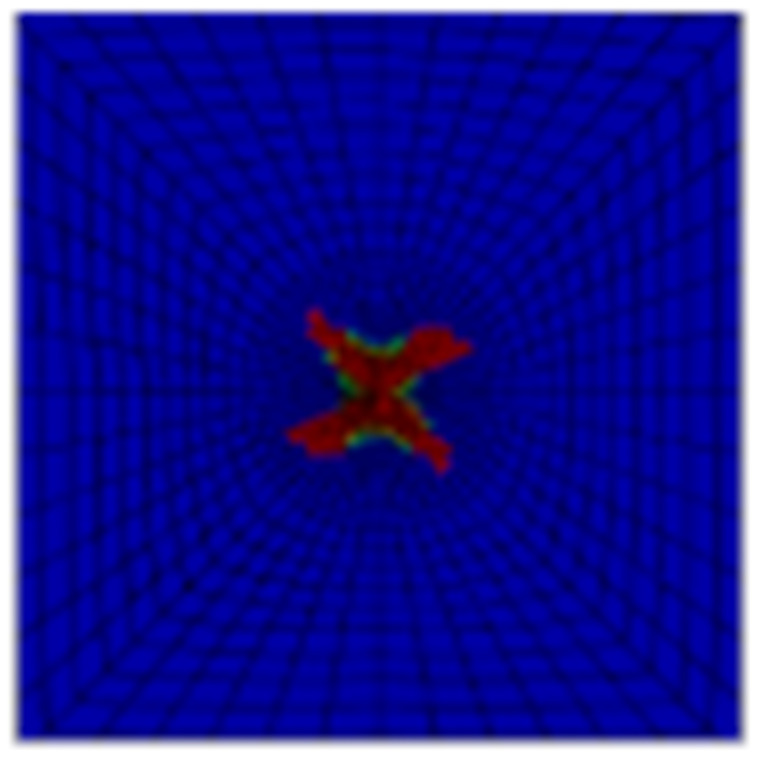
6	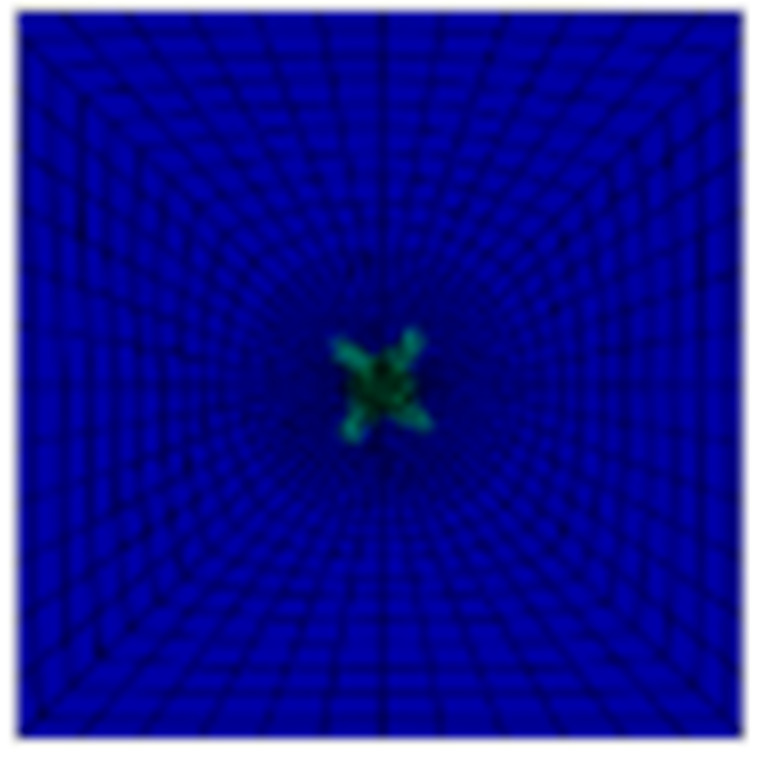	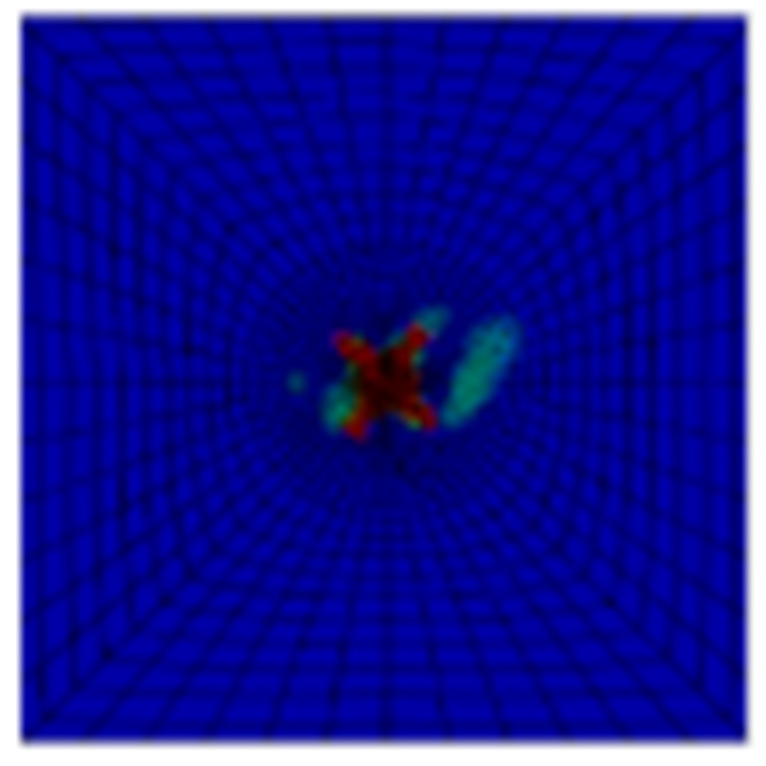	18	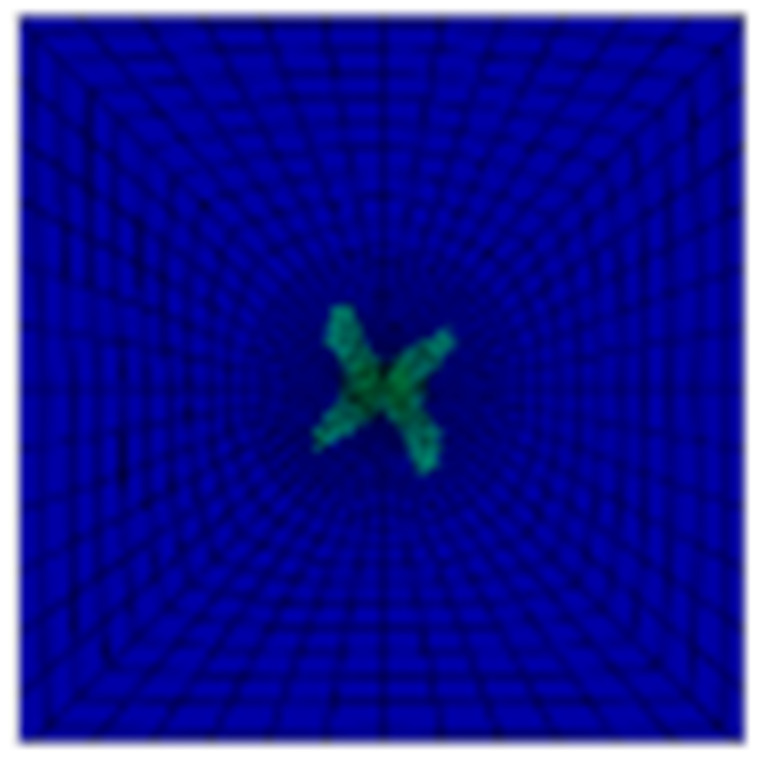	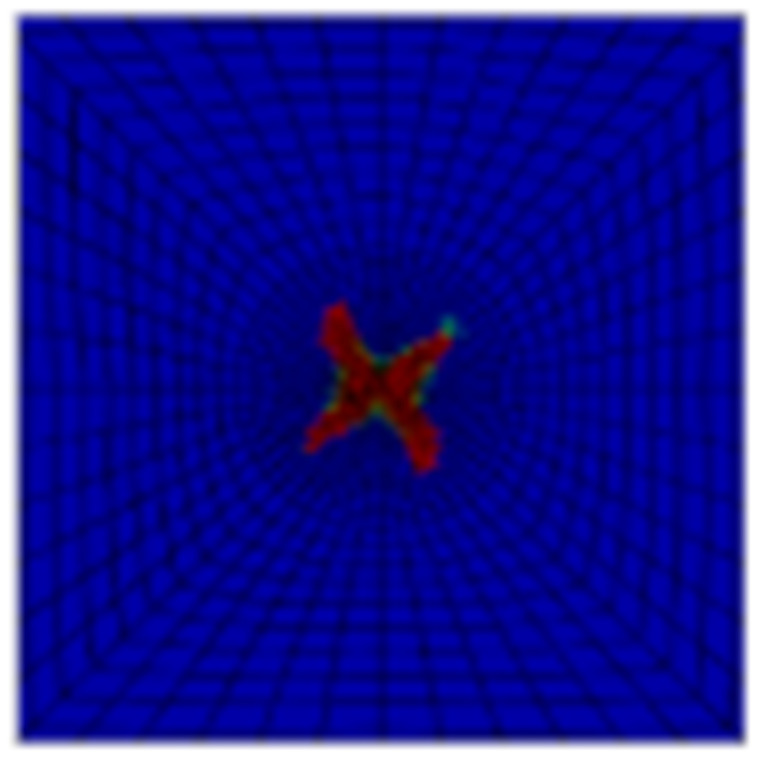
7	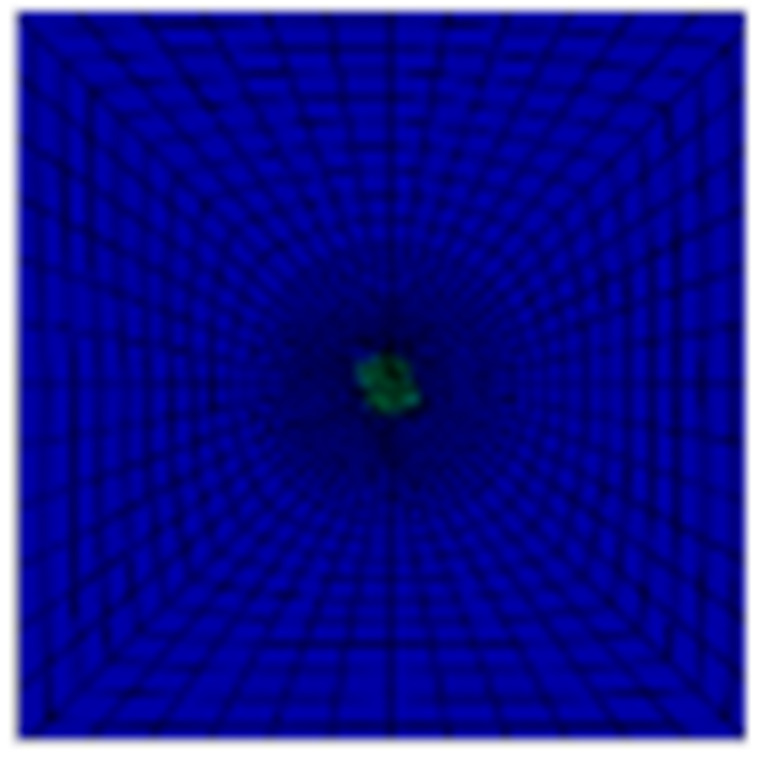	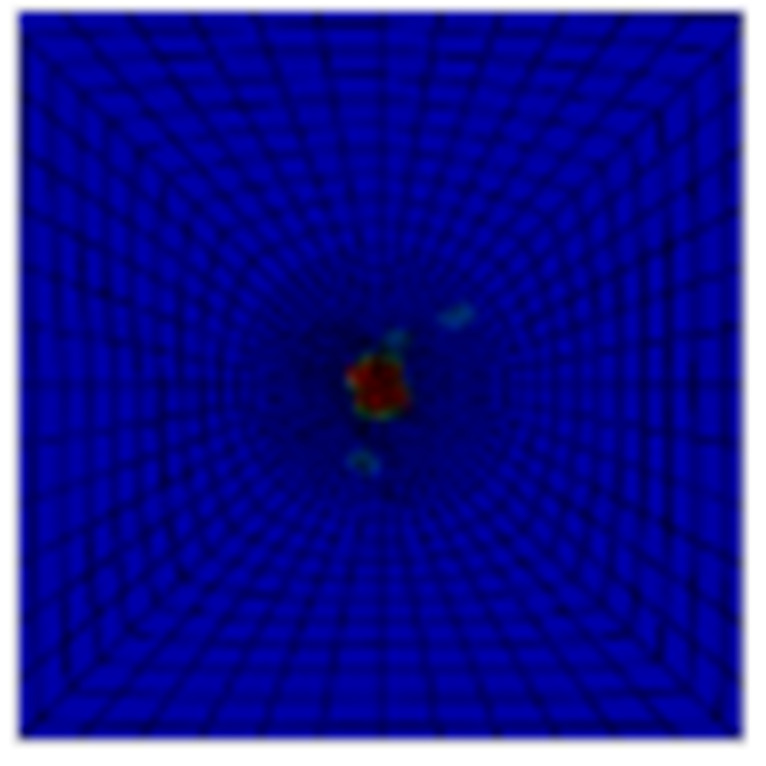	19	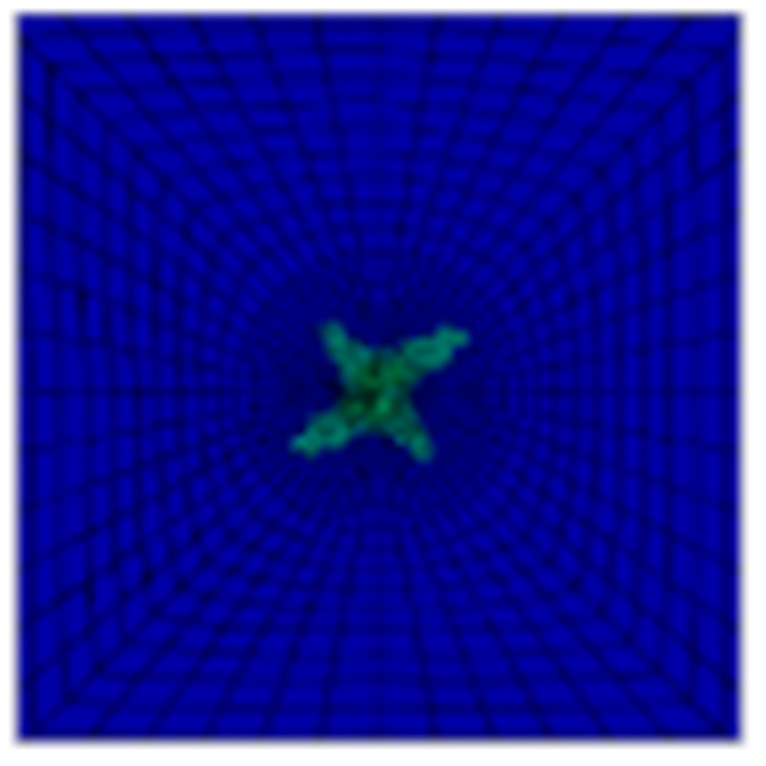	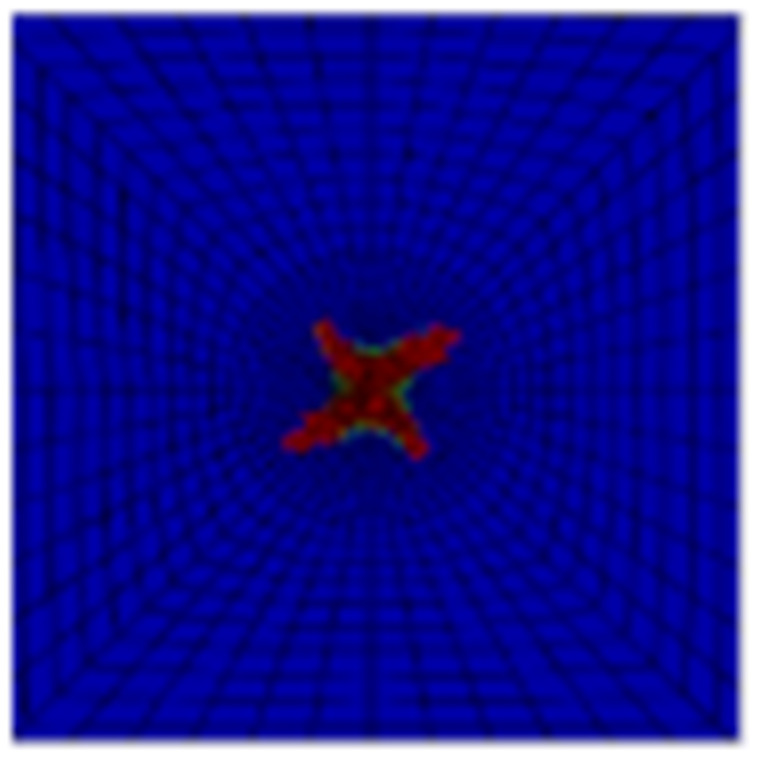
8	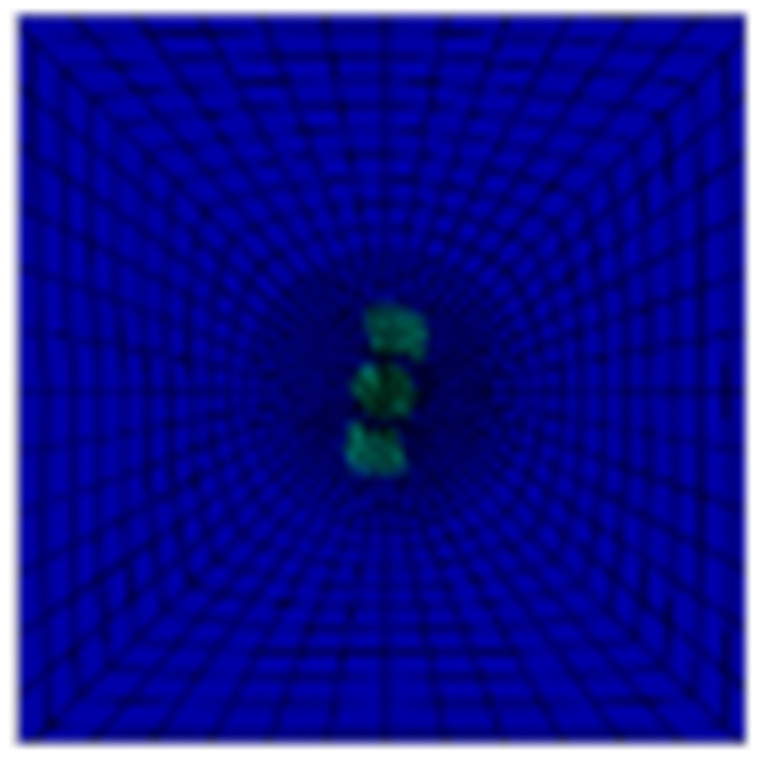	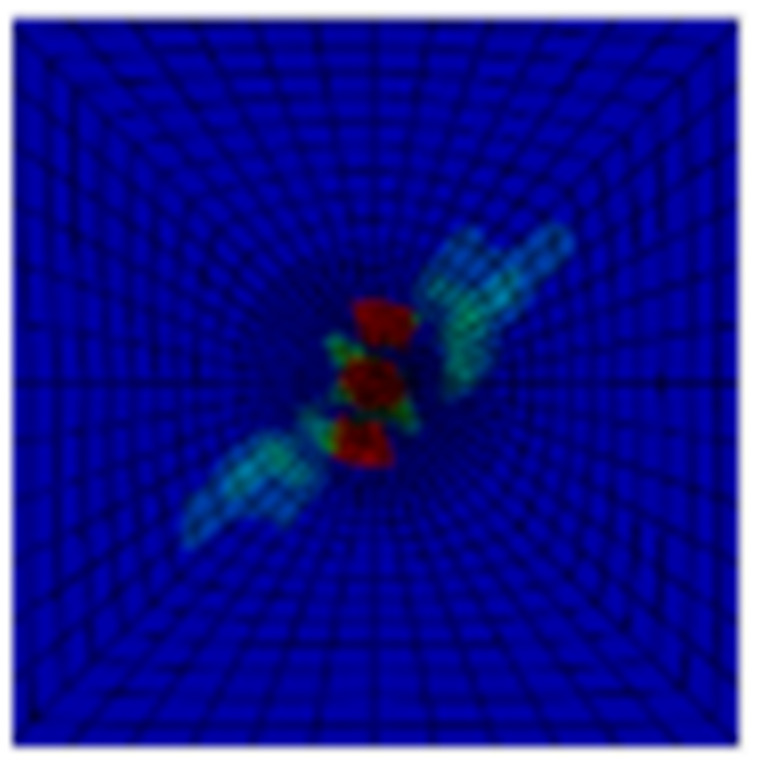	20	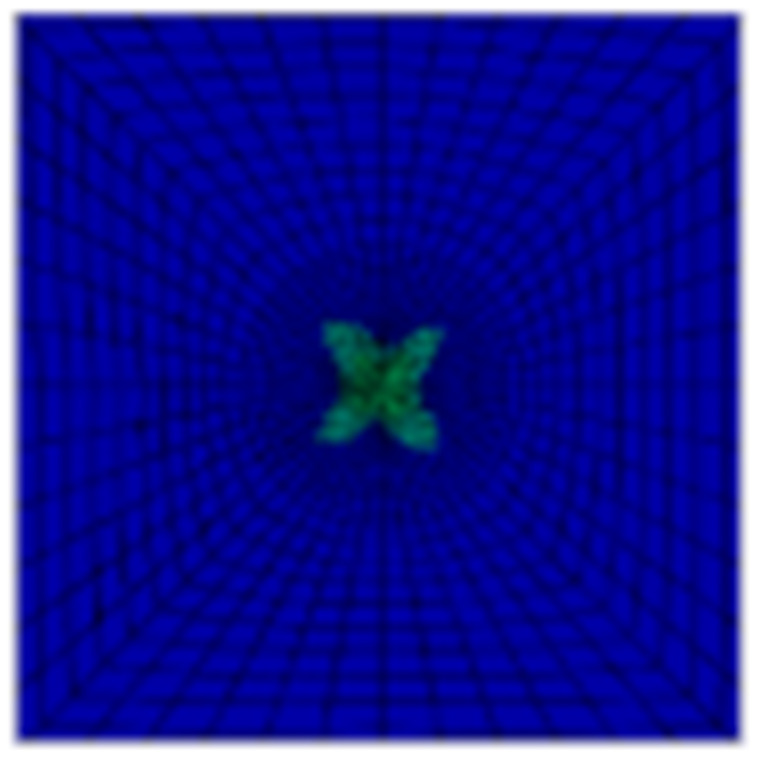	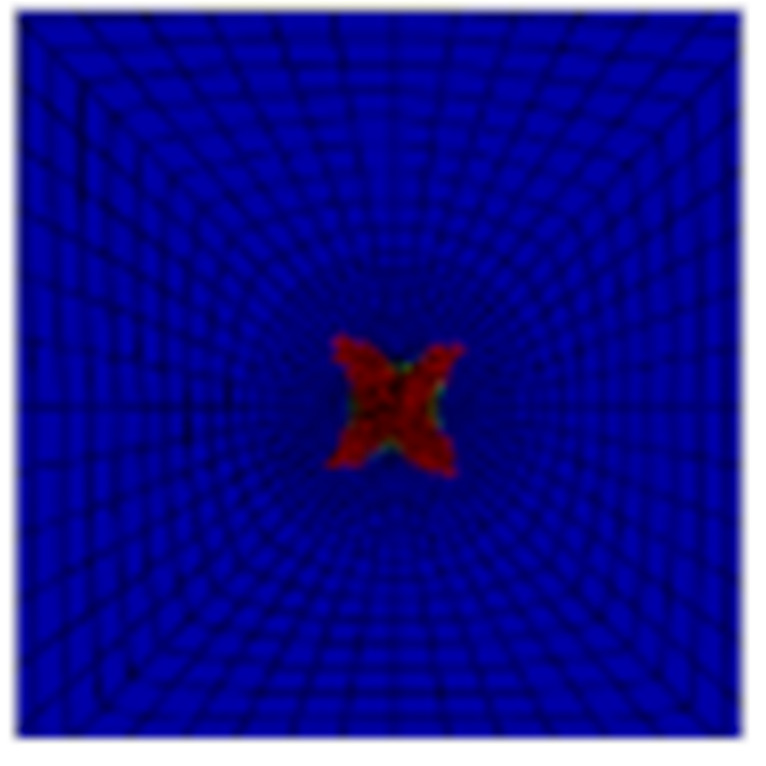
9	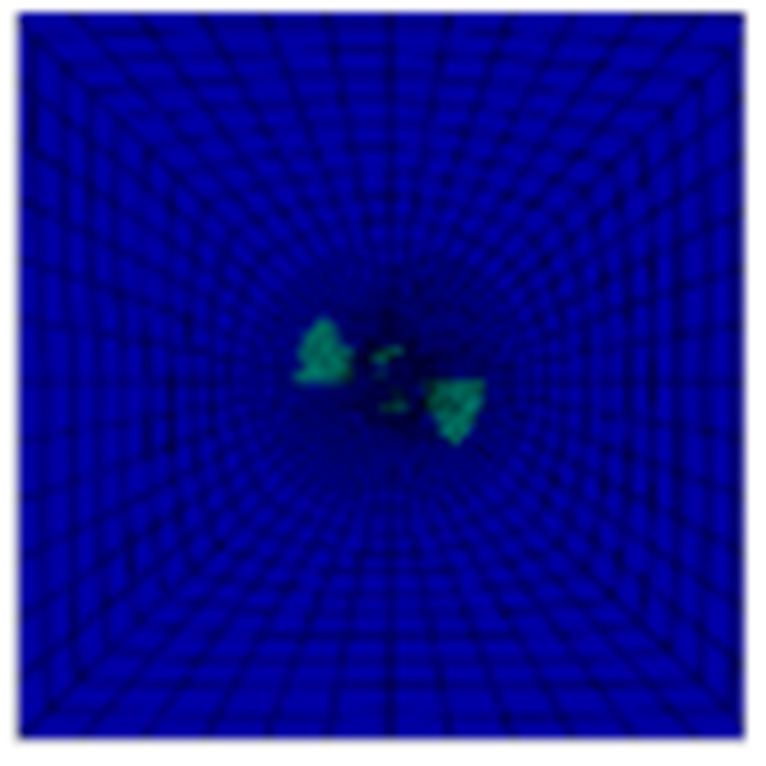	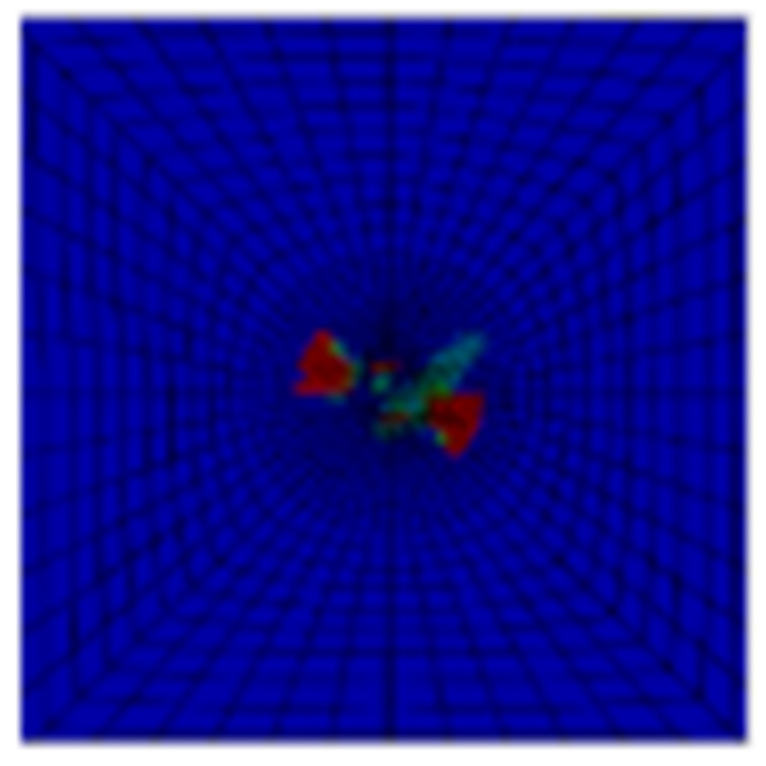	21	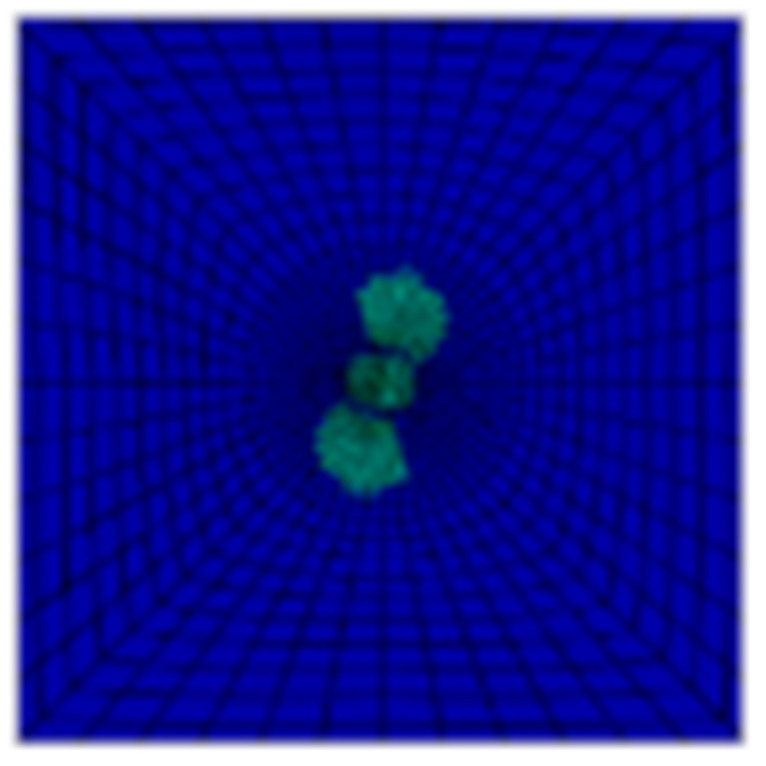	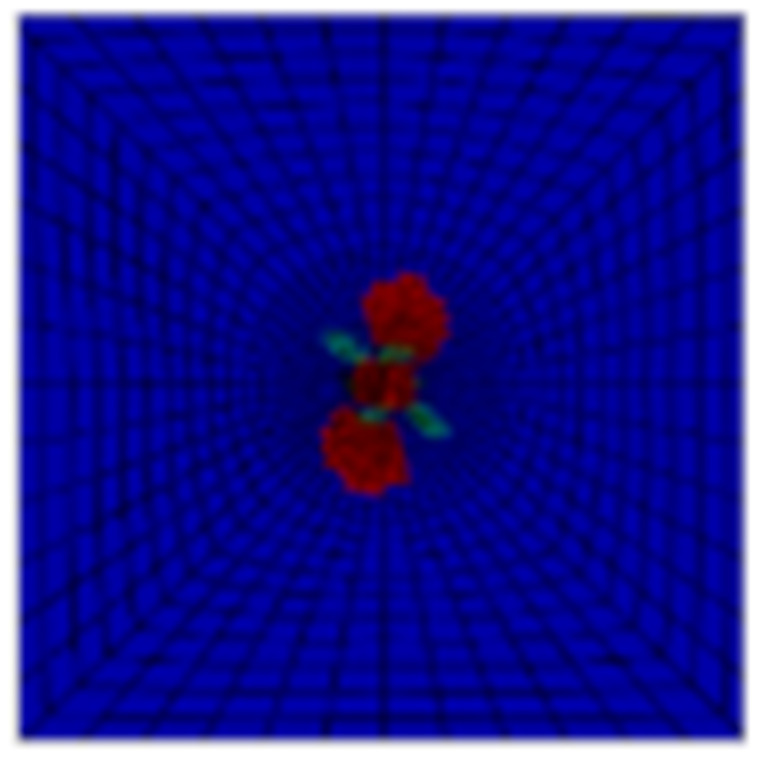
10	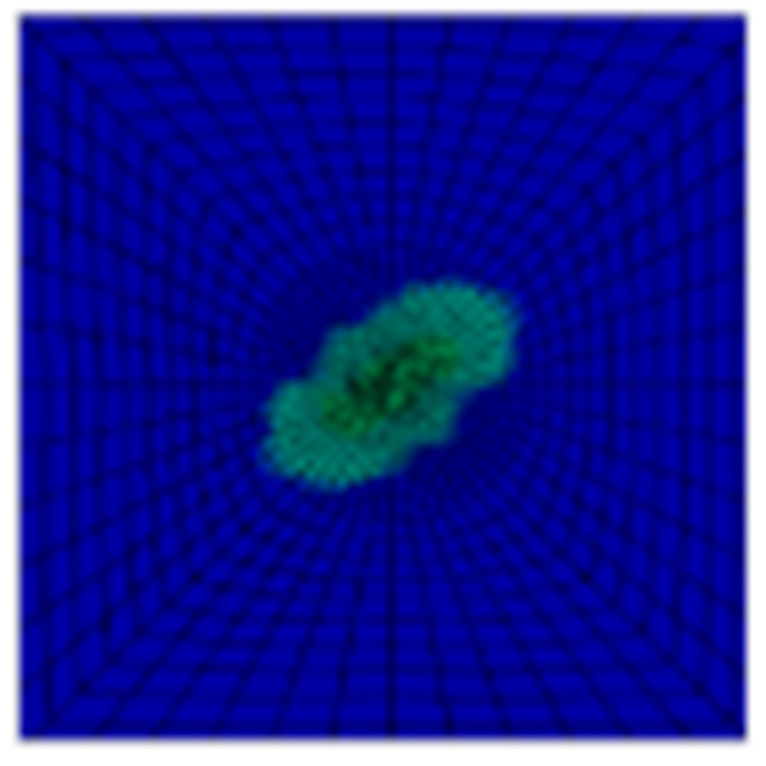	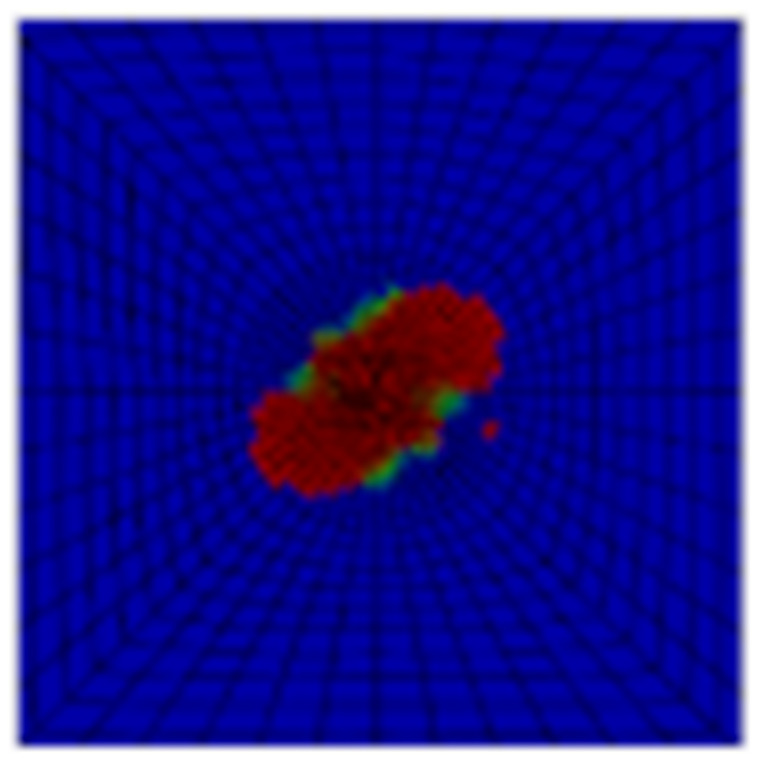	22	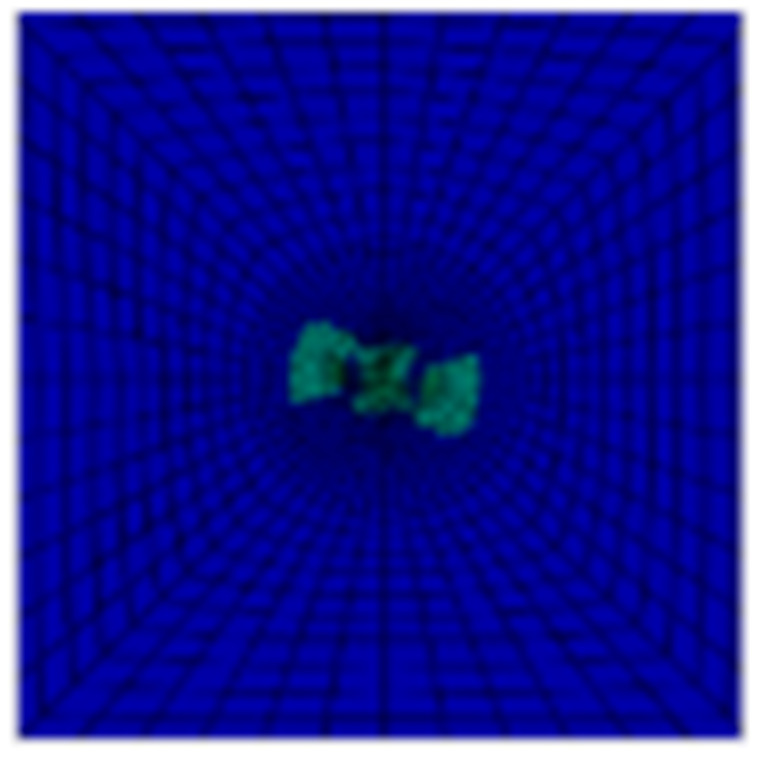	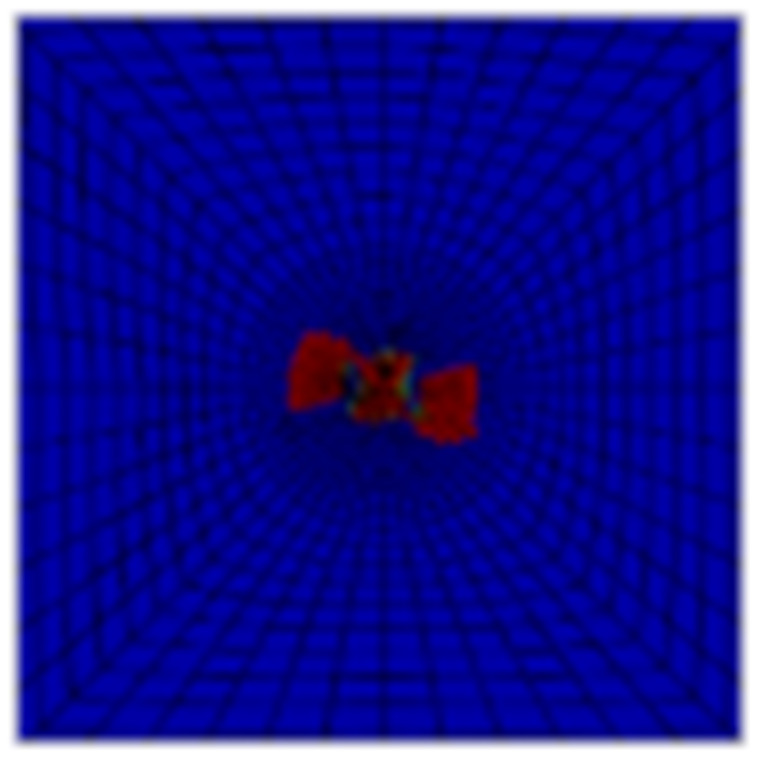
11	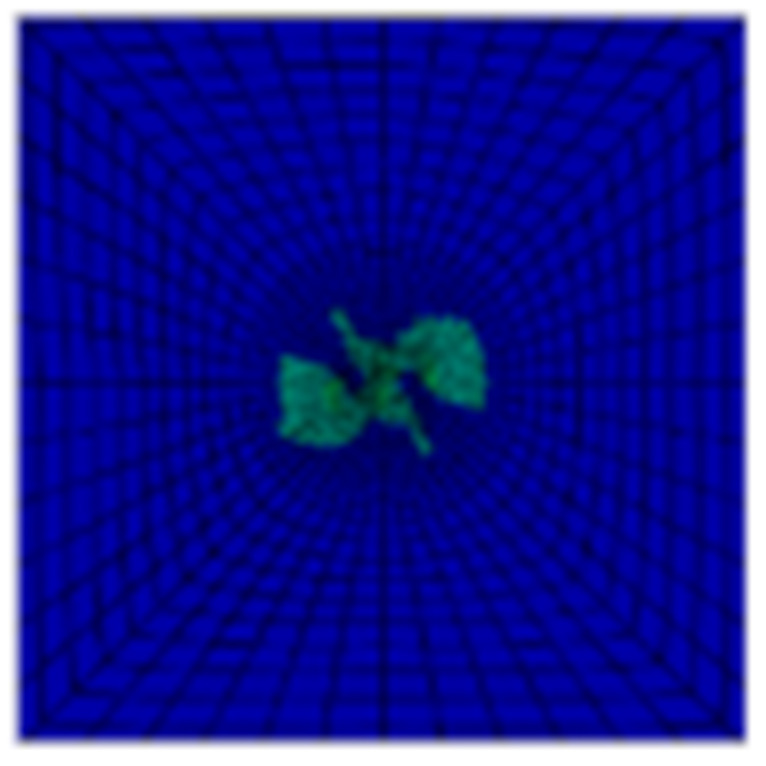	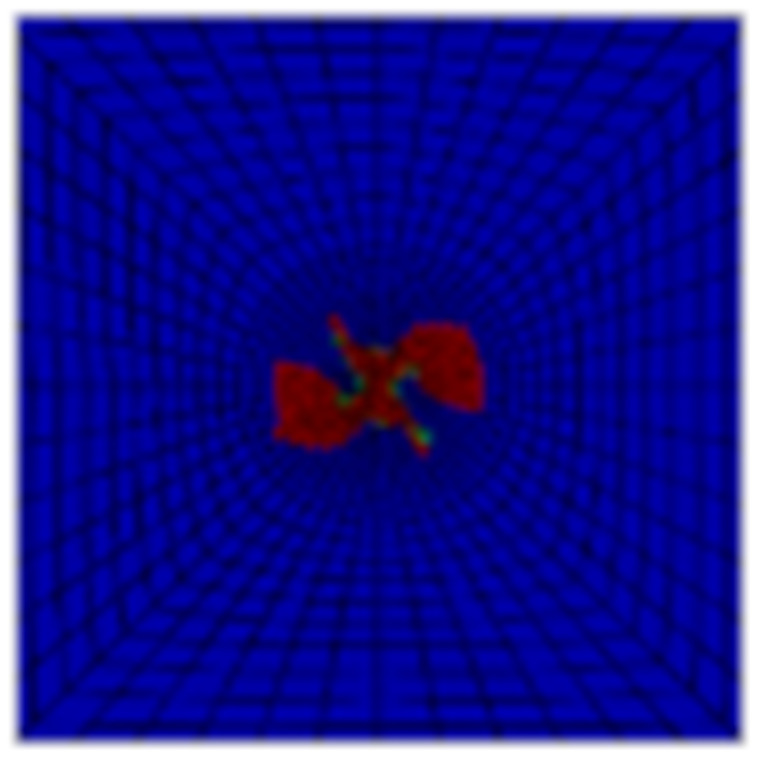	23	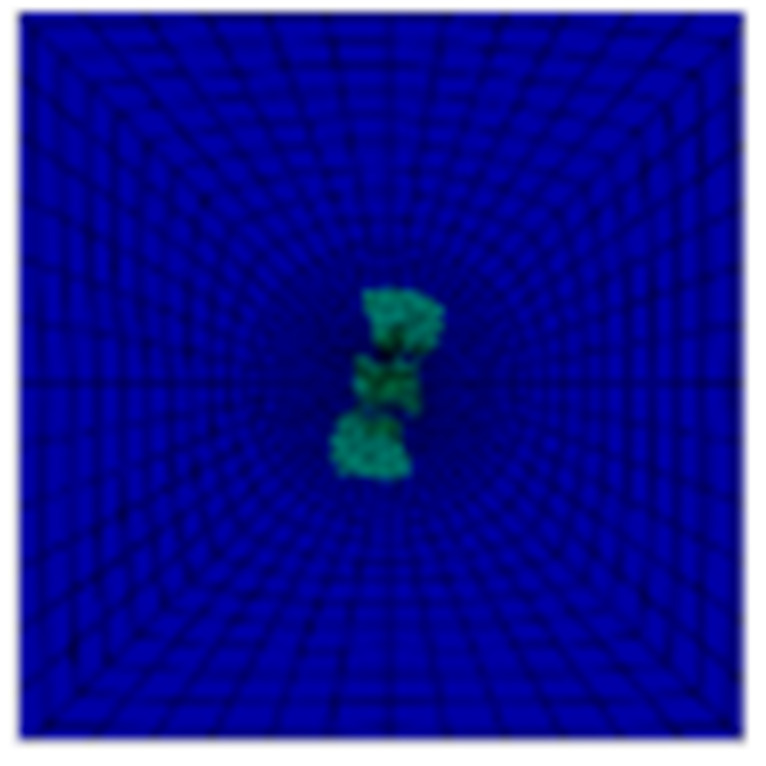	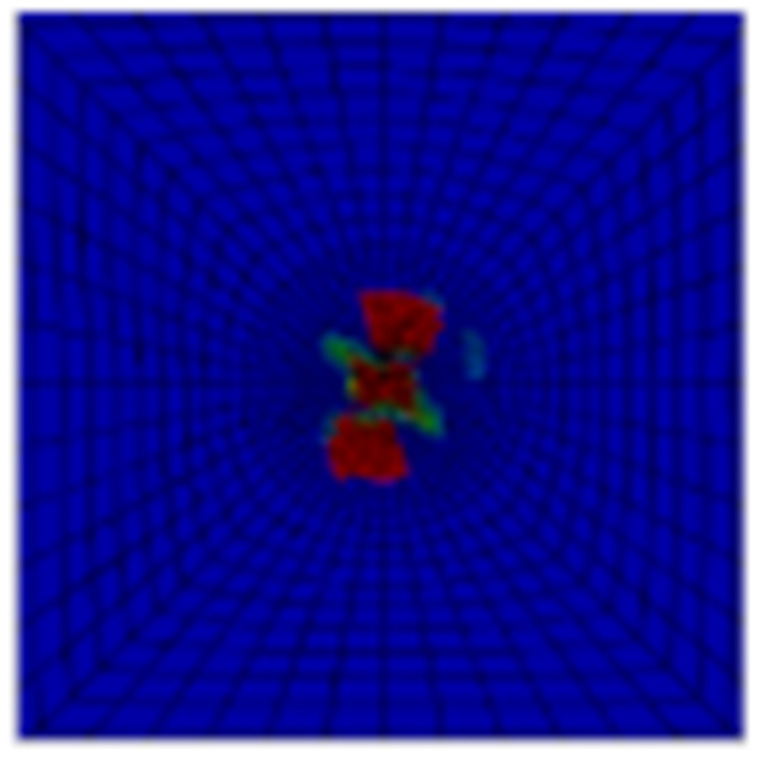
12	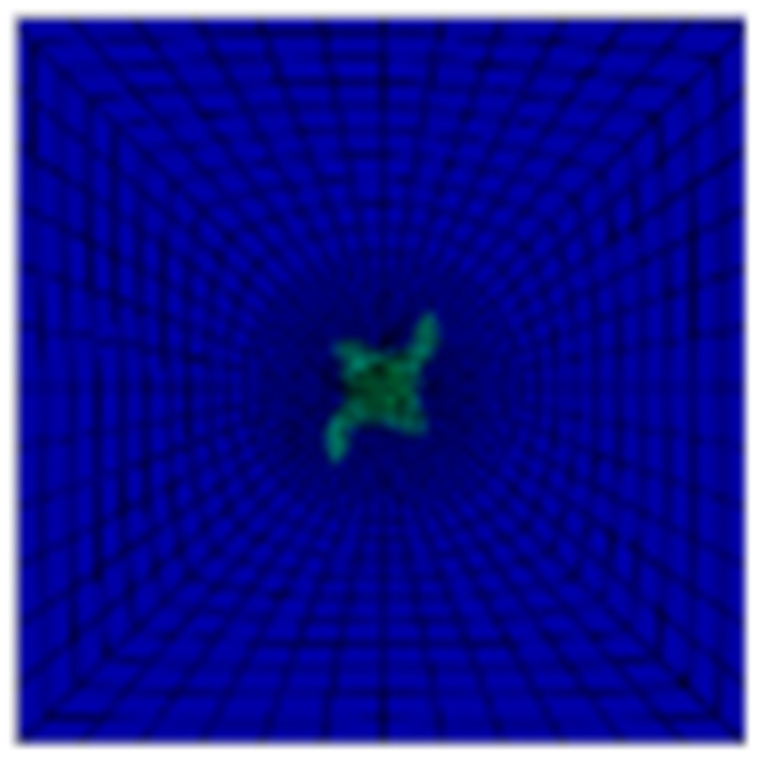	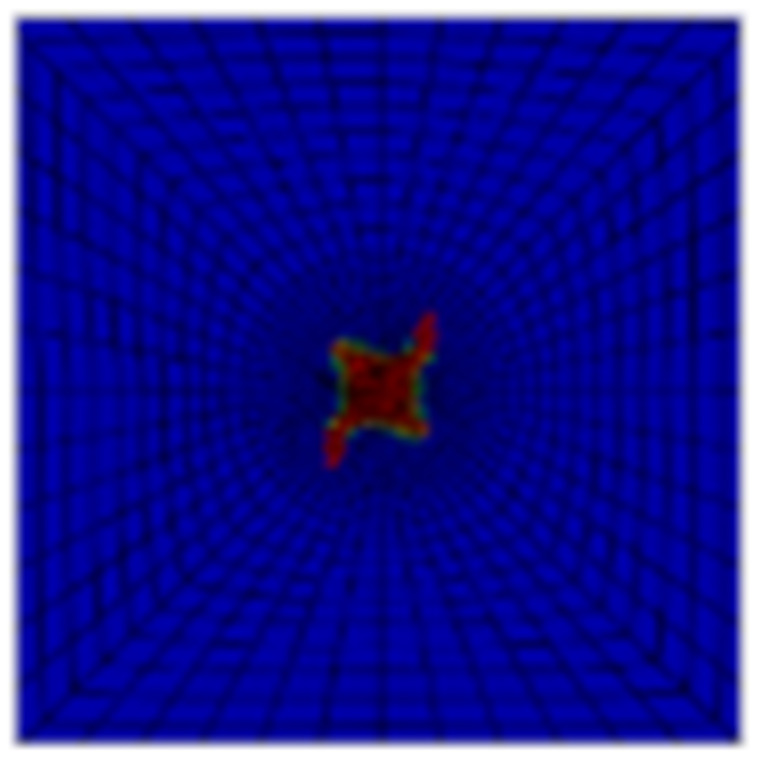	24	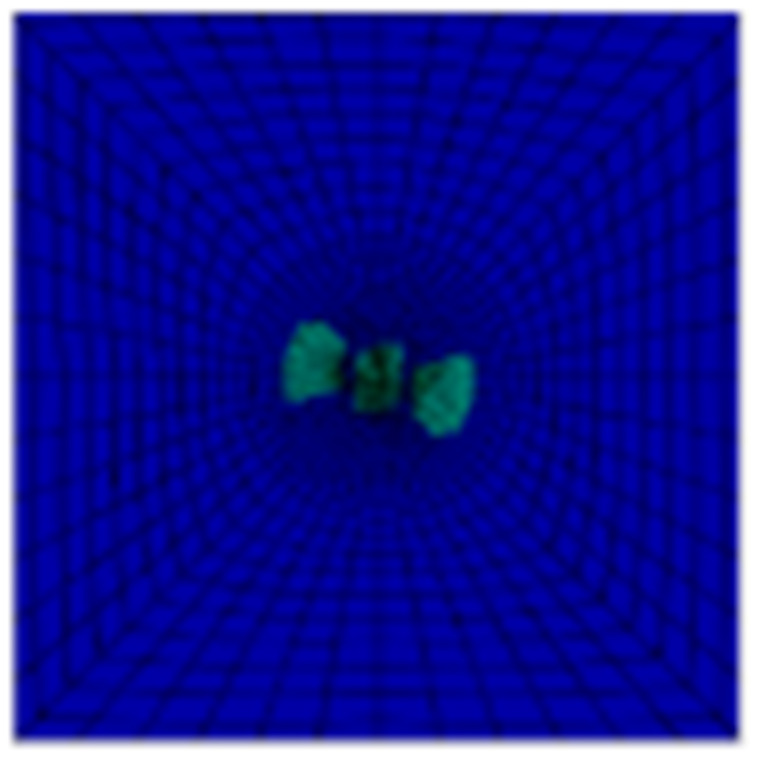	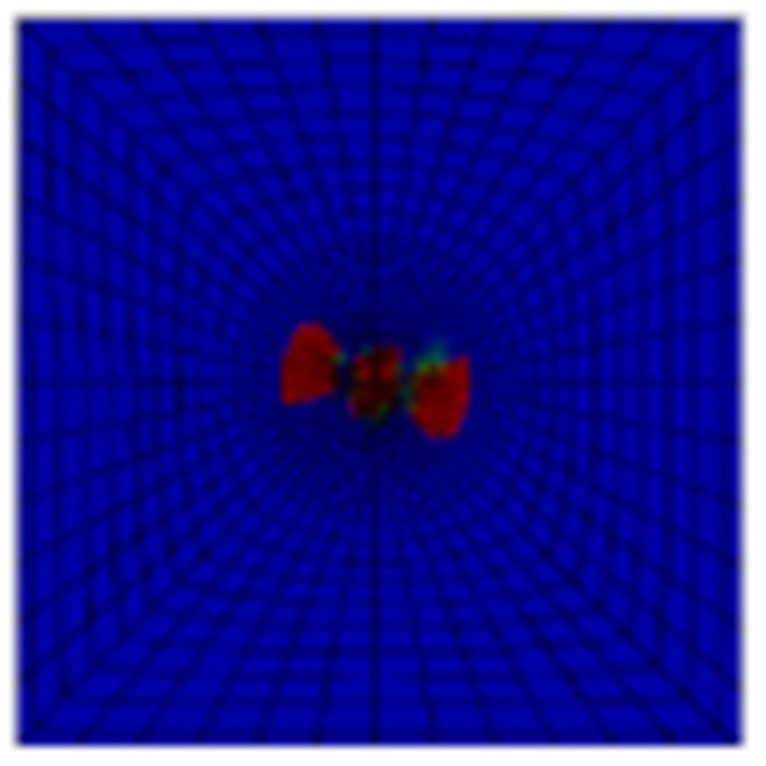

**Table 8 materials-15-05029-t008:** Matrix failure status of the composite laminate with an impact energy of 50 J.

Layer No.	Impacted Results	Shear Results	Layer No.	Impacted Results	Shear Results
1	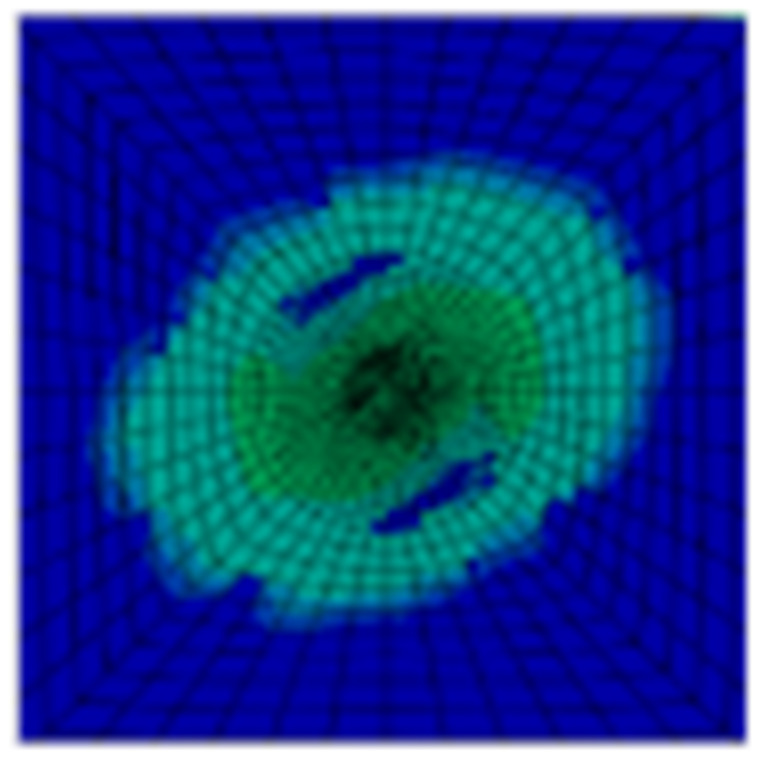	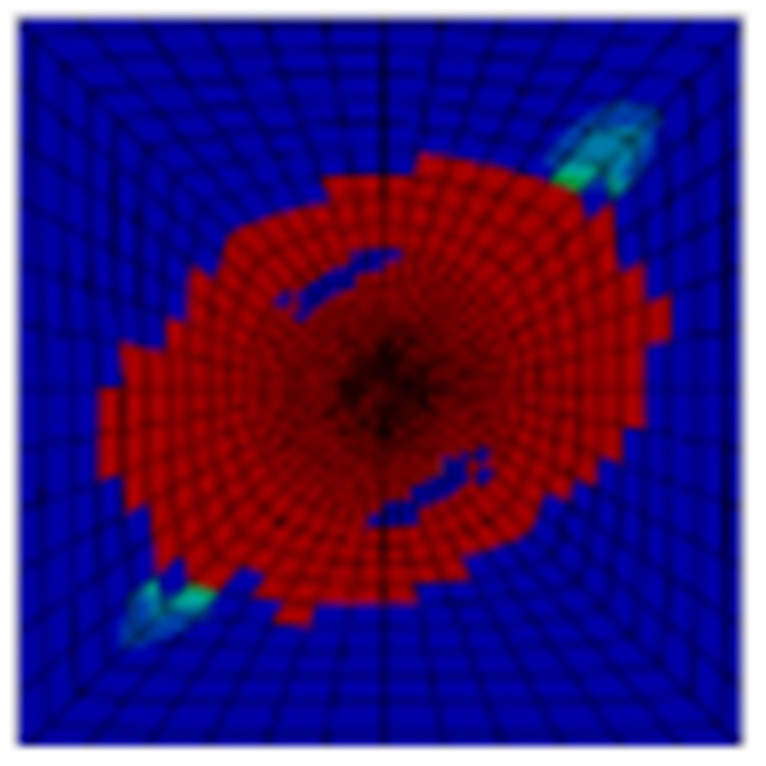	13	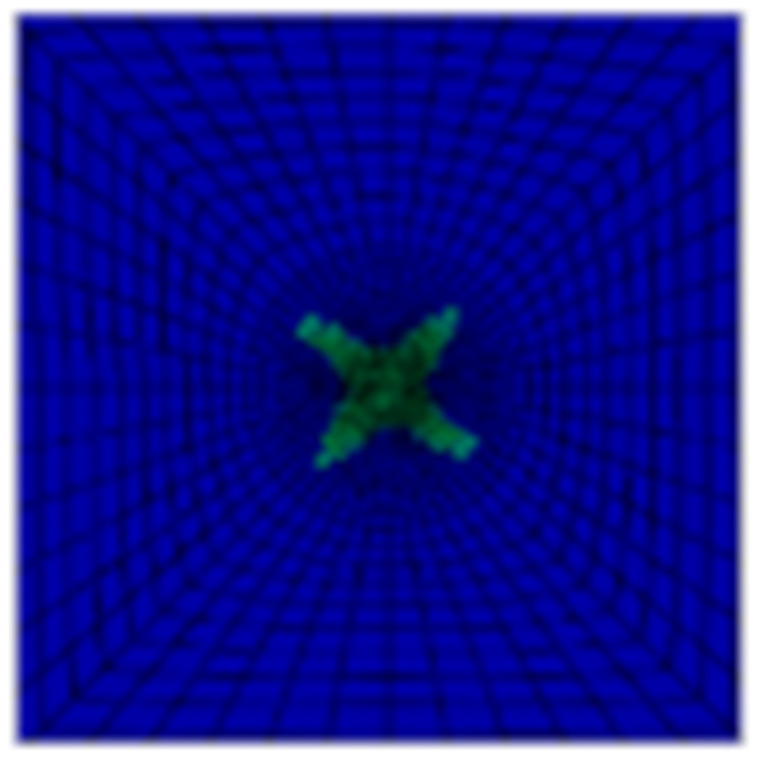	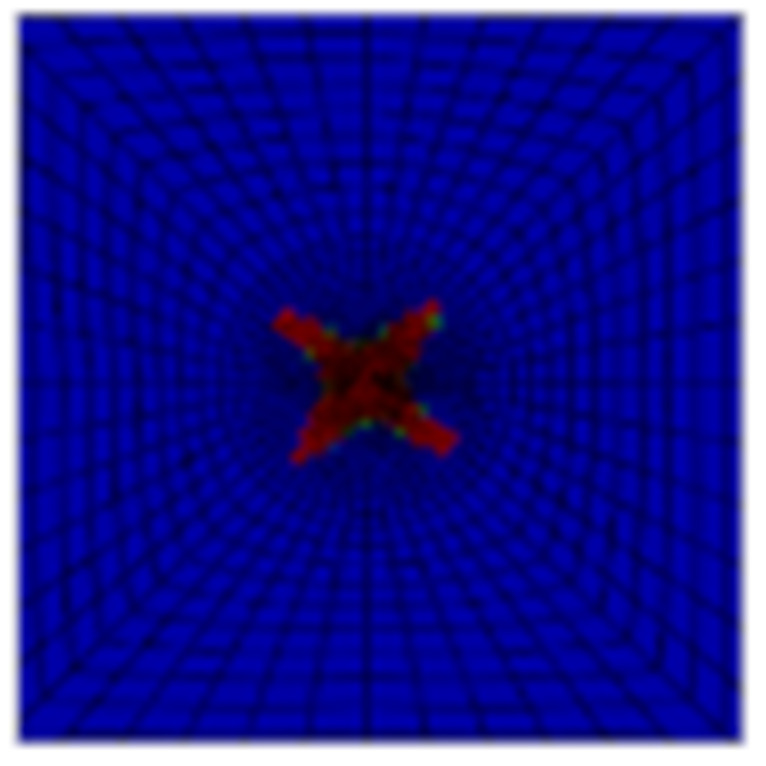
2	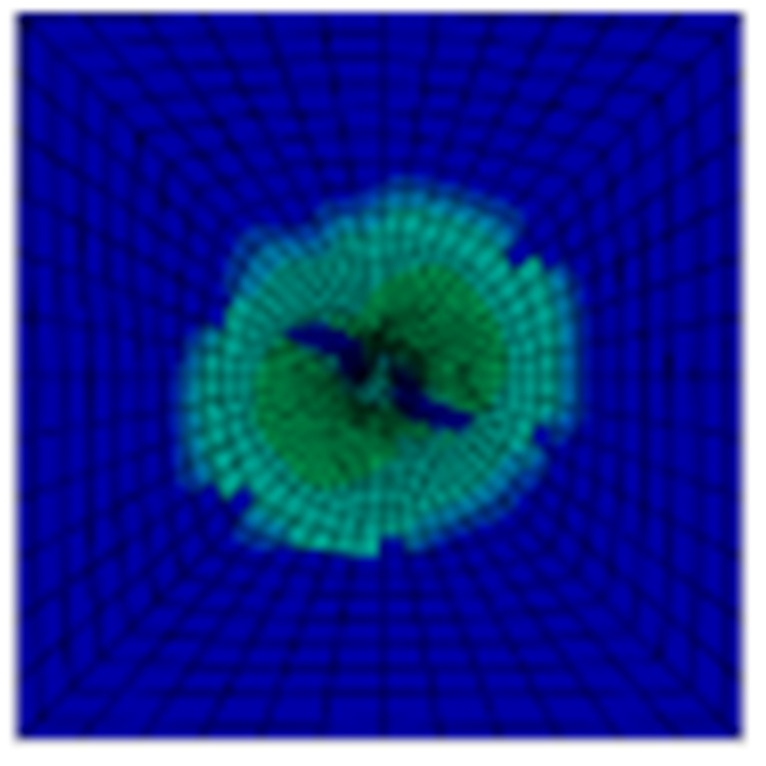	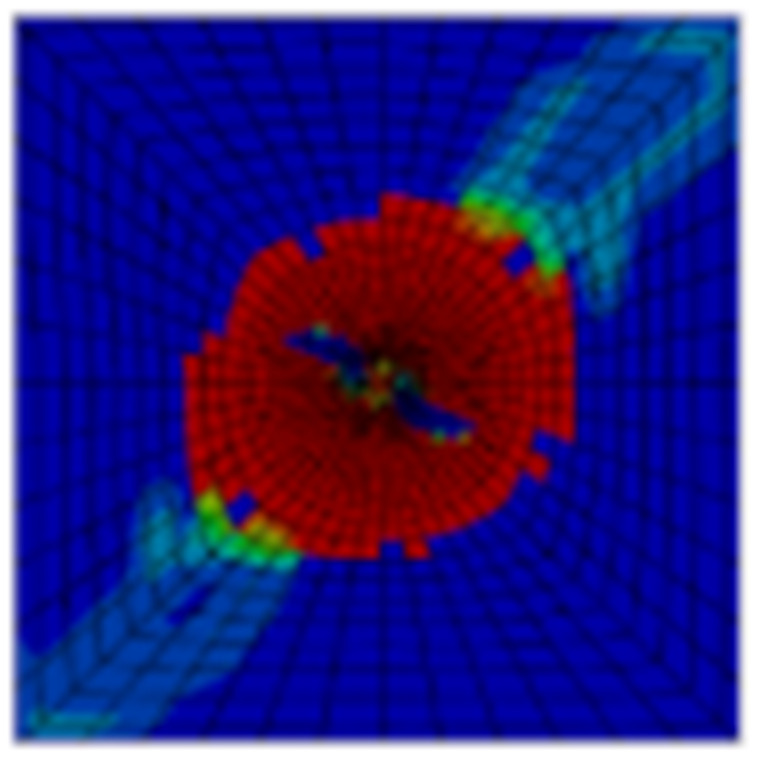	14	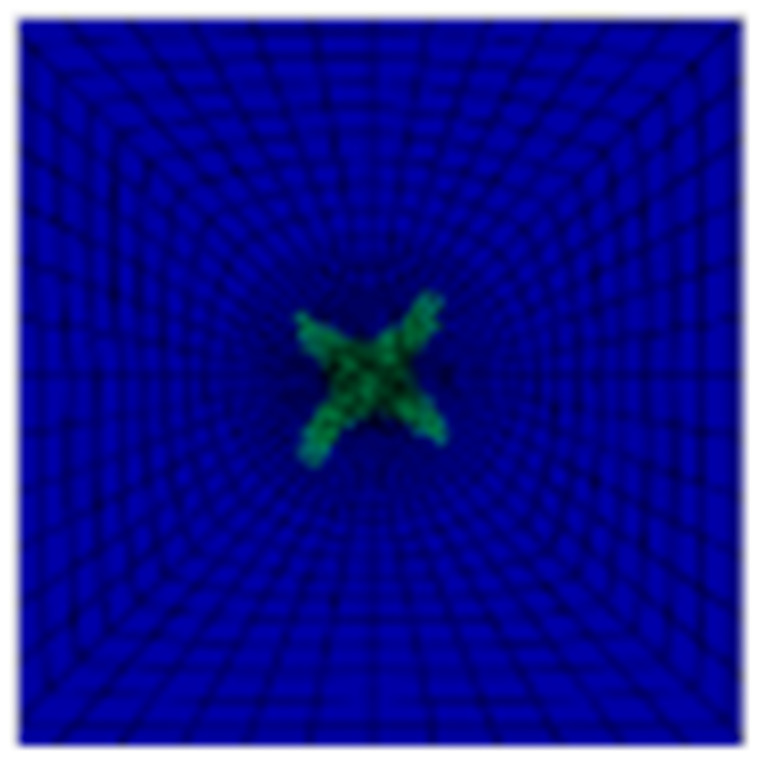	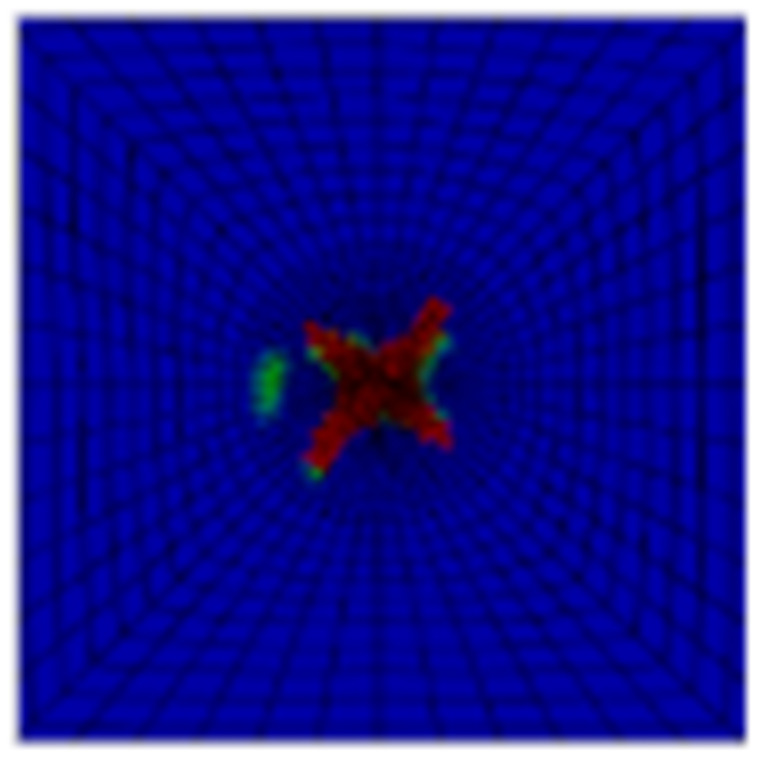
3	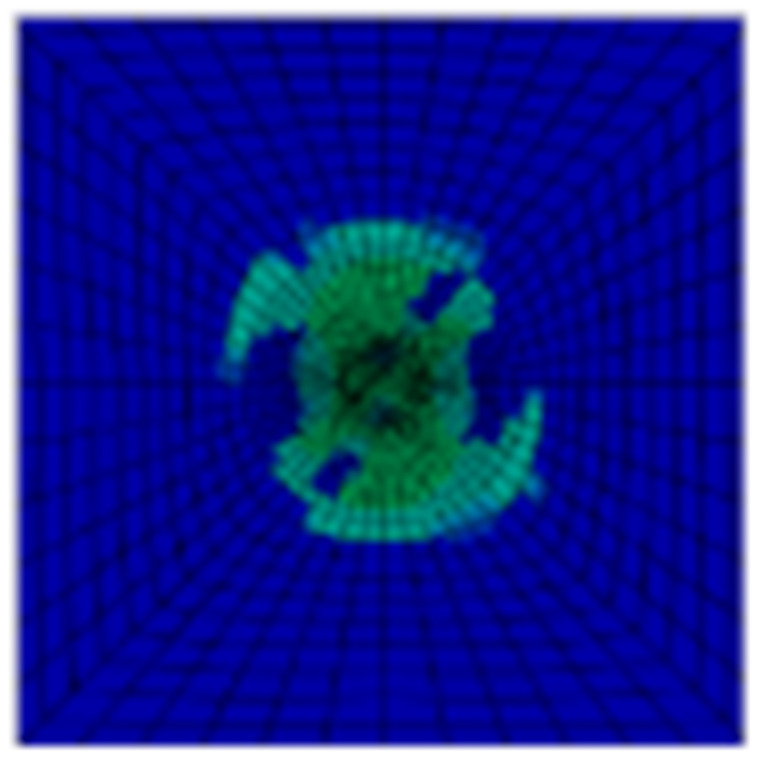	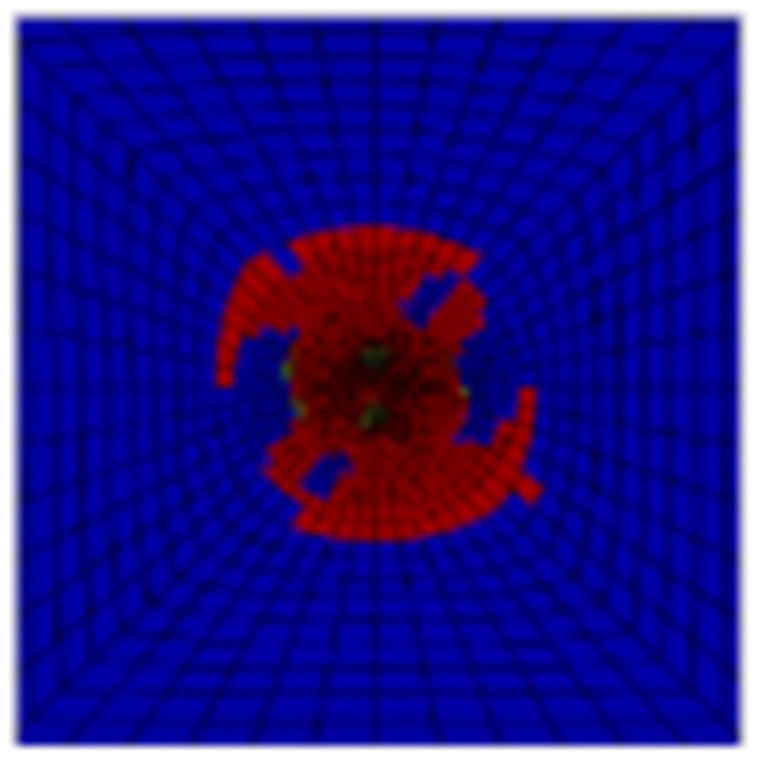	15	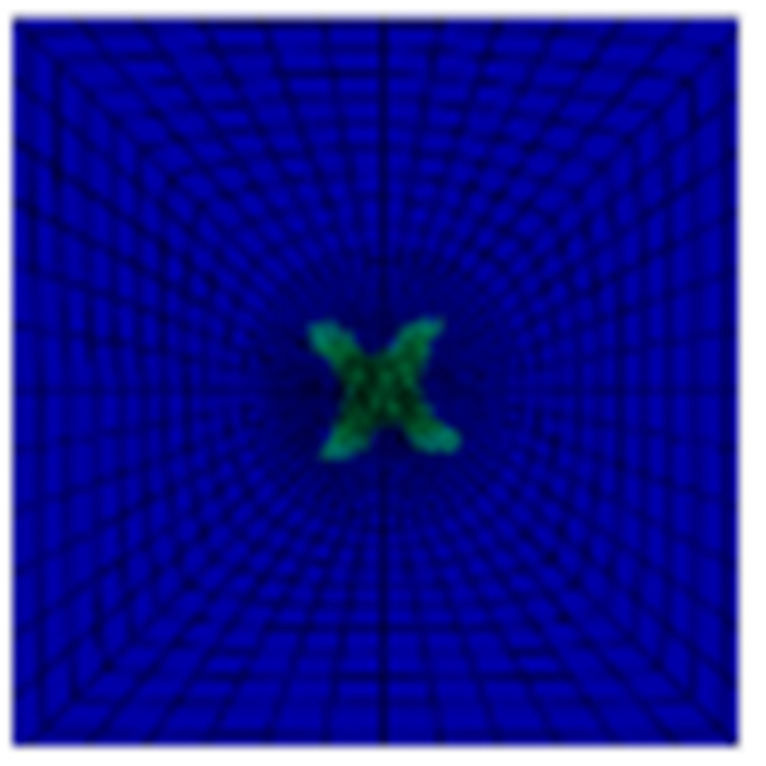	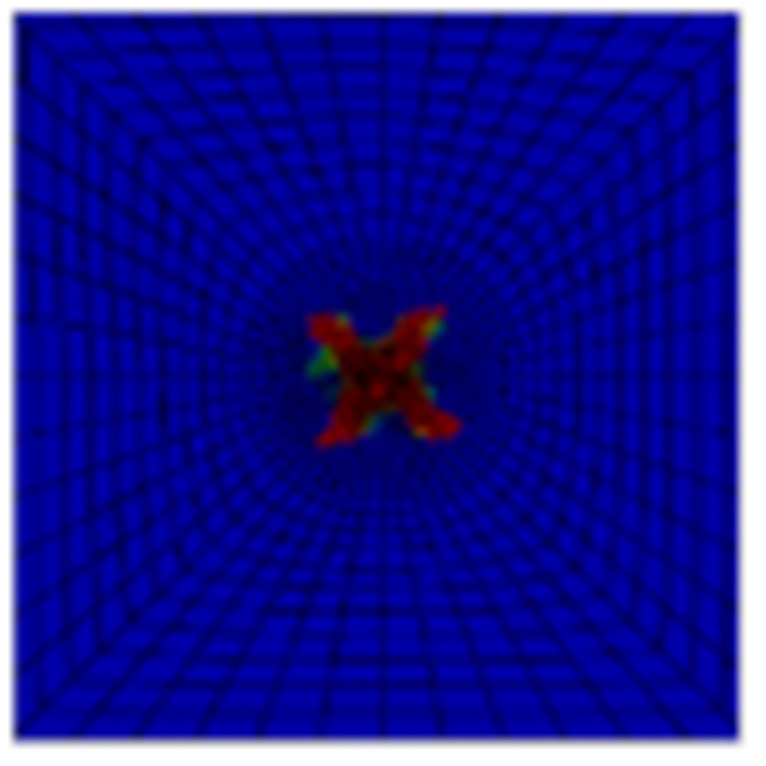
4	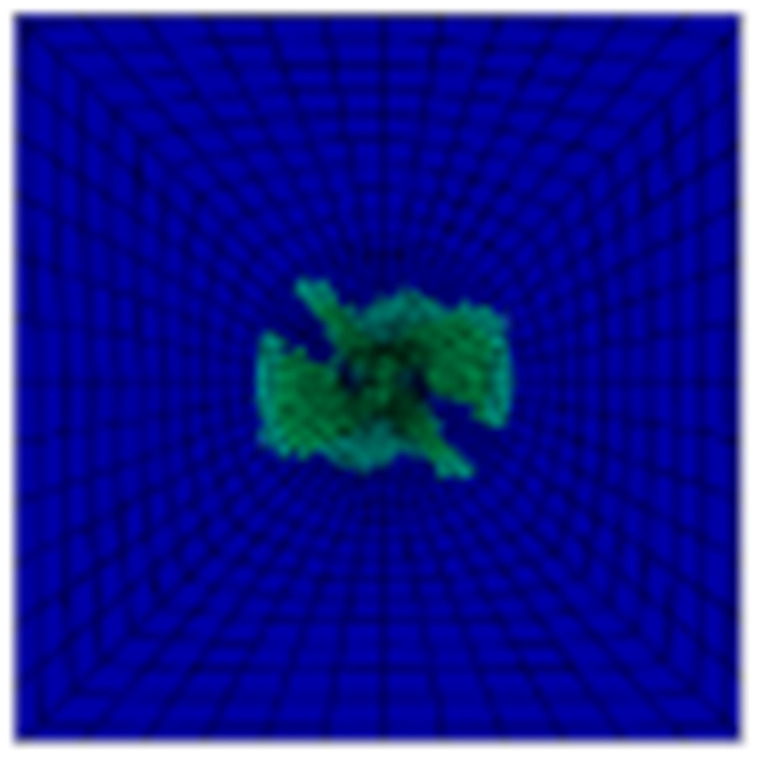	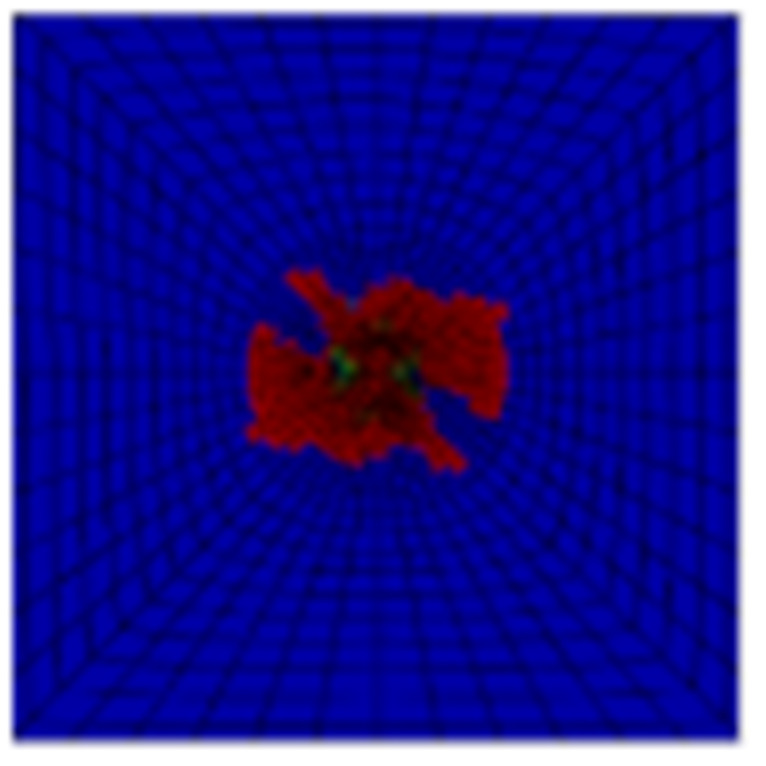	16	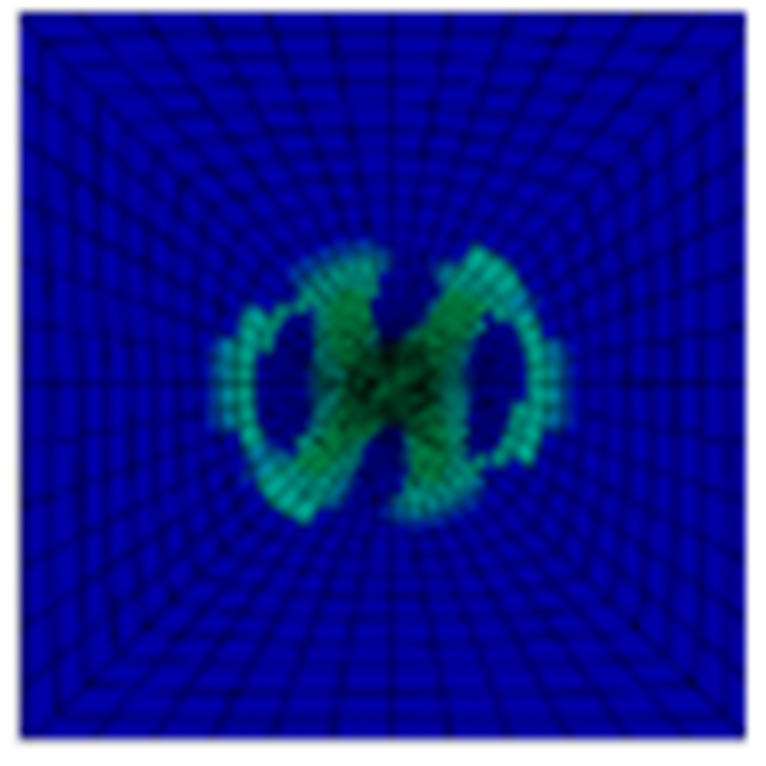	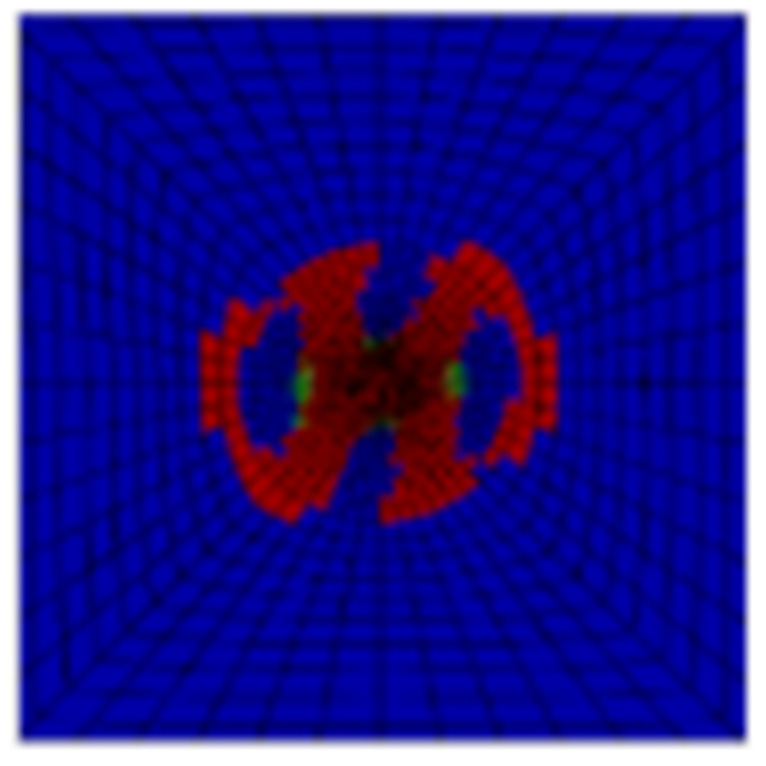
5	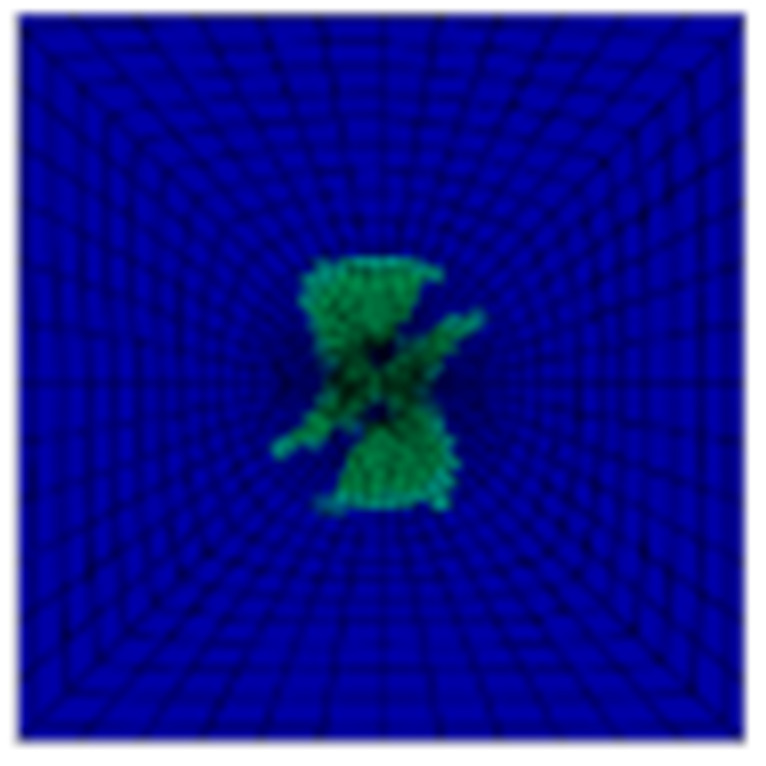	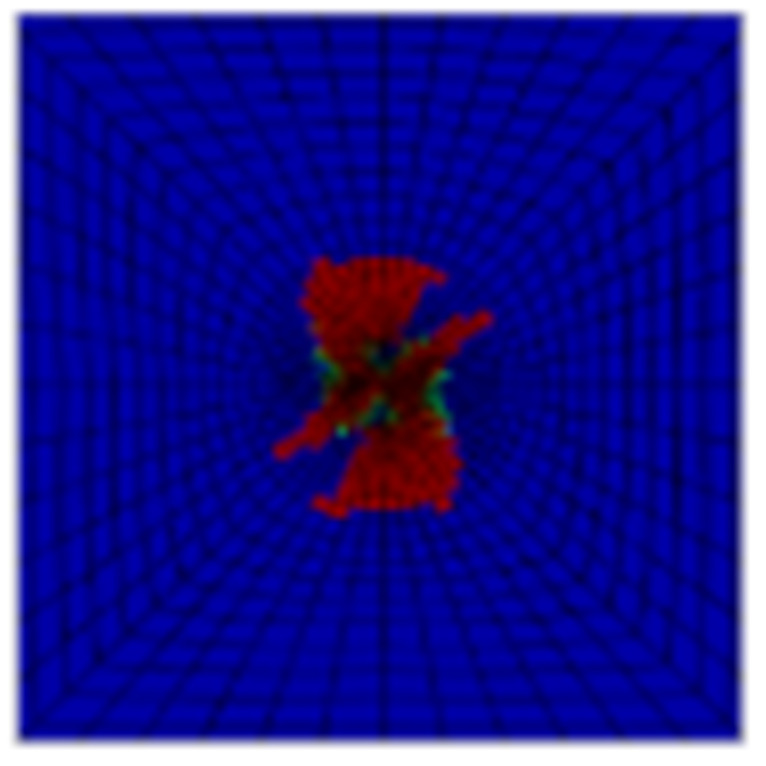	17	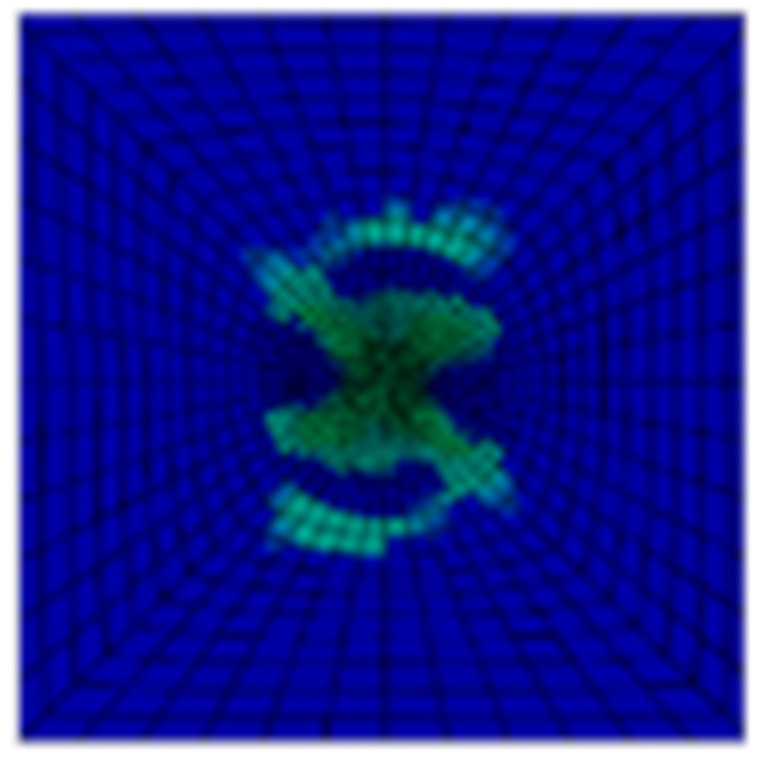	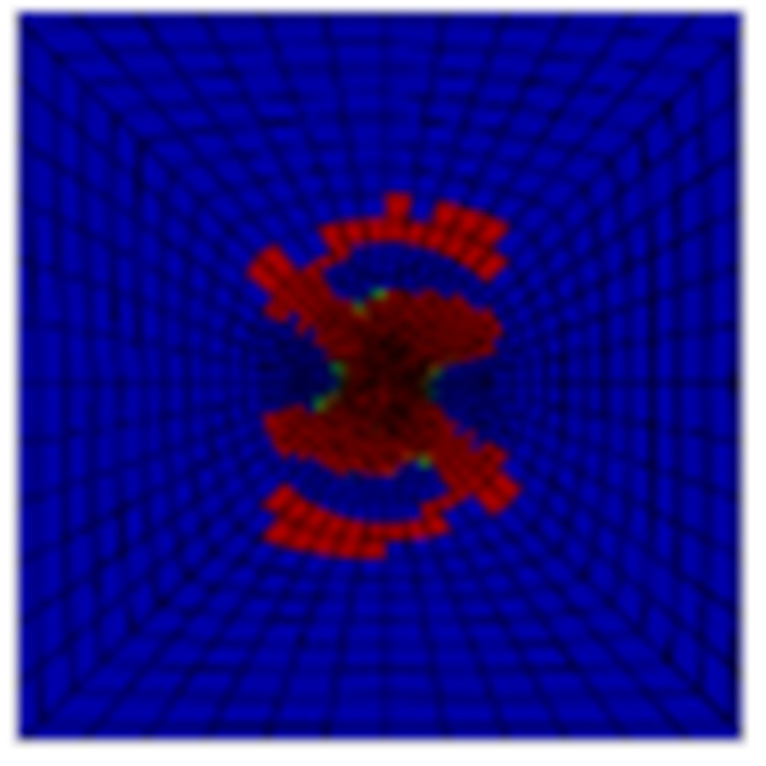
6	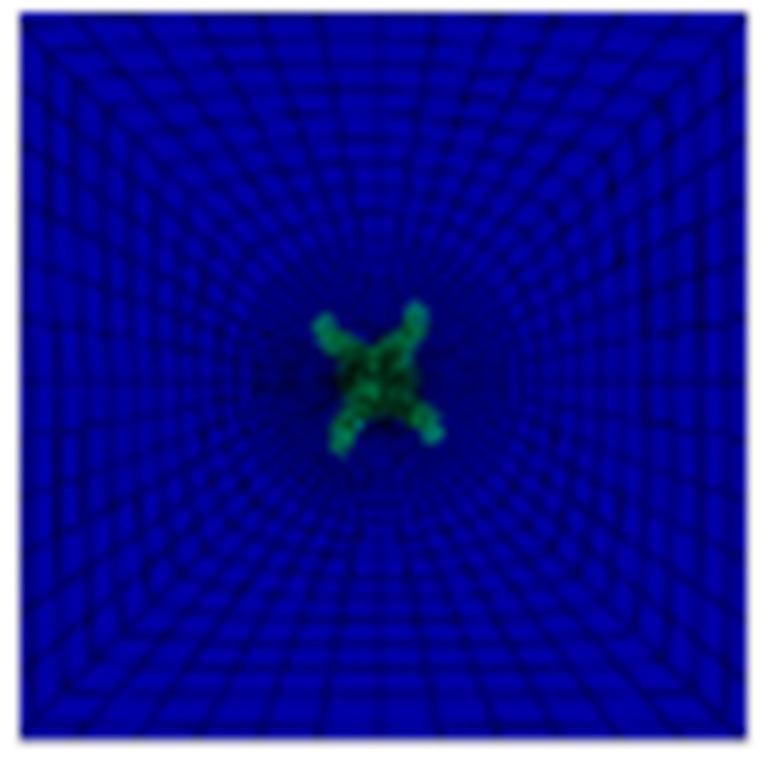	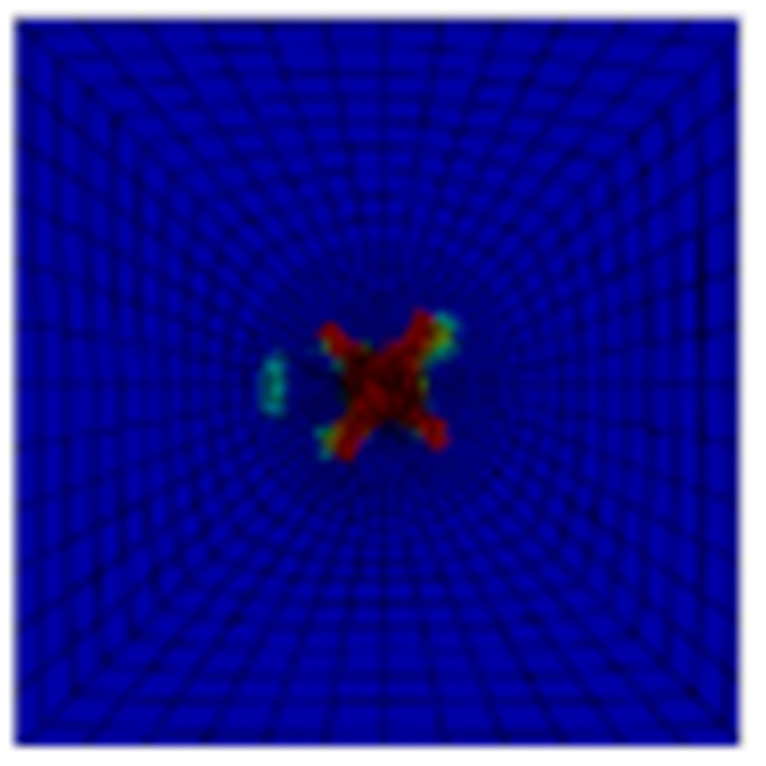	18	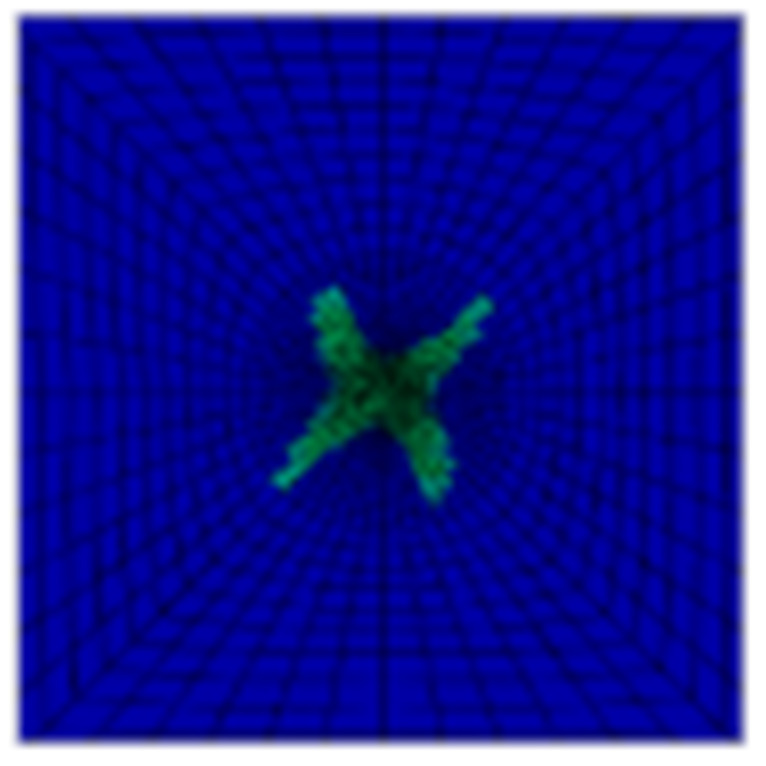	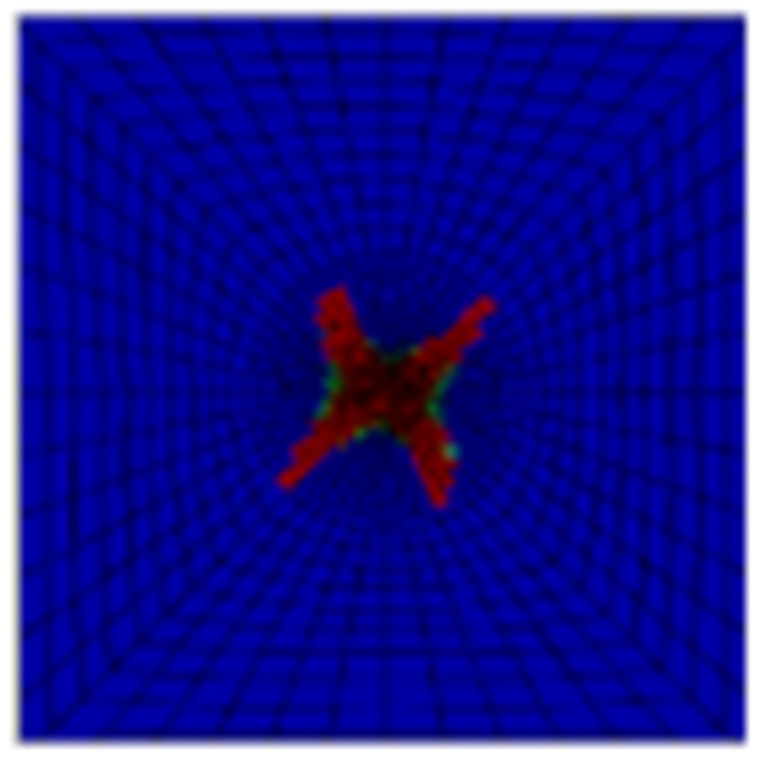
7	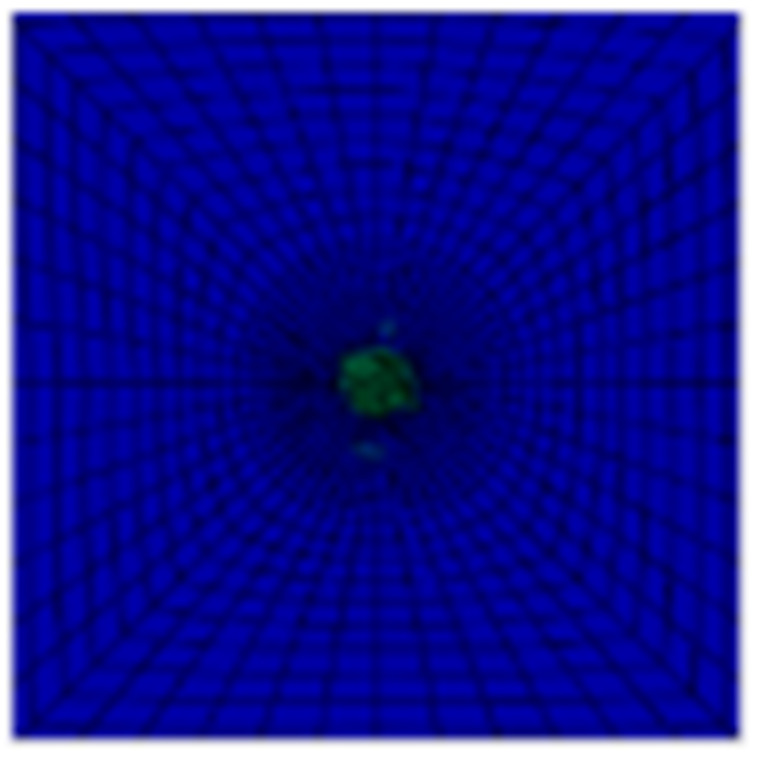	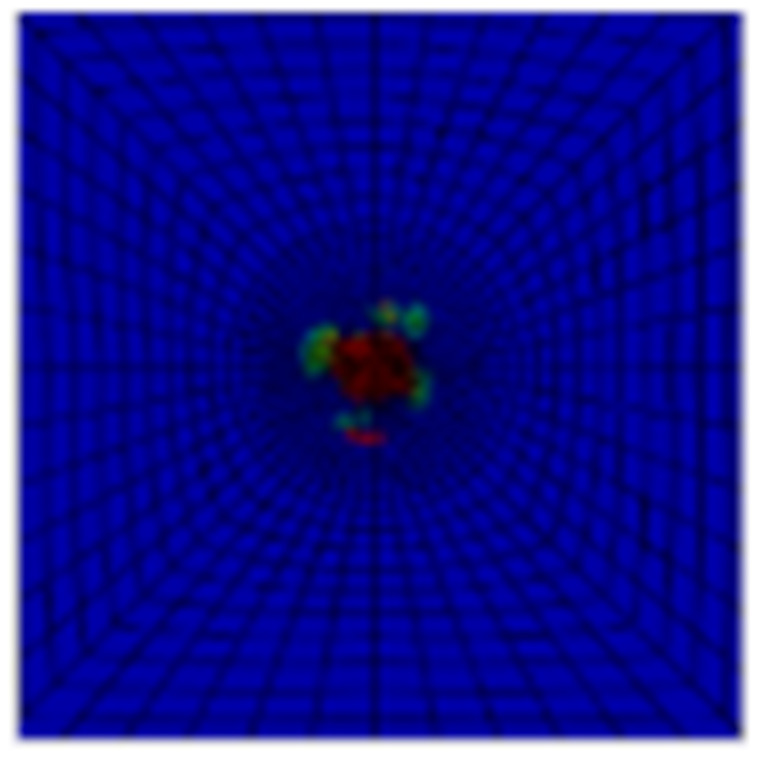	19	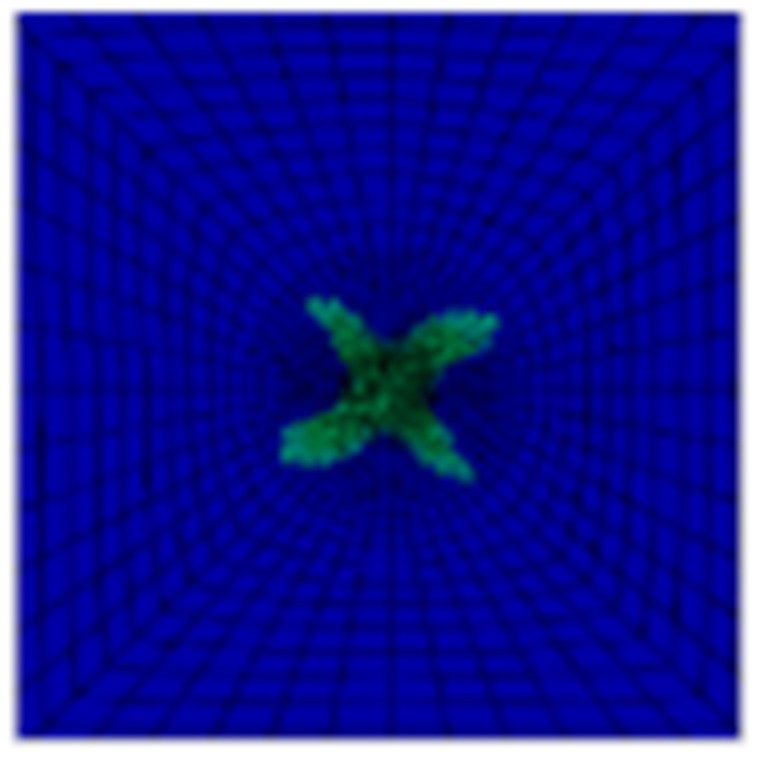	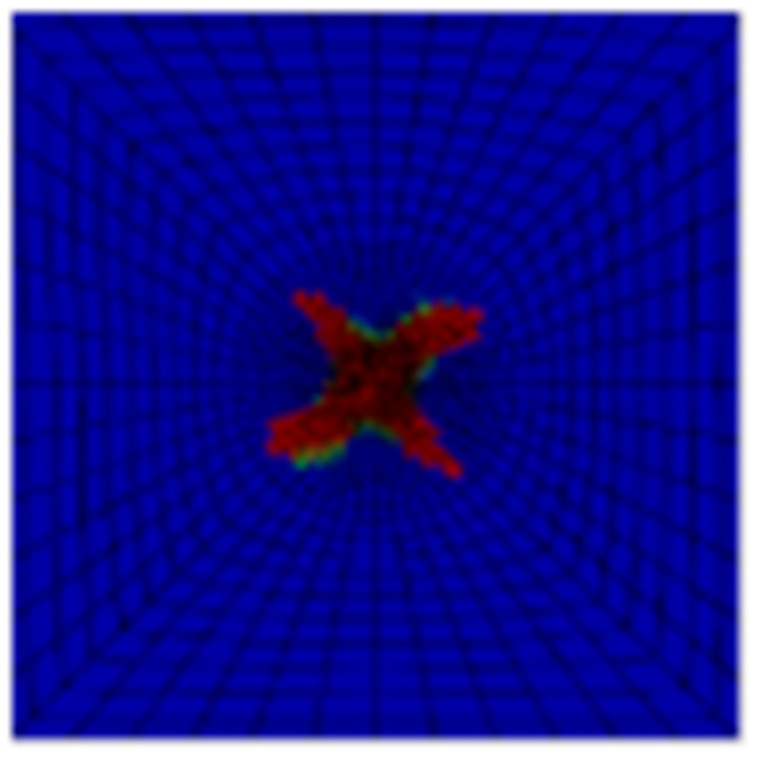
8	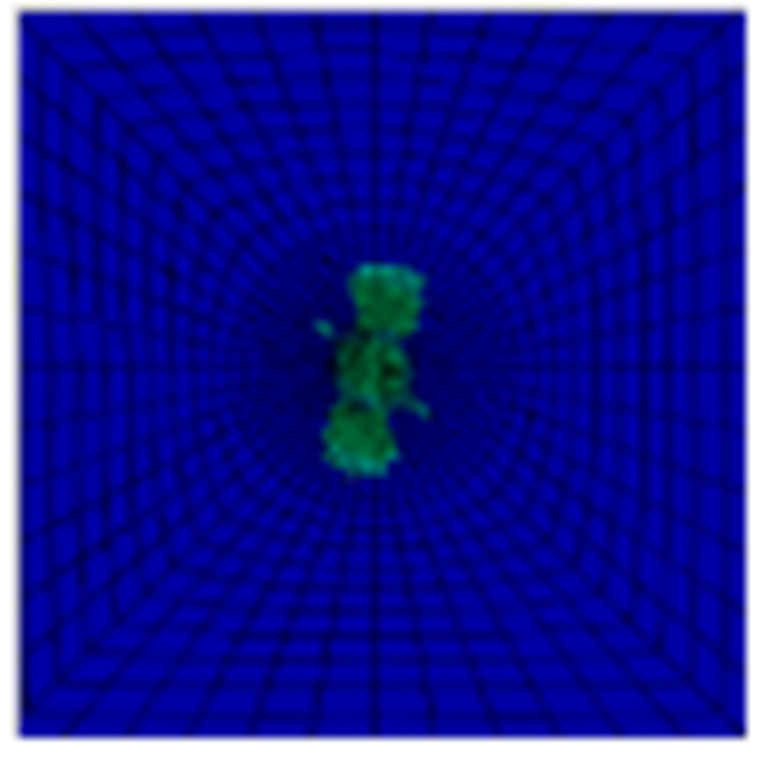	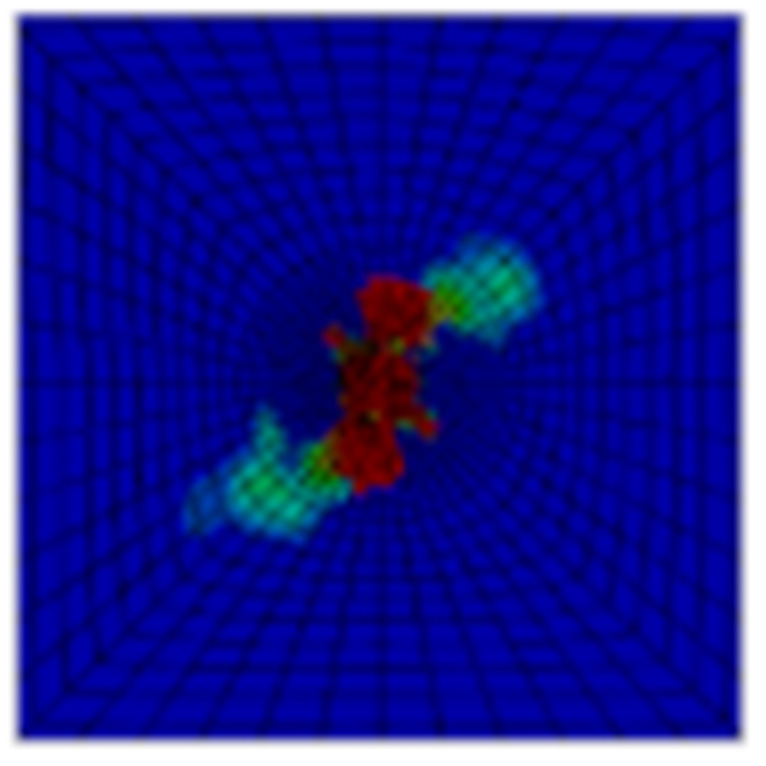	20	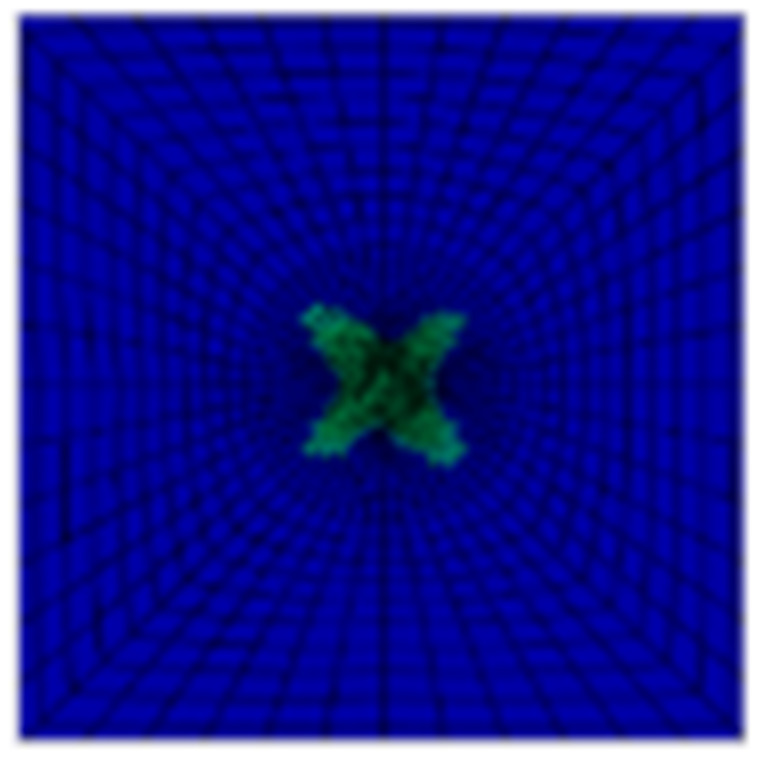	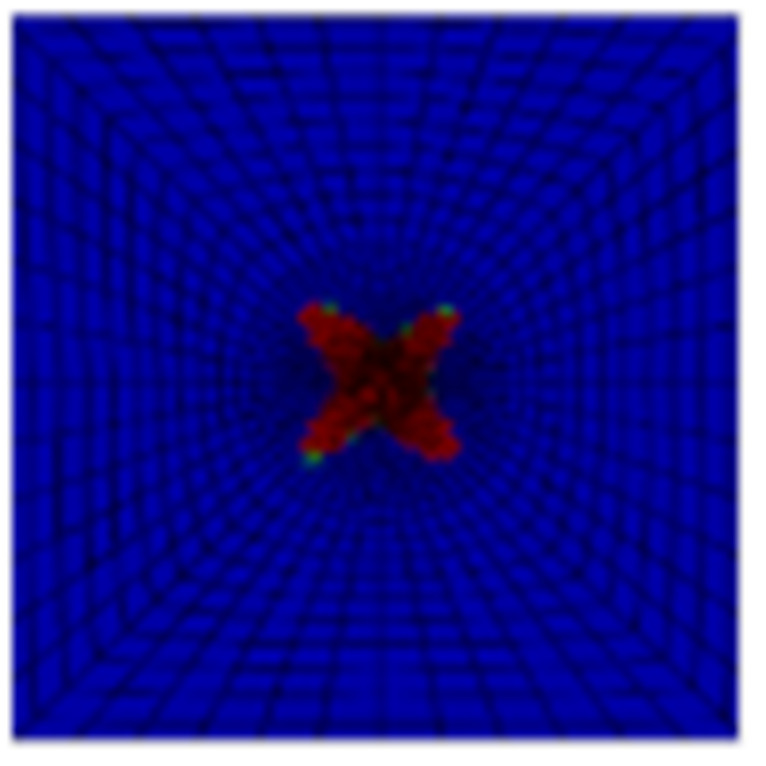
9	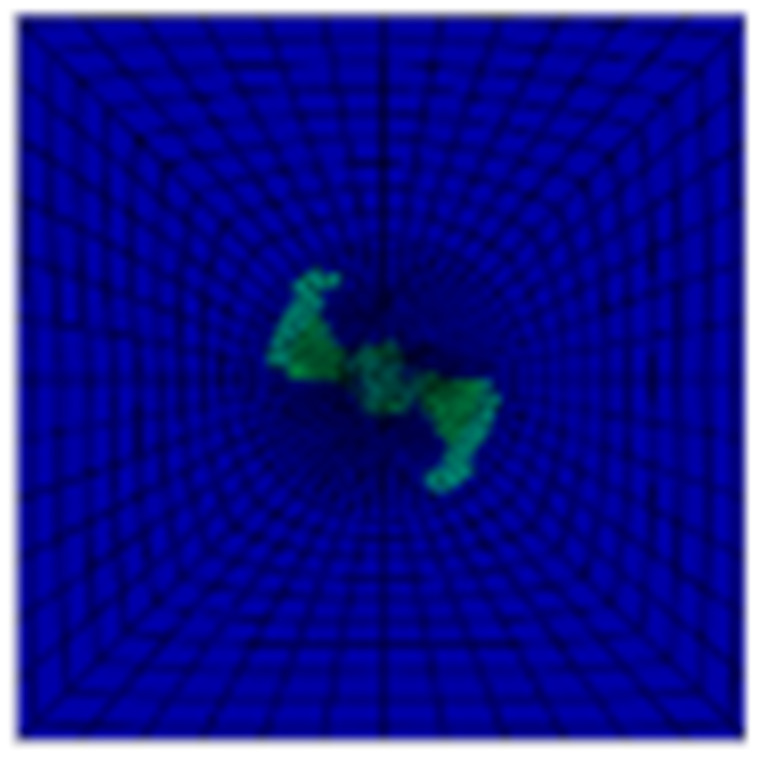	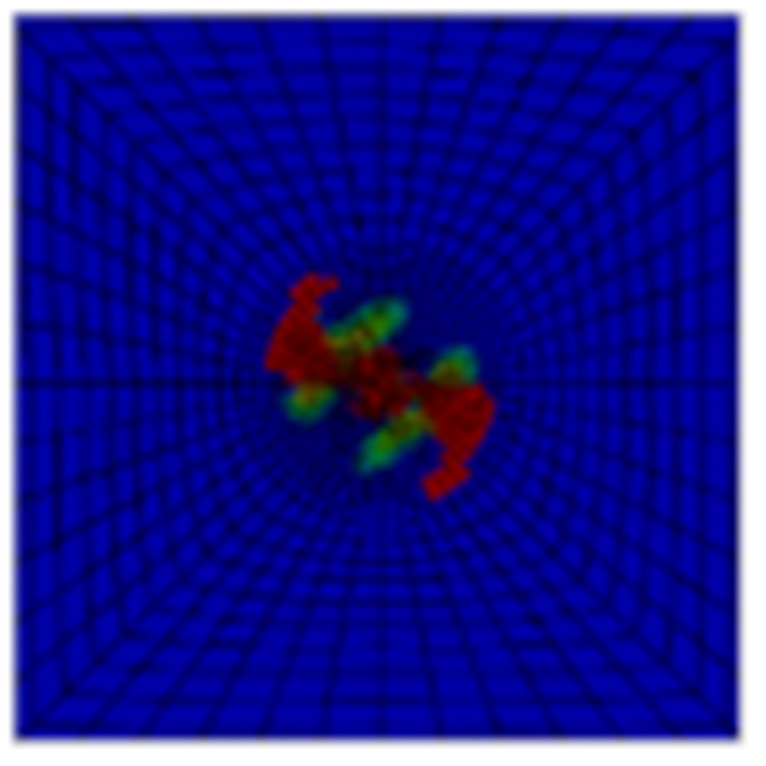	21	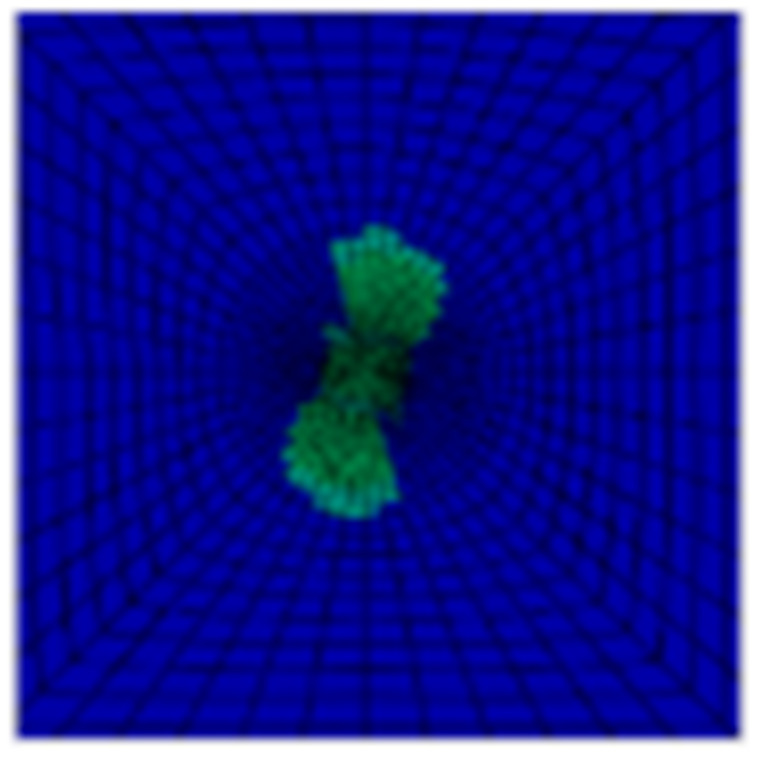	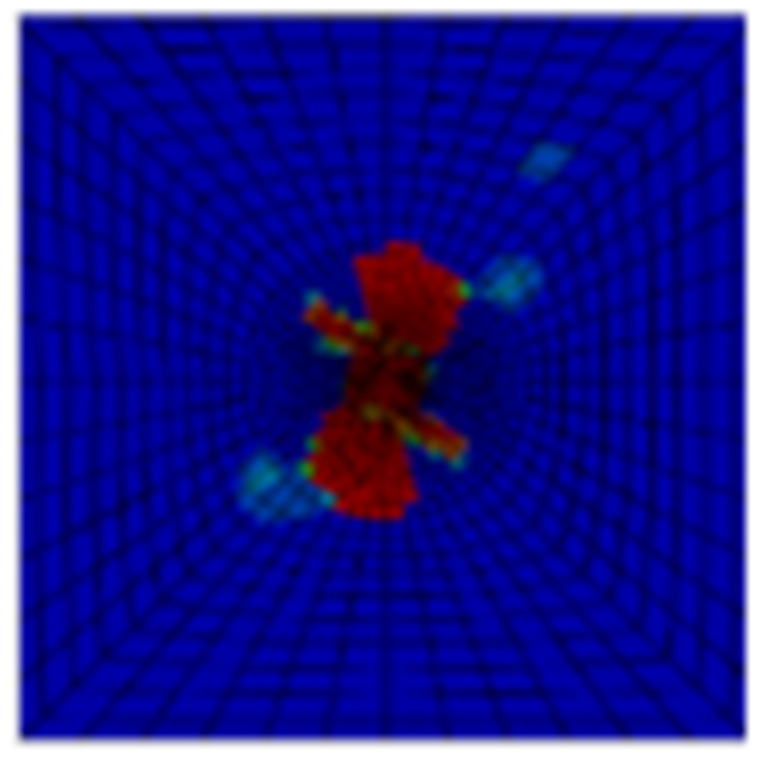
10	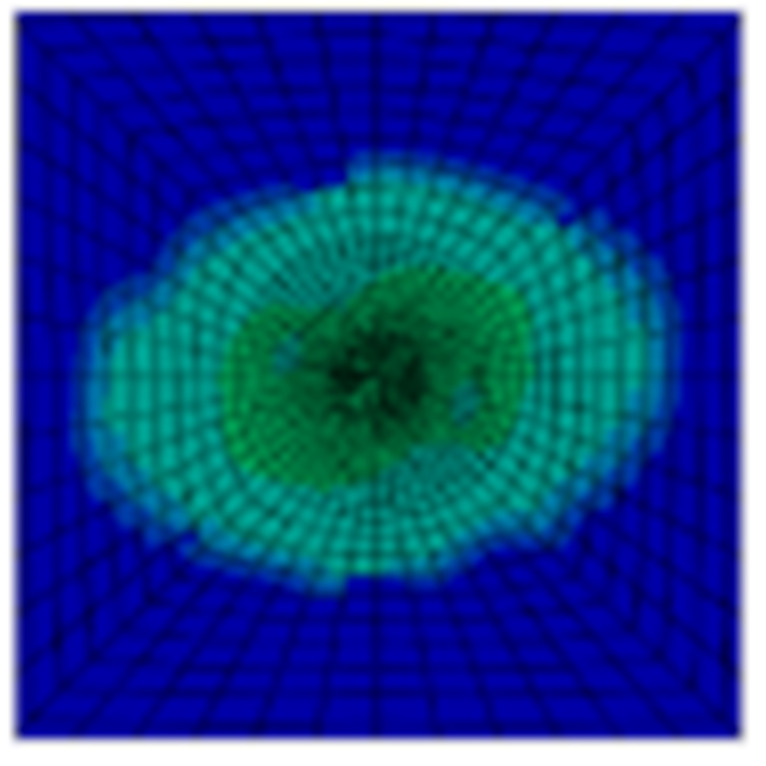	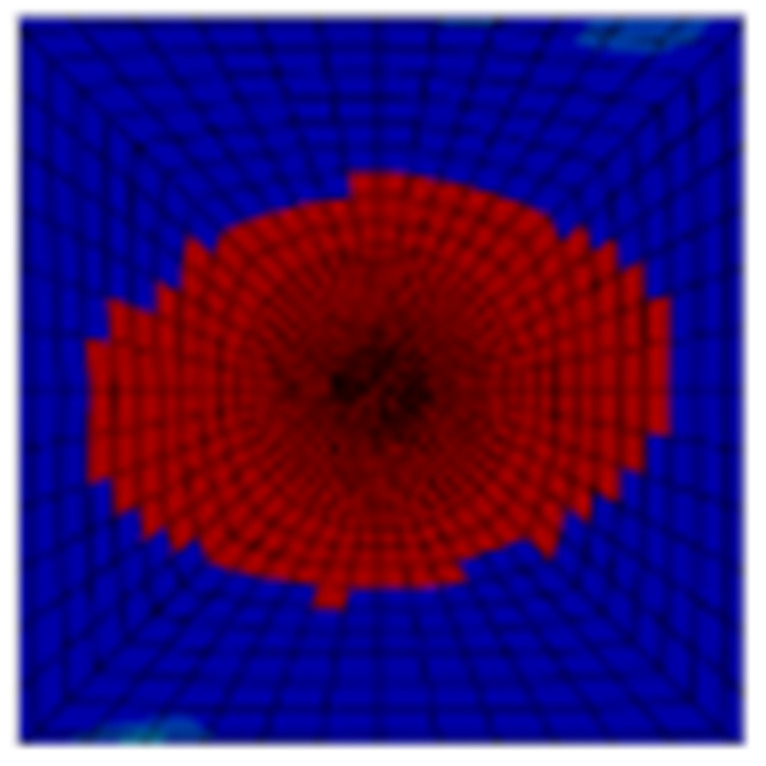	22	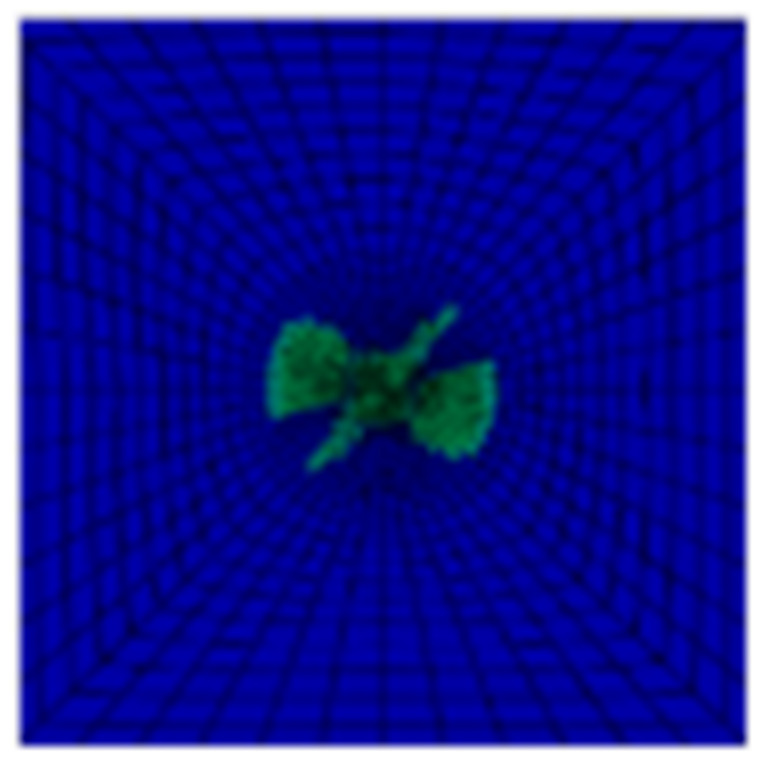	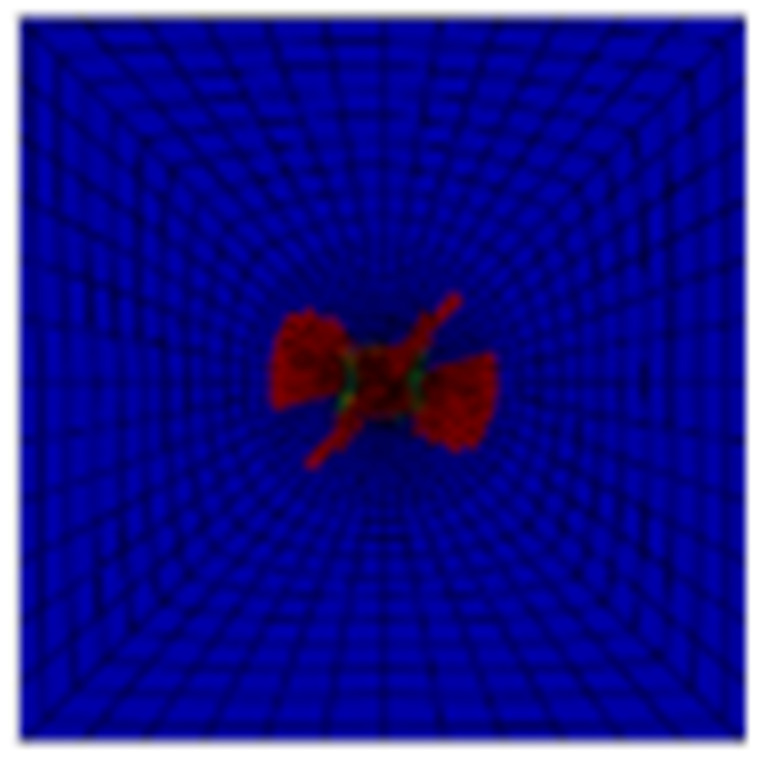
11	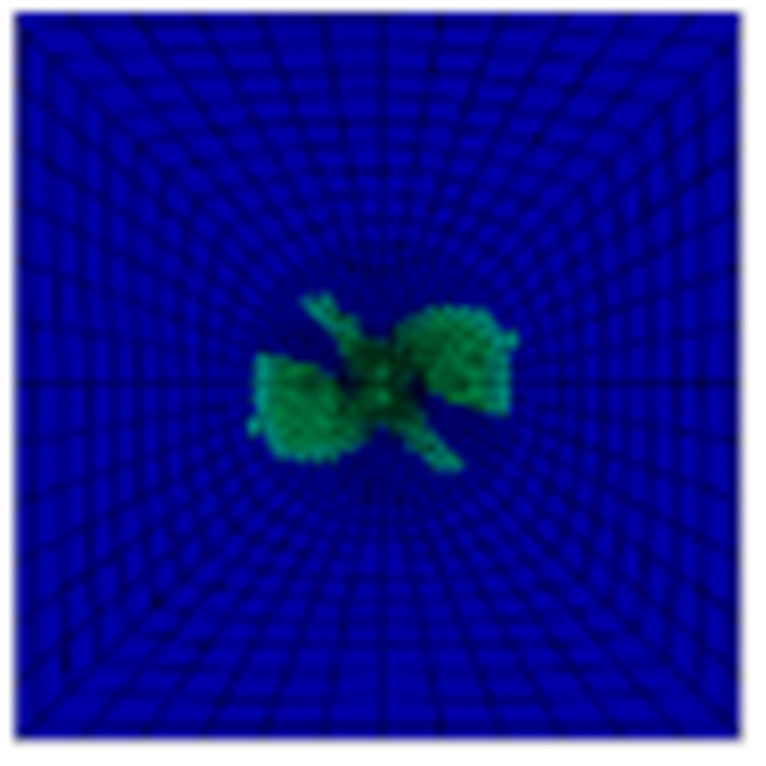	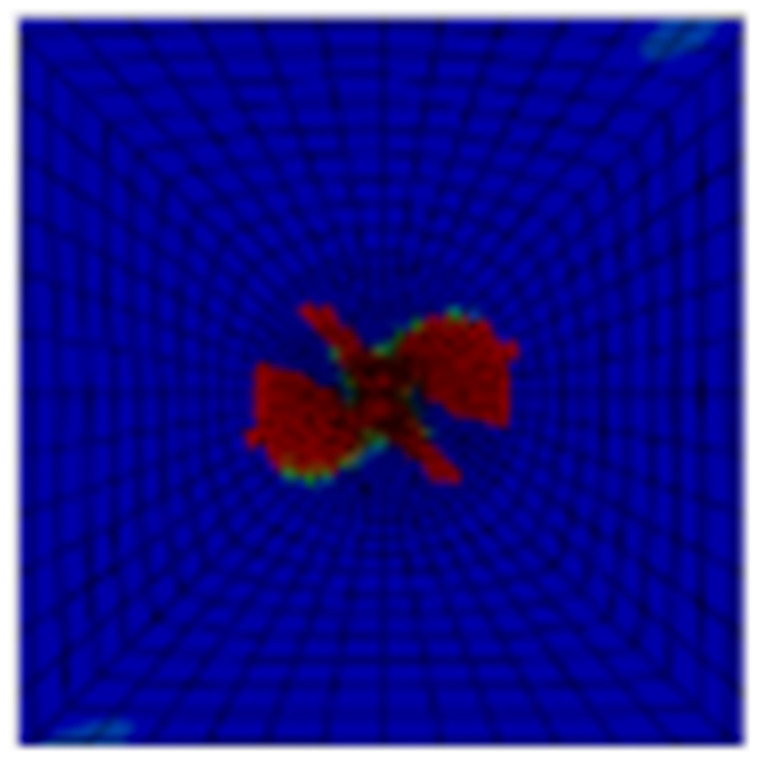	23	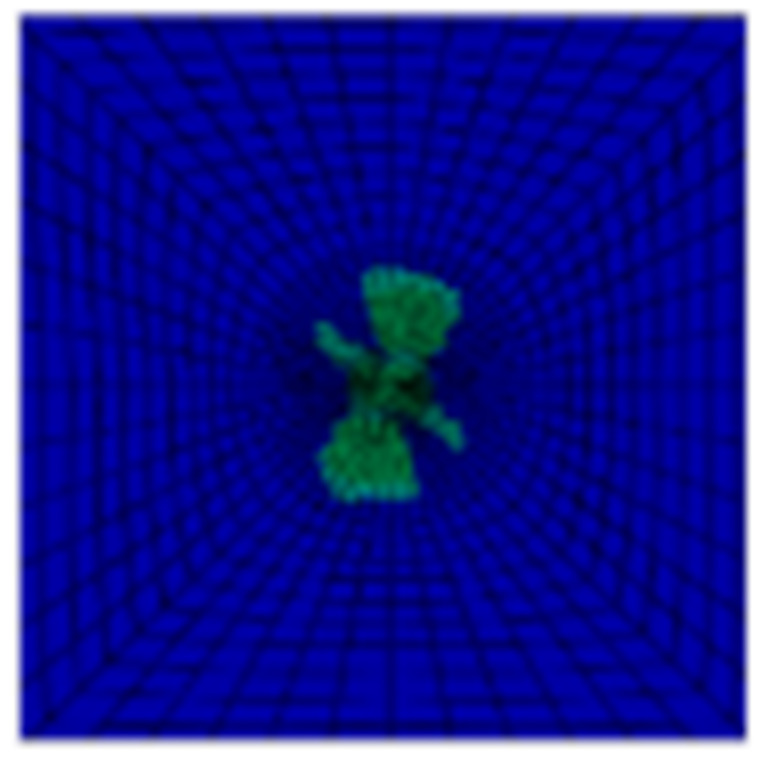	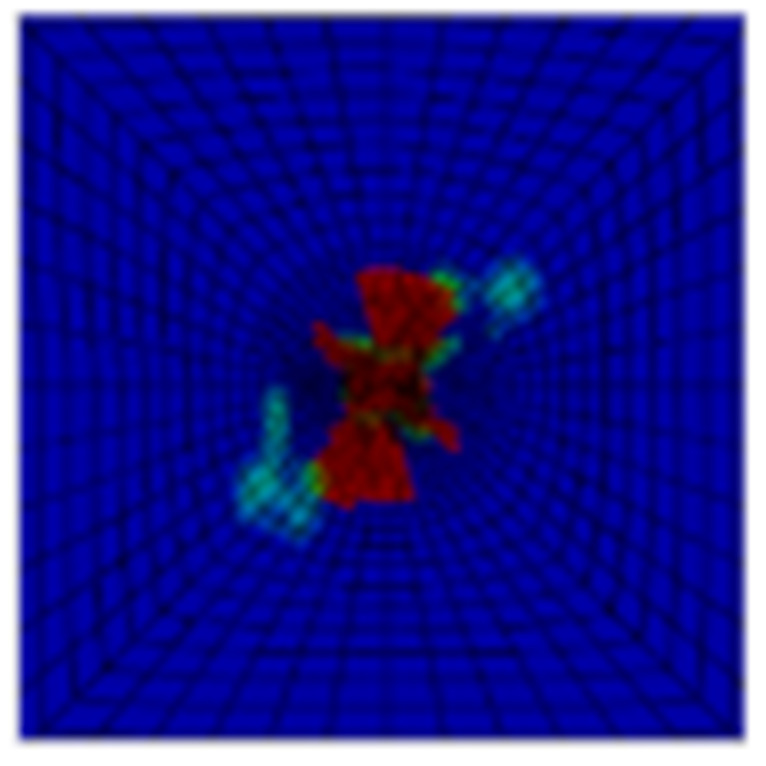
12	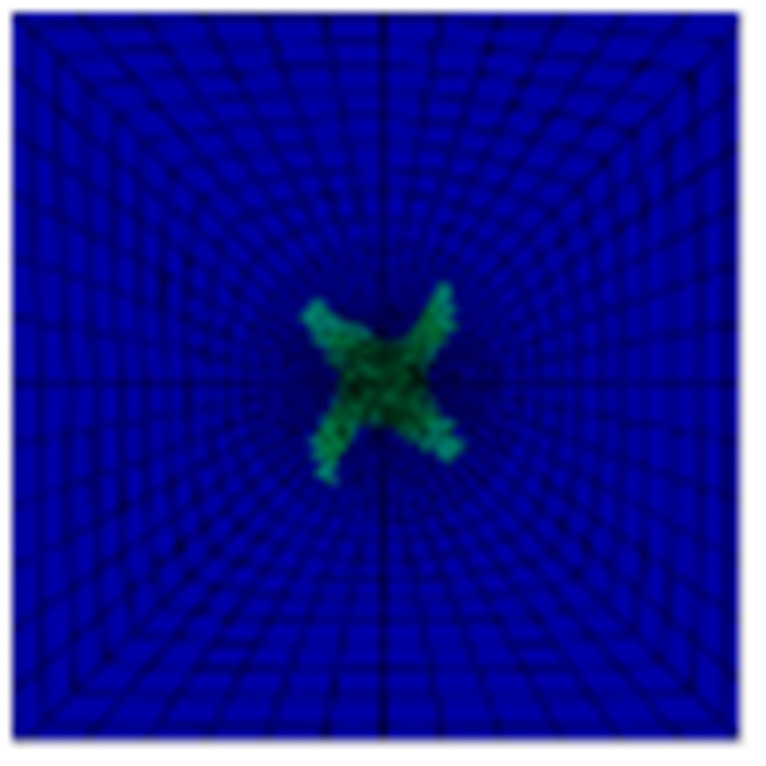	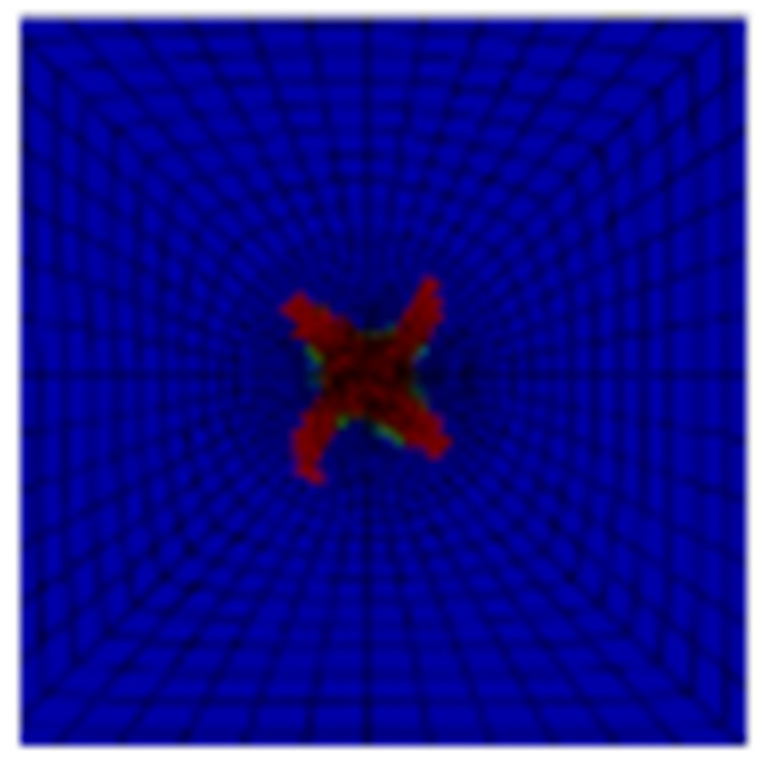	24	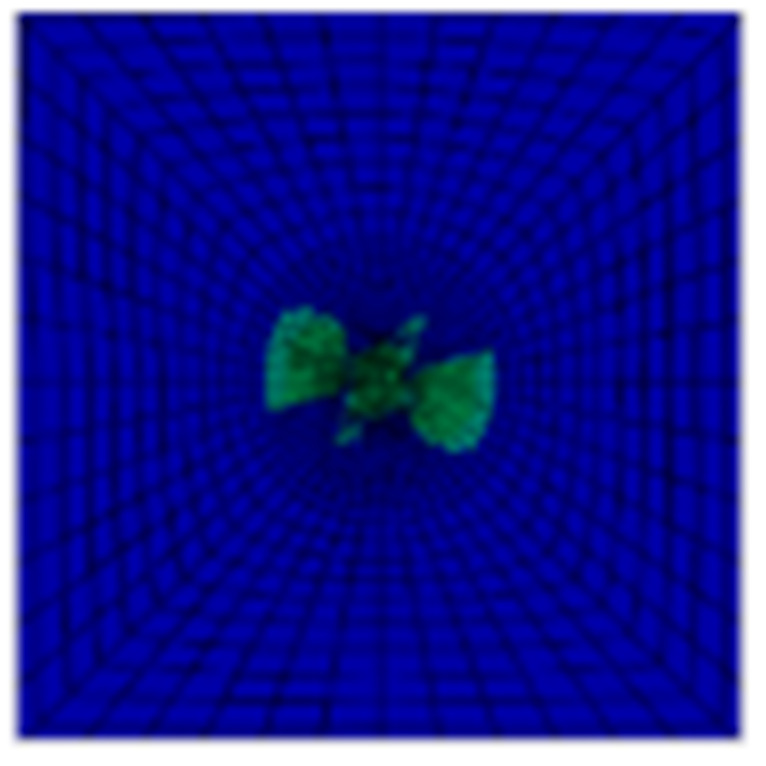	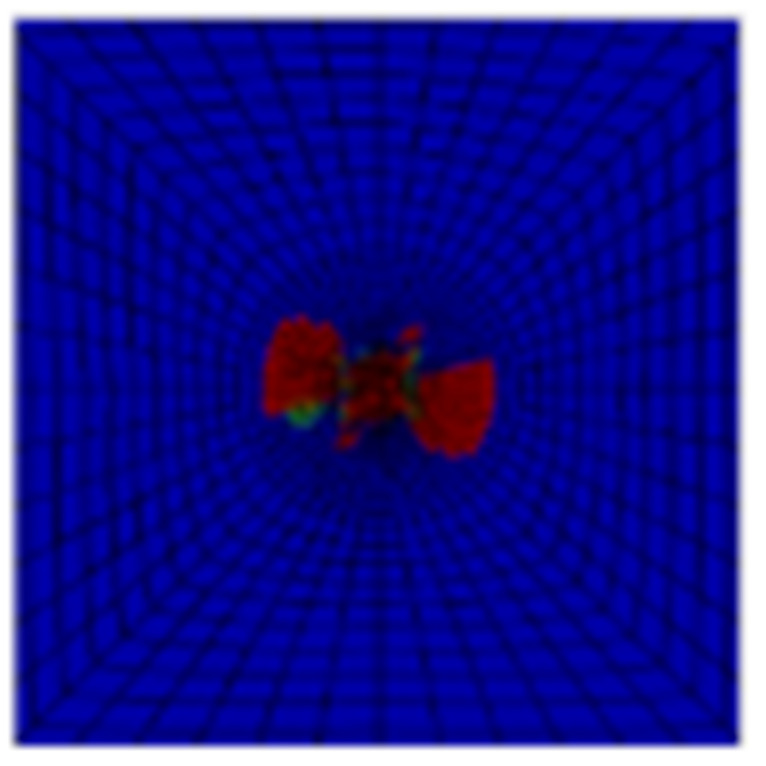

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
