# Peer review of "A Study on the In-Plane Shear-after-Impact Properties of CFRP Composite Laminates"

_materials, 2022, doi:10.3390/ma15145029_

Round 1

Reviewer 1 Report

Dear Authors,

Please find my comments in the attached pdf.

Thank you!

Reviewer 2 Report

The article presents a rather interesting engineering issue. The topics taken up in the paper are current, in the context of analyses conducted in this area. However, the paper requires a number of revisions to be considered for publication:

1) Please demonstrate the novelty of this paper, relative to many other thematically similar research papers - preferably at the introduction section.

2) Please include in the introduction papers, related to the described research topic, doi: 10.1016/j.compstruct.2019.01.073 ; 10.1002/nme.6757
3. in Section 5.2, Table 3 is missing from the first equation on the Hashin criterion the inclusion of the "alpha" coefficient in the formula, which determines the shear stress contribution.

4. In the notation of equation 10, the parameter D multiplied by the last term of the matrix is missing.

5. the conclusion should emphasize more the quantitative and qualitative evaluation of the study.
